# Chemosensor receptors are lipid-detecting regulators of macrophage function in cancer

Giulia Marelli[1,2], Nicolò Morina[1,2], Simone Puccio [3], Marta Iovino[2], Marta Pandini[1,2], Federica Portale[2], Mattia Carvetta[2], Divya Mishra[2], Elisabetta Diana[1,2], Greta Meregalli[1,2], Elvezia Paraboschi[4], Javier Cibella[5], Clelia Peano[5,6], Gianluca Basso[7], Gabriele De Simone[8], Chiara Camisaschi[8], Elena Magrini[9], Giulio Sartori[10], Elham Karimi[11], Piergiuseppe Colombo[1,12], Massimo Lazzeri [13], Paolo Casale[13], Lavinia Morosi[14], Giuseppe Martano [14,15], Rosanna Asselta[1,4], Eduardo Bonavita[1,16], Hiro Matsunami[17], Francesco Bertoni [10,18], Logan Walsh [11], Enrico Lugli [3] & Diletta Di Mitri [1,2] ✉

Infiltration of macrophages into tumors is a hallmark of cancer progression, and re-educating tumor-associated macrophages (TAMs) toward an antitumor status is a promising immunotherapy strategy. However, the mechanisms through which cancer cells affect macrophage education are unclear, limiting the therapeutic potential of this approach. Here we conducted an unbiased genome-wide CRISPR screen of primary macrophages. Our study confirms the function of known regulators in TAM responses and reveals new insights into the behavior of these cells. We identify olfactory and vomeronasal receptors, or chemosensors, as important drivers of a tumor-supportive macrophage phenotype across multiple cancers. In vivo deletion of selected chemosensors in TAMs resulted in cancer regression and increased infiltration of tumor-reactive CD8+ T cells. In human prostate cancer tissues, palmitic acid bound to olfactory receptor 51E2 (OR51E2) expressed by TAMs, enhancing their protumor phenotype. Spatial lipidomics analysis further confirmed the presence of palmitic acid in close proximity to TAMs in prostate cancer, supporting the function of this lipid mediator in the tumor micro-environment. Overall, these data implicate chemosensors in macrophage sensing of the lipid-enriched milieu and highlight these receptors as possible therapeutic targets for enhancing antitumor immunity.

Macrophages are highly plastic cells that exhibit a remarkable ability to change and adapt their functional characteristics in response to environmental cues. In vivo, macrophage activation occurs within a complex microenvironment that generates a range of diversified functional statuses within these cells. TAMs are considered to be alternatively activated cells able to sustain tumor growth and invasion, as well as promoting cancer therapy resistance and creating an immunosuppressive environment[1]. Given the influence that macrophages have on tumor progression, identifying factors that drive macrophage polarization and the molecular mechanisms involved is crucial for developing strategies to manipulate macrophage behavior and harness their potential in therapeutic interventions, including cancer immunotherapy. Therapeutic approaches that interfere with cancer–macrophage cross-talk hold promise for clinical applications; however, they show a limited efficacy that indicates the need for identification of novel regulators.

Recent studies of macrophage function revealed the expression of chemosensory-coding genes by macrophages, and expression of olfactory receptor 2 (OLFR2, also known as Or6a2) by vascular macrophages has been reported to drive inflammasome-dependent atherosclerosis[2]. Traditionally associated with the sense of smell, chemosensory receptors have been discovered to have physiological and pathological functions beyond the olfactory system and expression by sensory neurons[3–7]. In this regard, the involvement of chemosensory receptors in cancer-related processes has generated considerable interest. Chemosensory receptors are expressed by cancer cells in certain tissues and have been implicated in cell proliferation[8]. Moreover, recent studies have shown that engagement of OLFR78 (also known as Or51e2) modulates macrophage polarization in cancer[9], suggesting an unanticipated role for olfaction in the cross-talk between cancer and immune cells.

Unbiased genetic screenings have proven to efficiently identify unknown regulators of cell activation and function. In this context, genome-wide CRISPR screening has been successfully applied to both primary and lineage-specific immune cells[10–13] in vitro and in vivo[14–17] and has recently been applied to macrophages to identify host factors that confer resistance to infections and inflammation and determine key regulators of macrophage efferocytosis[18–21].

Here we performed a CRISPR screen of primary macrophages to identify regulators of macrophage re-education by tumor cells. This approach identified well-known genes involved in macrophage polarization and detected chemosensor-coding genes as mediators of the tumor–macrophage interaction. We showed that recognition of palmitic acid by human chemosensor OR51E2 conferred tumor-supportive and immunosuppressive functions on macrophages. Taken together, these results show that chemosensors expressed by macrophages mediate sensing of the lipid-enriched milieu, thus opening the way to strategies that target these receptors for enhancing antitumor immunity.

## Results

### Macrophages exposed to cancer cells adopt a protumoral state

To gain insights into the features of TAMs in prostate cancer, we employed the murine $Pten^{pc-/-} Trp53^{pc-/-}$ cancer model, which partially retraces the genetics of human invasive prostate cancer[21]. Profiling of the composition of the immune microenvironment by multiparametric flow cytometry revealed profound alterations when tumors were compared to healthy prostatic tissues (Fig. 1a and Extended Data Fig. 1a–h). In particular, the relative abundance of natural killer cells and dendritic cells was reduced in tumors, whereas polymorphonuclear neutrophils, CD4 regulatory T ($T_{reg}$) cells and antigen-activated $CD39^+CD8^+$ T cells showed increased infiltration (Extended Data Fig. 1c–g). Maturation and activation of $CD8^+$ and $CD4^+$ T cells were also reshaped in tumors (Extended Data Fig. 1d–g). The frequency of tumor-supporting $CD206^+MHCII^{+/-}$ macrophages was increased, and TAMs showed higher arginase 1 (ARG1) expression, in accordance with an immunosuppressive phenotype (Fig. 1b).

To achieve a more comprehensive understanding of the tumor microenvironment, we performed single-cell RNA sequencing (scRNA-seq) on prostate tissues of $Pten^{pc-/-} Trp53^{pc-/-}$ mice (Fig. 1c and Extended Data Fig. 1i,j). We reclustered myeloid cells according to CellTypist annotations (Extended Data Fig. 1h). The resulting macrophage clusters were then characterized based on their transcriptional profiles, drawing on phenotypes recently described[22]. Our analysis revealed distinct macrophage subpopulations, including proliferative Cl8 monocyte–macrophages, Cl4-5-7 lipid-laden macrophages, Cl2 angiogenic TAMs, Cl6 regulatory macrophages and Cl1 resident-tissue macrophages that expressed heat-shock proteins (previously described by Caronni et al.[22] as exhausted cells). Other clusters were defined by Cl3 interferon-related genes or by Cl0 Il-1β expression as inflammatory macrophages (Fig. 1d and Extended Data Fig. 1i). Trajectory inference identified a pseudotime progression, indicative of cell maturation and activation. In this trajectory, monocyte–macrophages appeared as the least differentiated population with a low pseudotime score, whereas Cl6 regulatory macrophages had a high pseudotime value, indicating a more advanced and specialized state (Fig. 1e). We then extended our analysis to a second prostate cancer model, in which $Pten^{-/-} Trp53^{-/-}$ tumor cells isolated from the prostate of $Pten^{pc-/-} Trp53^{pc-/-}$ mice were orthotopically injected in the right anterior lobes of 9-week-old C57Bl6/J mice (Fig. 1f and Extended Data Fig. 2a–f). The immune composition of the orthotopic tumors partially recapitulated the transgenic model, with some differences, including a higher abundance of neutrophils and $CD8^+$ T cells in the orthotopic tumors. Both tumor-promoting $CD206^+MHCII^{+/-}$ and $CD206^-MHCII^+$ inflammatory macrophages were increased in tumors, and TAMs showed higher ARG1 expression (Fig. 1f and Extended Data Fig. 2a–f). Bulk RNA sequencing of fluorescence-activated cell sorting (FACS)-sorted macrophages from orthotopic tumors revealed an altered transcriptional profile in tumor macrophages compared to macrophages infiltrating healthy tissues, with 5,632 genes significantly deregulated (Extended Data Fig. 2g). In accordance, gene set enrichment analysis (GSEA) performed on differentially expressed genes showed deregulation of biological pathways related to phagocytosis and inflammation and activation of pathways related to extracellular matrix reorganization, angiogenesis and wound healing (Fig. 1g).

**Fig. 1 | Profiling the immune cell infiltration in prostate cancer models.** **a**, Experimental scheme for the profiling of the composition of the immune microenvironment by multiparametric flow cytometry and scRNA-seq in the prostate of transgenic $Pten^{pc-/-} Trp53^{pc-/-}$ (tumor) and healthy (nontumor) mice. **b**, FACS analysis of macrophages in $Pten^{pc-/-} Trp53^{pc-/-}$ transgenic prostate compared to nontumor tissue. Quantification of immune infiltrating cells ($n = 4$ nontumor-bearing mice and $n = 6$ $Pten^{pc-/-} Trp53^{pc-/-}$ mice). Total macrophages were gated on $CD45^+$ cells. The percentages of $CD206^-MHCII^+$, $CD206^+MHCII^{+/-}$ and ARG1-expressing macrophages were gated on $F4/80^+CD11b^+$ cells. **c**, Uniform manifold approximation and projection (UMAP) of $CD45^+$ cells in $Pten^{pc-/-} Trp53^{pc-/-}$ transgenic prostate. Fourteen clusters characterized by lineage-specific and cluster-enriched genes were identified by integrated analysis. **d**, UMAP of scRNA-seq data from macrophages from $Pten^{pc-/-} Trp53^{pc-/-}$ transgenic prostate ($n = 2$). **e**, Trajectory analysis of macrophages using Monocle3 inference methods. **f**, FACS analysis of macrophages in murine prostate orthotopically injected with $Pten^{-/-} Trp53^{-/-}$ cells compared to macrophages in nontumor tissue ($n = 4$ mice per group). **g**, GSEA showing downregulated biological pathways (pathways down) and upregulated biological pathways (pathways up) in TAMs. The size of each dot indicates the number of enriched genes relative to the pathway of interest. The fraction of genes represents the proportion of the total number of genes in the pathway that were significantly enriched. **h**, Heat map illustrating all the differentially expressed genes according to bulk mRNA-seq from nonconditioned macrophages (untreated (Untr.), left) and macrophages exposed to conditioned media from $Pten^{-/-} Trp53^{-/-}$ cells (CM-tr., right). **i**, Volcano plot showing differentially expressed genes in CM-tr. macrophages compared to Untr. Genes are colored according to their $\log_2 FC$ value (blue, $\log_2 FC \leq -0.5$; red, $\log_2 FC \geq 0.5$). **j**, Proliferation of $CD8^+$ T cells exposed to supernatant from Untr. and CM-tr. macrophages: bar graph shows the number of divisions ($n = 3$ per group). **k**, Scratch assay: graph shows the quantification of distance (μm) covered by tumor cells over time after exposure to supernatant from Untr. or CM-tr. macrophages ($n = 9$ per group). **l**, FACS analysis of macrophages upon exposure to $Pten^{-/-} Trp53^{-/-}$ conditioned media: percentages of cells were gated on $F4/80^+CD11b^+$ cells ($n = 5$ per group). Statistical analyses were performed using two-tailed unpaired Student's $t$-test. Values are presented as the mean ± s.e.m. Schematic in **a** created using BioRender.com. NK, natural killer; DC, dendritic cells; PMN, polymorphonuclear neutrophils; Mono-Mac, monocyte–macrophages; Inflam-Macs, inflammatory macrophages; Angio-Macs, angiogenic macrophages; LA-Macs, lipid-laden macrophages; RTM-Macs, Reg-Macs, Cl6 regulatory macrophages; INF-Macs, macrophages defined by Cl3 interferon-related genes; LPS, lipopolysaccharide; ECM, extracellular matrix; Pos., positive; NA, not applicable; MFI, mean fluorescence intensity; NS, not significant.

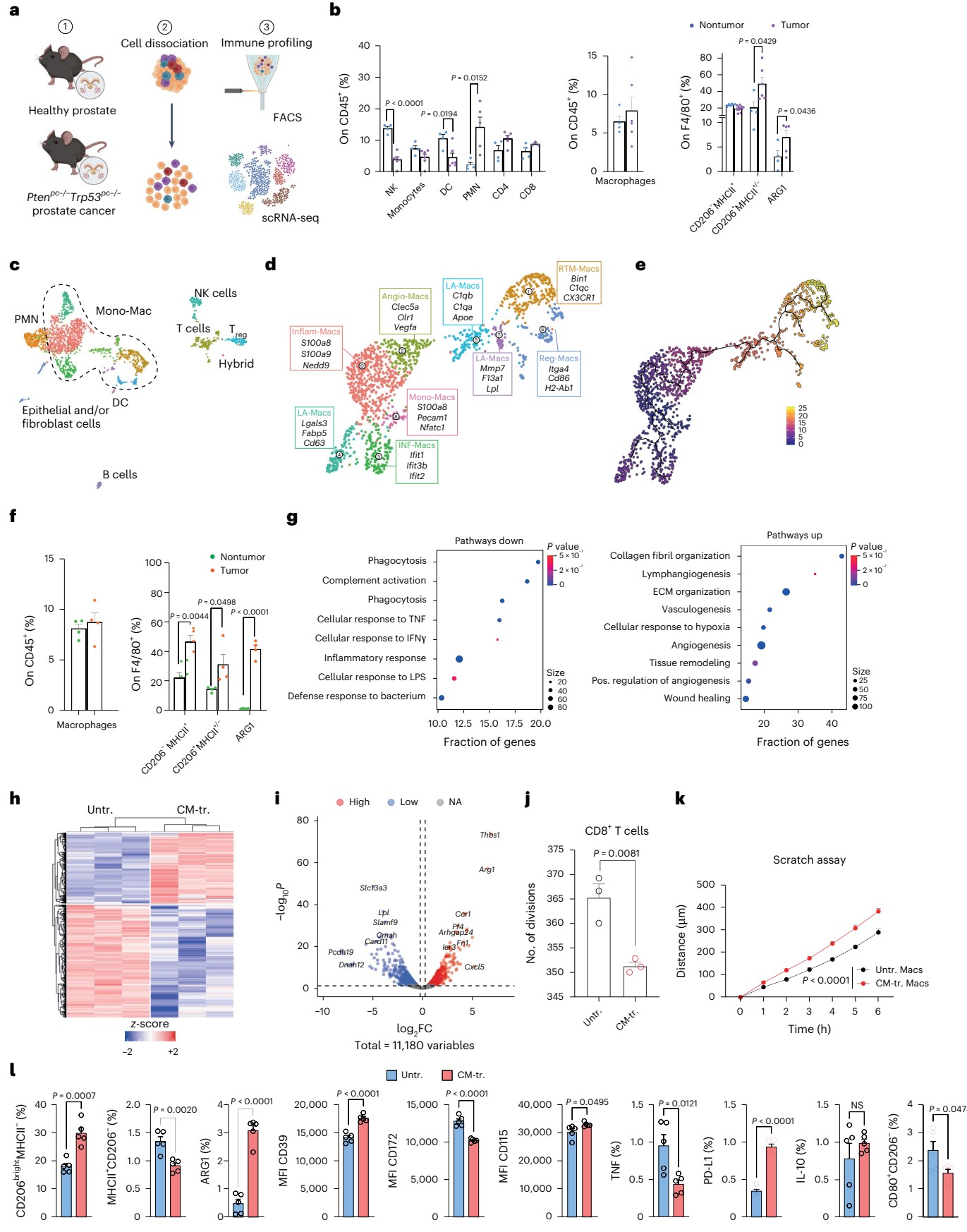

To obtain insight in the cross-talk between cancer cells and macrophages, we set up an in vitro system in which we activated bone-marrow-derived macrophages (BMDMs) by exposure to conditioned media isolated from $Pten^{-/-}Trp53^{-/-}$ tumor cells (Extended Data Fig. 2h). Bulk mRNA sequencing (mRNA-seq) analysis revealed enrichment of genes related to protumoral macrophage functions and immune suppression, including *Arg1*, *Trem1* and *Thbs1* (Fig. 1h,i). Functionally, the supernatant of macrophages conditioned in vitro decreased the proliferative rate of CD8+ T lymphocytes (Fig. 1l) and enhanced tumor cell migration (Fig. 1m). Supernatant of macrophages sorted from the prostate cancer model also showed CD8+-suppressive activity (Extended Data Fig. 2j). In accordance with transcriptional data, protein-level analysis confirmed an immunosuppressive phenotype in conditioned macrophages (CM-tr.), with upregulation of CD206, ARG1, CD39, CD115 and PD-L1 and reduced expression of MHCII, CD172, TNF and CD80 compared to controls (Fig. 1n). Notably, the production of ARG1 was highest in the CD206bright macrophages, confirming that CD206 expression is correlated with the immunosuppressive function of tumor-conditioned macrophages (Extended Data Fig. 2k). Finally, mRNA analysis demonstrated that CD206bright macrophages upregulated protumoral prototypic markers, including *Cd206*, *Il10* and *Arg1*, while downregulating classical proinflammatory markers *MhcII*, *Tnf* and *Il6* (Extended Data Fig. 2l). Taken together, these results demonstrate that macrophages exposed to prostate tumor cells acquire protumor profiles and behavior.

### Genome-wide CRISPR screen identifies TAM regulators

To obtain insight in cancer–macrophage cross-talk in an unbiased manner, we used genome-wide genetic screening to identify regulators exploited by the tumor to educate infiltrating macrophages in its favor. We performed CRISPR knockout (KO) screening on murine BMDMs isolated from Rosa26-Cas9 knock-in mice that constitutively express Cas9 endonuclease. Macrophages were transduced with the lentiviral GeCKO v2 Library B, which targets 20,611 genes and includes 62,804 gRNAs and 1,000 nontargeting (NT) gRNAs. After infection, cells were selected with puromycin, and $40 \times 10^6$ cells and $35 \times 10^6$ cells were recovered for experiment 1 and experiment 2, respectively. Infected cells were exposed to conditioned media derived from the $Pten^{-/-}Trp53^{-/-}$ prostate cancer cell line. We then FACS-sorted two populations based on the expression of MHCII and CD206 (CD206brightMHCII− and CD206−MHCII+ macrophages), which we sequenced by next-generation sequencing (NGS) (Fig. 2a,b). The screening was performed in biological replicate, and results were analyzed using the Model-based Analysis of Genome-wide CRISPR–Cas9 Knockout (MAGeCK) algorithm. The two experiments were consistent, showing a comparable distribution of the guides in the sorted populations (Fig. 2c). Positively regulated genes were targeted by single-guide RNAs (sgRNAs) that were enriched compared to the control, whereas negatively regulated genes were targeted by sgRNAs depleted compared to the control. We hypothesized that the comparison of CD206brightMHCII− cells to CD206−MHCII+ cells would identify regulators of the protumoral and proinflammatory macrophage phenotypes, respectively. We focused on gRNAs exhibiting negative regulation, with the goal of identifying genes whose silencing could effectively reprogram macrophages into the CD206−MHCII+ proinflammatory state (Fig. 2d). We ranked the genes by $P$ value using a threshold of $P < 0.005$ and excluded genes with a $\log_2$ fold change (FC) > −0.56 (Supplementary Tables 1 and 2). We classified the top 200 ranked genes into functional families based on their biological role (Extended Data Fig. 3a). Importantly, known inducers of the protumoral phenotype, including STAT6 and SPI1, were enriched among the negative regulators, confirming the efficiency of the screening strategy (Fig. 2d and Supplementary Table 1). To validate the results, we independently silenced *Stat6* in tumor-conditioned macrophages (Extended Data Fig. 3b and Fig. 2e). As expected, $Stat6^{-/-}$ macrophages showed impaired

capacity to acquire the protumoral phenotype compared to controls (Fig. 2f,g and Extended Data Fig. 3c). $Stat6^{-/-}$ macrophages also showed lower expression of CD39 and ARG1 (Extended Data Fig. 3d,e). Accordingly, transcriptional analysis showed that pivotal genes associated with the protumoral state of macrophages, including *Arg1*, *Il10* and *Fizz1*, were downregulated in $Stat6^{-/-}$ macrophages compared to controls, whereas proinflammatory genes, including *Nos2*, *Ifng* and *Il12*, were upregulated (Extended Data Fig. 3f). Functionally, $Stat6^{-/-}$ macrophages lost the ability to suppress CD8+ T cell proliferation (Fig. 2h) and to promote tumor cell migration in vitro (Fig. 2i and Extended Data Fig. 3g). Taken together these results demonstrate the efficiency of CRISPR screening to identify molecules that support the cross-talk between cancer cells and macrophages. Importantly, the screening identified multiple putative regulators (Supplementary Table 1) that deserve further investigation. Among the genes identified as significant, substantial numbers corresponded to olfactory and vomeronasal receptors (94 and 36, respectively), and we annotated these as chemosensors (Extended Data Fig. 3h). This enrichment suggests a prominent role for chemosensors in regulation of macrophage functions, potentially influencing their ability to detect and respond to environmental and tumor-derived signals. Recent studies have shown the expression of olfactory receptors in macrophages within cancer contexts. However, the exact mechanisms through which chemosensors operate in TAMs have remained largely unclear.

### Chemosensor-coding genes regulate macrophage functional status

Olfactory receptors and vomeronasal receptor genes have redundant functions and were first described in the nervous system as odorant and pheromones receptors. Chemosensors have recently been reported to have a pleiotropic role and to affect the activation of macrophages[2,8]. To explore the role of chemosensors in TAMs, we selected the top-ranked olfactory gene (*Olfr644*, also known as *Or51a43*, which had the lowest $P$ value) and the top-ranked vomeronasal gene (*Vmn2r29*, with the lowest $P$ value and highest $\log_2$FC) among the negative regulators and independently deleted them in macrophages (Supplementary Table 1). After genetic modification, macrophages were exposed to the tumor-conditioned media and analyzed by flow cytometry (Extended Data Fig. 4a). Genetic deletion of *Olfr644* and *Vmn2r29* conferred a proinflammatory phenotype on macrophages, as evidenced by decreased expression of CD206, decreased abundance of CD206brightMHCII− cells and increased abundance of CD206−MHCII+ cells, and lower expression of CD39 and ARG1 (Fig. 3a–e). NT guides used as controls did not alter the macrophage phenotype (Extended Data Fig. 4b). Transcriptional analysis of $Olfr644^{-/-}$ and $Vmn2r29^{-/-}$ macrophages by bulk mRNA-seq and quantitative PCR with reverse transcription (RT-qPCR) confirmed the acquisition of a proinflammatory status (Fig. 3f,g and Extended Data Fig. 4c). As a further control, we selected chemosensor-coding genes among the positive regulators (*Olfr229*, also known as *Or8g2*, and *Vmn1r87*) and an additional olfactory gene among the negative regulators (*Olfr192*, also known as *Or5h24-ps1*). In accordance with our hypothesis, deletion of *Olfr229* and *Vmn1r87* did not alter either the transcriptome (Extended Data Fig. 4c) or protein expression (Extended Data Fig. 4d,e) in macrophages, whereas deletion of negative regulator *Olfr192* resulted in decreased levels of CD206 and increased levels of MHCII in macrophages. mRNA bulk analysis of $Olfr644^{-/-}$ and $Vmn2r29^{-/-}$ macrophages revealed shared transcriptional changes, suggesting a common downstream signaling mechanism for these two chemosensors (Fig. 3f,g). Ingenuity Pathway Analysis of lentiGuide-Puro (LGP, control) and conditioned-media-exposed macrophages predicted upregulation of HIF1A and downregulation of its gene targets, effects that were absent from KO macrophages (Extended Data Fig. 4f,g). These data indicate a potential role for *Hif1A* in receptor-mediated signaling. In addition, Ingenuity Pathway Analysis predicted the activation of other transcription factors, including MYC and HIC1,

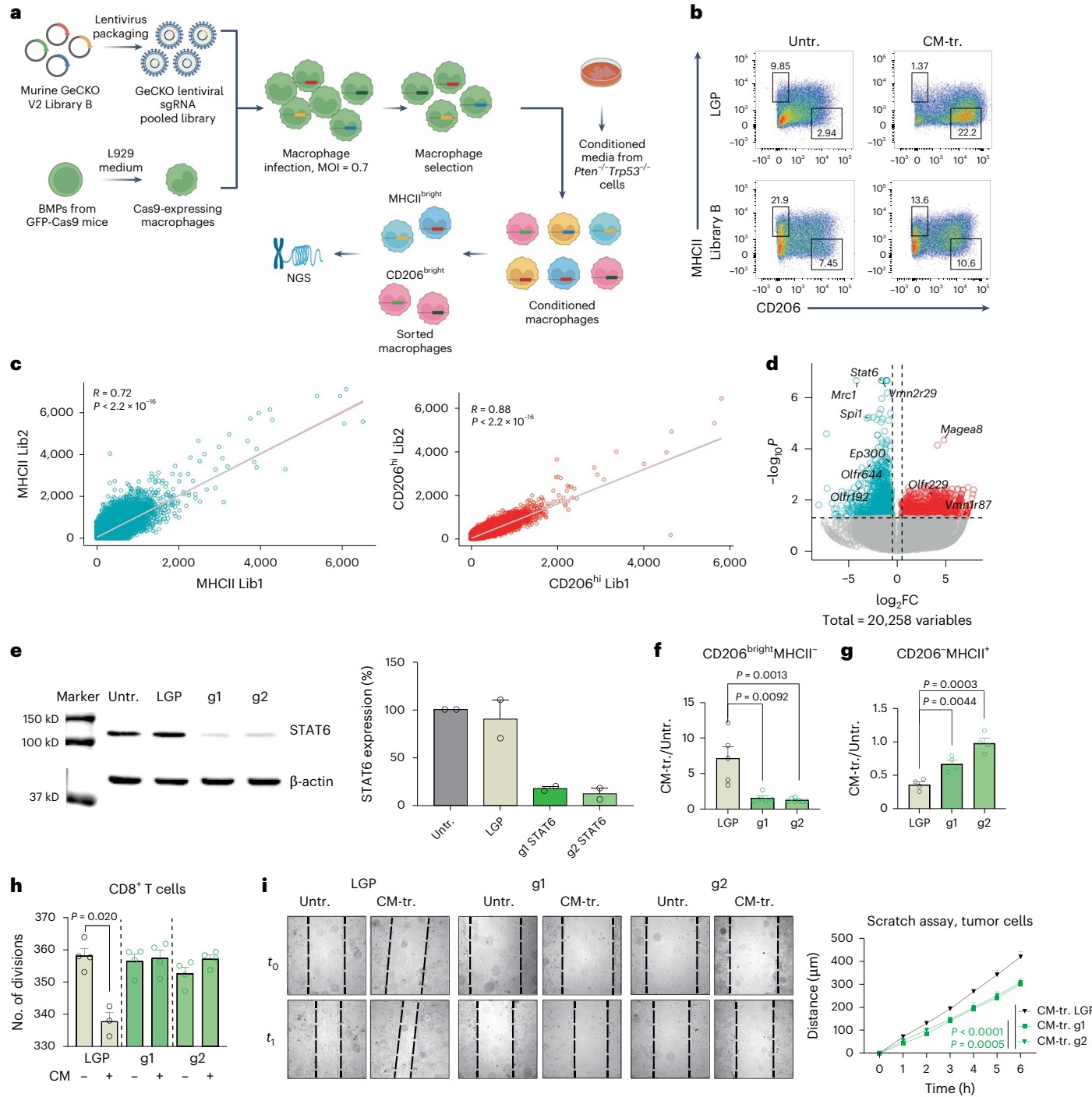

**Fig. 2 | CRISPR–Cas9 screening of primary mouse macrophages to identify TAM regulators. a**, Experimental scheme of genome-wide CRISPR–Cas9 Knockout GecKO v2 Library B screening in primary murine macrophages. **b**, Representative plot of backbone (LGP) or library-B-infected macrophages exposed or not to conditioned media from $Pten^{-/-}Trp53^{-/-}$ tumor cells. **c**, Two independent experiments were performed. Graphs show the correlations between the distribution of the guides found in the CD206⁻MHCII⁺ population (MHCII) and in the CD206^bright^MHCII⁻ population (CD206) from the two experiments. Lib1, library 1; Lib2, library 2. **d**, Volcano plot showing genes related to the differentially enriched sgRNA guides from CD206⁻MHCII⁺ versus CD206^bright^MHCII⁻ cells. Negative regulators of the CD206^bright^MHCII⁻ population are shown in light blue, and positive regulators are shown in red. $\log_2$FC ± 0.56, $P < 0.005$. Statistical analyses and comparisons from NGS output

were performed with MAGeCK. **e**, Western blot analysis showing the percentage of expression of total STAT6. Two independent sgRNA guides (g1 and g2) were utilized to silence *Stat6* in macrophages ($n = 2$). **f,g**, FACS analysis of control (LGP) and *Stat6*-silenced (g1 and g2) macrophages following exposure to $Pten^{-/-}Trp53^{-/-}$ conditioned media, with events gated on F4/80⁺CD11b⁺ cells: LGP $n = 5$, g1 $n = 5$, g2 $n = 6$ (**f**); LGP $n = 4$, g1 $n = 4$, g2 $n = 4$ (**g**). **h**, Proliferation of CD8⁺ T cells exposed to supernatant from Untr. and CM-tr. macrophages: bar graph shows the number of divisions. **i**, Scratch assay: graph and curves showing the distance (µm) covered by tumor cells over time after exposure to supernatant from Untr. or CM-tr. macrophages ($n = 8$). Statistical analyses were performed using two-tailed unpaired Student's *t*-test. Values are presented as the mean ± s.e.m. All replicates represent biological replicates. Schematic in **a** created using BioRender.com.

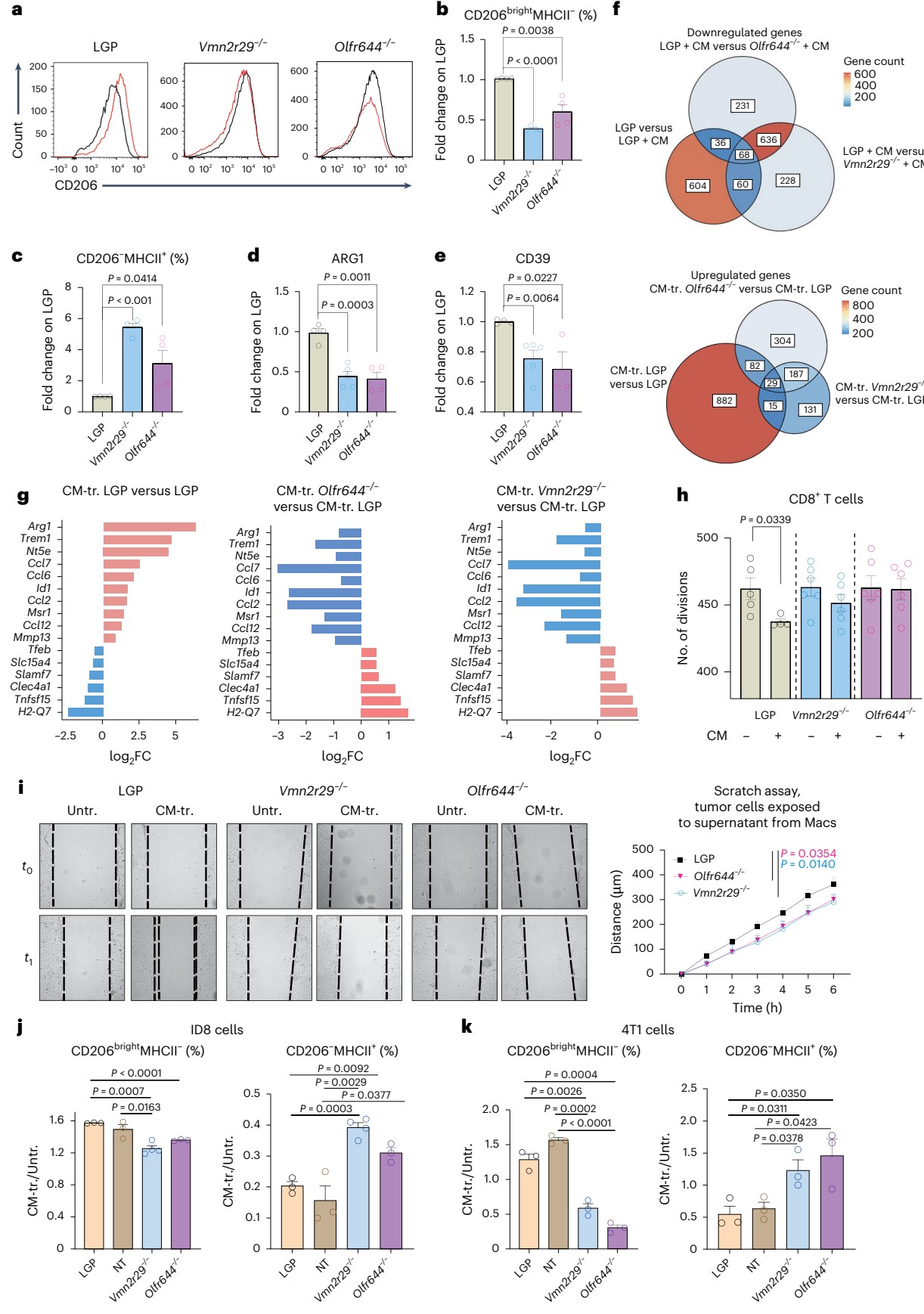

**Fig. 3 | Genetic deletion of selected chemosensors on tumor-conditioned macrophages. a–e**, FACS analysis of macrophages in the absence (LGP) or presence (OLFR644⁻/⁻) of *Olfr644* deletion and *Vmn2r29* deletion (*Vmn2r29*⁻/⁻). Macrophages were exposed to *Pten*⁻/⁻ *Trp53*⁻/⁻ conditioned media. Events were gated on F4/80⁺CD11b⁺ cells. Bar graphs show ratios between conditions: mean fluorescence intensity of CD206 (**a**); percentages of CD206^bright^MHCII⁻ (**b**) and CD206⁻MHCII⁺ (**c**) cells gated on F4/80⁺CD11b⁺ cells (LGP *n* = 4, *Vmn2r29*⁻/⁻ *n* = 4, *Olfr644*⁻/⁻ *n* = 4); and percentages of ARG1⁺ (**d**) and CD39⁺ (**e**) cells gated on F4/80⁺CD11b⁺ cells (LGP *n* = 4, *Vmn2r29*⁻/⁻ *n* = 5, *Olfr644*⁻/⁻ *n* = 3). **f**, Venn diagrams showing common and specific genes among the differentially expressed genes calculated for the three conditions: LGP + *Pten*⁻/⁻ *Trp53*⁻/⁻ conditioned media versus LGP; OLFR644 + *Pten*⁻/⁻ *Trp53*⁻/⁻ conditioned media versus LGP + *Pten*⁻/⁻ *Trp53*⁻/⁻ conditioned media; VMN2R29 + *Pten*⁻/⁻ *Trp53*⁻/⁻ conditioned media versus LGP + *Pten*⁻/⁻ *Trp53*⁻/⁻ conditioned media. **g**, Graphs showing change in expression of selected genes among differentially expressed genes from the three comparisons (red, upregulated; blue, downregulated). **h**, Proliferation of CD8⁺ T cells exposed to supernatant from Untr. and CM-tr. macrophages: the bar graph shows the number of divisions (LGP Untr. *n* = 5, LGP CM-tr. *n* = 4, *Vmn2r29*⁻/⁻ Untr. or Cm-tr. *n* = 6, *Olfr644*⁻/⁻ Untr. or Cm-tr. *n* = 6). **i**, Scratch assay: graph and curves showing the distance (μm) covered by tumor cells over time after exposure to supernatant from Untr. or CM-tr. macrophages (LGP *n* = 6, VMN2R29 *n* = 6, OLFR644 *n* = 6). **j,k**, Flow cytometry analysis to assess the impact of chemosensor gene silencing on macrophage phenotypes. *Olrf644*⁻/⁻ and *Vmn2r29*⁻/⁻ macrophages were compared to control macrophages (LGP) after exposure to conditioned media from ovarian ID8 cancer cells (LGP *n* = 3, NT *n* = 3, *Vmn2r29*⁻/⁻ *n* = 4, *Olfr644*⁻/⁻ *n* = 3) (**j**) or breast 4T1 cancer cells (LGP *n* = 3, NT *n* = 3, *Vmn2r29*⁻/⁻ *n* = 3, *Olfr644*⁻/⁻ *n* = 3) (**k**). Events were gated on F4/80⁺CD11b⁺ cells. Statistical analyses were performed using two-tailed unpaired Student's *t*-test. Values are presented as the mean ± s.e.m. All replicates represent biological replicates.

which were downregulated in chemosensor-deficient macrophages (Extended Data Fig. 4f,g). Functionally, the impairment in CD8⁺ cells proliferation was lost if cells were silenced for *Olfr644* or *Vmn2r29* (Fig. 3h), and cancer cell migration was impaired when tumor cells were exposed to supernatant from *Olfr644*⁻/⁻ and *Vmn2r29*⁻/⁻ macrophages compared to control (Fig. 3i and Extended Data Fig. 4h). Notably, exposure of macrophages to supernatant from ovarian (ID8) and breast (4T1) cancer cells promoted their re-education toward the protumoral CD206^bright^MHCII⁻ phenotype (Extended Data Fig. 5a,b), with CD206^Bright^ cells being the most responsible for ARG1 production upon conditioning (Extended Data Fig. 5c). Also, in these models, genetic deletion of *Olfr644* and *Vmn2r29* resulted in re-education toward the CD206⁻MHCII⁺ proinflammatory status (Fig. 3j,k and Extended Data Fig. 5d,e). Genetic deletion of *Stat6* also resulted in macrophage re-education in this context (Extended Data Fig. 5f,g).

## Deletion of chemosensors in macrophages affects tumor growth

The results of the genome-wide CRISPR screening and in vitro validation prompted us to evaluate the impact of chemosensors expressed by macrophages in an in vivo model of cancer. We employed *Olfr644*⁻/⁻ and *Vmn2r29*⁻/⁻ primary macrophages to treat mice bearing subcutaneous *Pten*⁻/⁻ *Trp53*⁻/⁻ tumors. Genetic deletion of chemosensors was performed on BMDMs isolated from Cas9 mice, which constitutively express GFP as a reporter gene and may thus be traced once injected. This in vivo approach is based on subsequent infusion of macrophages by intravein injection and results in macrophage migration to the tumor bed and partial replacement of tissue macrophages, as reported previously[22] (Fig. 4a). Intravenous infusion of macrophages as therapeutic agents has been employed in various studies[23–26]. We reasoned that the absence of the selected chemosensors should result in the macrophages being unable to acquire a protumoral status, instead conferring antitumoral functions. We injected mice subcutaneously with *Pten*⁻/⁻ *Trp53*⁻/⁻ tumor cells and intravenously infused macrophages genetically deleted for chemosensors or wild-type cells treated with empty vector (LGP) twice per week for 3 weeks (Fig. 4a). We observed significant reductions in tumor volume throughout the treatment time and at the endpoint in mice injected with both *Olfr644*⁻/⁻ and *Vmn2r29*⁻/⁻ macrophages compared with controls, which instead showed continued tumor growth (Fig. 4b). The composition of the tumor microenvironment was analyzed by flow cytometry, and tumor-infiltrating macrophages derived from infused cells were detected on the basis of GFP expression (Fig. 4c). The relative abundance of the total TAM fraction was not affected; however, we observed a reprogramming of tumor-infiltrating GFP⁺ macrophages, favoring a proinflammatory phenotype (Fig. 4d and Extended Data Fig. 6a). We did not observe any alterations in the GFP⁻ macrophages (Extended Data Fig. 6b,c), but there were reductions in the frequencies of other myeloid populations, including monocytes and dendritic cells (Extended Data Fig. 5d–f). In tumors, we detected a higher abundance of proliferating CD8⁺CD39⁺ cancer-specific T lymphocytes and a decrease in the frequency of CD4⁺FoxP3⁺ T_reg cells following infusion of GFP⁺ *Olfr644*⁻/⁻ and GFP⁺ *Vmn2r29*⁻/⁻ macrophages (Fig. 4e). Finally, we evaluated the responsiveness of the splenocytes to tumor antigens by performing splenocyte restimulation ex vivo using mytomicin-C-treated *Pten*⁻/⁻ *Trp53*⁻/⁻ cells to supply tumor antigens. Enzyme-linked immunosorbent assay (ELISA) and FACS analysis demonstrated significant increases in IFNγ production by splenocytes and CD8⁺CD39⁺ T cells, respectively, in mice infused with GFP⁺ *Olfr644*⁻/⁻ and GFP⁺ *Vmn2r29*⁻/⁻ macrophages (Extended Data Fig. 6g–i). This was indicative of systemic activity of the macrophages or recirculation of tumor-infiltrating T cells from the tumor to the spleen, as previously reported[23]. We validated these findings by injecting LGP, *Vmn2r29*⁻/⁻ or *Olfr644*⁻/⁻ macrophages along with *Pten*⁻/⁻ *Trp53*⁻/⁻ tumor cells directly into the anterior lobe of the prostate of control mice (Fig. 4f). At sacrifice, mice injected with *Vmn2r29*⁻/⁻ or *Olfr644*⁻/⁻ showed reduced tumor volumes compared to controls (Fig. 4g). In accordance with the systemic infusion model, we observed reshaping of the tumor microenvironment, with an increase in CD206⁻MHCII⁺ macrophages and a decrease in CD206⁺MHCII⁻ macrophages (Fig. 4h), an increase in CD8⁺CD39⁺ cancer-specific T lymphocytes and a decrease

**Fig. 4 | Infusion of genetically modified macrophages in a prostate cancer model. a**, Experimental scheme. Mice were injected twice per week with LGP-Macs, *Olfr644*⁻/⁻ Macs or *Vmn2r29*⁻/⁻ Macs (*n* = 7 mice per group). **b**, Tumor growth expressed as a percentage of the initial volume. **c**, Representative FACS plot of GFP⁻ and GFP⁺ tumor-infiltrating macrophages. **d,e**, Results of FACS analysis to determine the immune infiltrate in mice injected with LGP, OLFR644-KO or VMN2R29-KO, showing: percentages of cells gated on GFP⁺F4/80⁺CD11b⁺ cells (LGP *n* = 7, VMN2R29-KO *n* = 8, OLFR644-KO *n* = 8) (**d**); and percentages of proliferating CD39⁺ cells gated on CD8⁺ T cells and FoxP3⁺CD25⁺ T_reg cells gated on CD4⁺ T lymphocytes (LGP *n* = 7, VMN2R29-KO *n* = 7, OLFR644-KO *n* = 8) (**e**). **f**, Experimental scheme: mice were contextually injected orthotopically with *Pten*⁻/⁻ *Trp53*⁻/⁻ and with LGP-Macs (*n* = 4 mice), *Olfr644*⁻/⁻ Macs (*n* = 5 mice) or *Vmn2r29*⁻/⁻ Macs (*n* = 4 mice). **g**, Tumor volumes calculated at sacrifice. **h,i**, FACS analysis to determine the immune infiltrate in LGP, OLFR644-KO or VMN2R29-KO injected mice; percentages of cells gated on GFP⁺F4/80⁺CD11b⁺ cells (**h**) and percentages of CD39⁺ cells gated on CD8⁺ T cells and T_reg cells gated on CD4⁺ T lymphocytes (**i**) are shown. **j**, Experimental scheme: mice were injected twice per week with LGP-Macs or *Olfr644*⁻/⁻ Macs and with anti-CD8 antibody or isotype control. **k,l**, Tumor growth expressed as a percentage of the initial volume (**k**) and tumor volumes at the day of sacrifice (**l**) (LGP isotype *n* = 4, LGP anti-CD8 *n* = 7, OLFR644 isotype *n* = 8, OLFR644 anti-CD8 *n* = 8). Statistical analyses were performed using two-tailed unpaired Student's *t*-test. Values are presented as the mean ± s.e.m. All replicates represent biological replicates. Schematic in **a** created using BioRender.com.

in the frequency of T_reg cells (Fig. 4i). FACS analysis demonstrated a significant increase in IFNγ production by splenic CD8⁺CD39⁺ T cells in mice injected with chemosensor-KO macrophages compared to controls (Extended Data Fig. 6j). To assess the importance of antitumor immunity, we depleted CD8⁺ T lymphocytes and infused LGP-control or *Olfr644*⁻/⁻ macrophages into tumor-bearing mice (Fig. 4j and Extended

Data Fig. 6k,l). Endpoint analysis revealed that the absence of CD8⁺ T lymphocytes in mice infused with *Olfr644*⁻/⁻ macrophages increased tumor growth, although not to the same extent as in LGP-injected mice (Fig. 4k,l). This finding indicates that CD8⁺ T cells partially mediate the antitumor effects of *Olfr644*⁻/⁻ macrophages. However, other mechanisms may be involved.

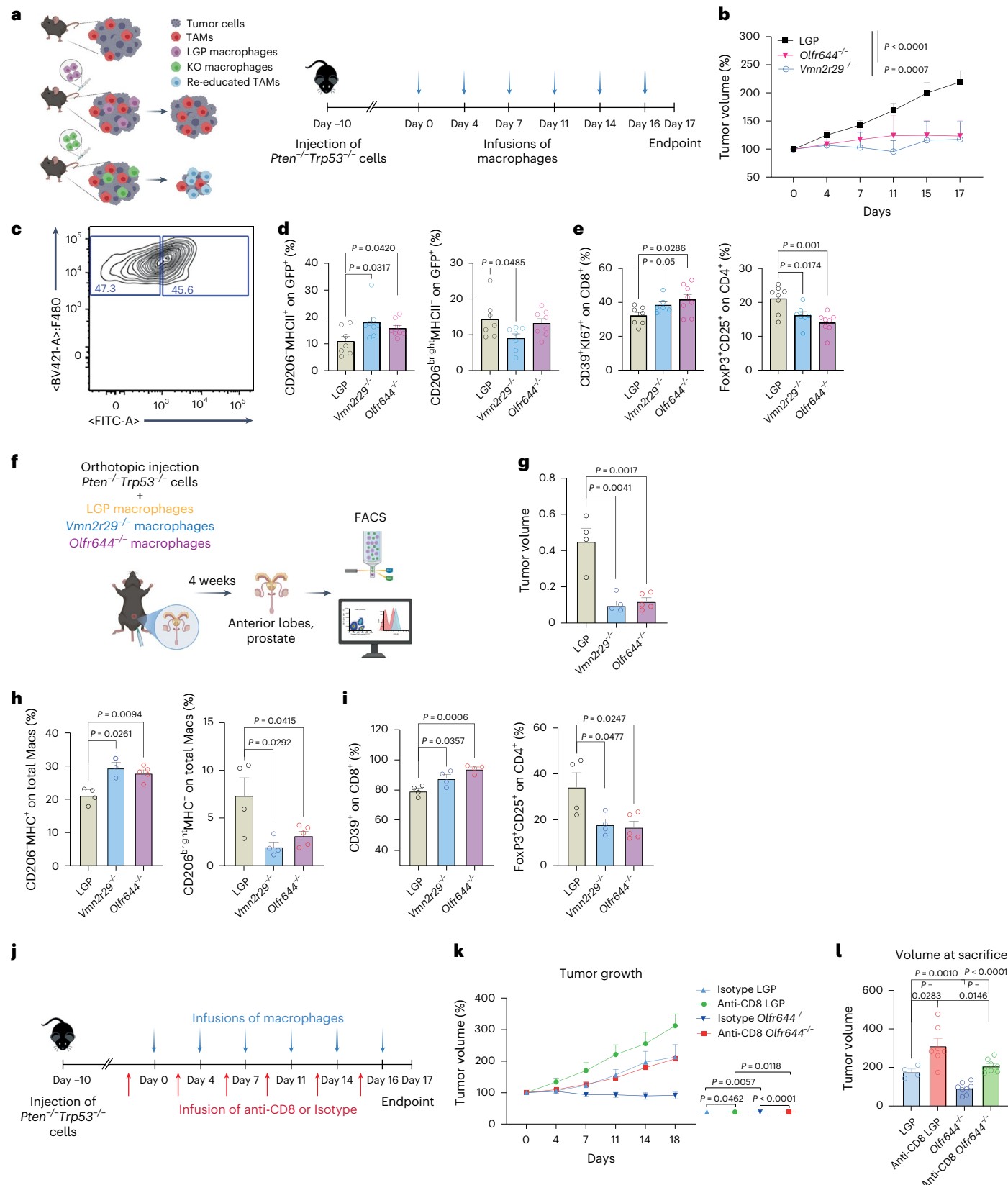

## Pharmacological inhibition of OLFR644 affects tumor growth

Motivated by the evidence described above, we used a pharmacological approach to inhibit the activity of chemosensors on macrophages in our cancer model. Among predicted interactors of chemosensors, all-*trans* retinoic acid (13-cRA) has been identified as an antagonist of OR51E2, a known human olfactory receptor[24]. In light of the affinity between human and murine olfactory receptors, we tested the possibility that 13-cRA would hinder OLFR644 activity in murine macrophages. In vitro, administration of 13-cRA prevented re-education of cells exposed to tumor-conditioned media (Fig. 5a). Notably, the effect of 13-cRA was partially mitigated by deletion of *Olfr644*, indicating that its role is in part dependent on the expression of the chemosensor on macrophages (Fig. 5a). To corroborate these results, we performed an in vivo experiment in which we pretreated macrophages with 13-cRA before injecting them into mice harboring the tumor (Fig. 5b). Mice were either left untreated or infused with dimethyl sulfoxide (DMSO; vehicle)-treated GFP+ macrophages or 13-cRA-treated GFP+ macrophages. We observed a significant reduction in tumor volume throughout time and at the endpoint in mice injected with 13-cRA-treated GFP+ macrophages (Fig. 5c). Flow cytometry analysis of tumor-infiltrating GFP+ macrophages (Fig. 5d) revealed that treatment with 13-cRA increased their MHCII expression but did not lower their CD206 expression, indicating partial polarization toward an inflammatory phenotype (Fig. 5e). The relative abundances of the total TAM fraction, monocytes and dendritic cells were diminished in the 13-cRA-treated group with respect to both controls (Extended Data Fig. 7a–d). Importantly, CD8+CD39+ T cells were more abundant in 13-cRA treated mice, whereas the abundance of $T_{reg}$ cells was decreased (Fig. 5f). ELISA and flow cytometry analysis showed increased production of IFNγ from splenocytes and CD8+CD39+ T cells in the 13-cRA-treated group with respect to controls (Fig. 5g). We conducted the same in vivo experiment by pretreating LGP (control) or *Olfr644*−/− macrophages with 13-cRA or DMSO. Administration of 13-cRA-treated LGP macrophages, 13-cRA-treated *Olfr644*−/− macrophages and *Olfr644*−/− macrophages all resulted in a similar reduction in tumor growth compared to LGP untreated macrophages (Fig. 5h). CD206 expression in GFP+ macrophages was reduced across all treatment groups compared to untreated LGP-control macrophages (Fig. 5i), and we observed increased abundance of CD8+CD39+ cancer-specific T lymphocytes and decreased frequency of $T_{reg}$ cells across all treatment groups compared to the untreated LGP control (Fig. 5j). Finally, no alterations were detected in GFP− macrophages (Extended Data Fig. 7f,g). However, we observed a reduction in the frequency of monocytes, whereas that of dendritic cells remained unchanged (Extended Data Fig. 7h–j). Taken together, these findings indicate that modulation of selected chemosensors controls TAM function in vivo.

## OR51E2 is expressed by human TAMs

The evidence reported above prompted us to explore the expression and role of chemosensors in the human context. Genes coding for olfactory receptors compose the largest gene family in the human genome and determining their orthologous relationships is complex.

To identify chemosensors of potential relevance in the human context, we analyzed a human gene expression dataset from patients with prostate cancer and derived differentially expressed genes between tumoral tissues and normal tissues to identify olfactory genes possibly correlated with disease, as no intact vomeronasal receptors exist in human. Notably, OR51E1 (olfactory receptor E1 belonging to family 51) and OR51E2 (olfactory receptor E2 belonging to family 51) are known to be upregulated in prostate cancer tissues[27,28]. Analysis of a dataset from The Cancer Genome Atlas using GEPIA2 confirmed that OR51E2 and OR51E1 were significantly overexpressed in prostate cancer tissues, with OR51E2 being the more deregulated of the two (Fig. 6a and Extended Data Fig. 8a). Importantly, immunofluorescence analysis of human prostate cancer sections confirmed that OR51E2 is expressed by tumor-infiltrating macrophages (Fig. 6b and Extended Data Fig. 8b). In addition, we were able to detect OR51E2 in macrophages derived from the monocytic THP1 cell line and primary monocyte-derived macrophages from healthy donors (Fig. 6c and Extended Data Fig. 8c). We genetically silenced the gene encoding the OR51E2 receptor in THP1-derived macrophages and exposed cells to the supernatant from PC3 human prostate cancer cells (Fig. 6c). Silencing of *OR51E2* favored polarization of tumor-conditioned macrophages toward a proinflammatory phenotype, while hindering acquisition of a protumoral status (Fig. 6d). Notably, in accordance with human data, silencing of the orthologous murine gene *Olfr78* (*Or51e2*) similarly increased levels of MHCII and decreased expression of CD206 and ARG1 in murine BMDMs exposed to conditioned media from *Pten*−/− *Trp53*−/− tumor cells (Extended Data Fig. 8d).

## Palmitic acid activates OR51E2 and shapes macrophage function

Olfactory receptors are known to be activated by a plethora of exogenous odorants, but recent evidence has identified selected fatty acids as endogenous ligands[2,29]. In an attempt to identify ligands that engage OR51E2 in human macrophages, we performed a lipidomic analysis of supernatant from the PC3 prostate cancer cell line. Among the most abundant molecules, we detected the presence of palmitic acid, previously reported as a predicted ligand of OR51E2 (refs. 30,31) (Fig. 7a). To investigate the engagement of olfactory receptor OR51E2 by ligands, we employed the Dual-Glo Luciferase Assay System. Sodium acetate and sodium propionate were used as positive controls. Hana3A cells were transfected with OR51E2 and subsequently stimulated with palmitic acid, sodium acetate or sodium propionate to induce cAMP response element (CRE)–luciferase expression. Luminescence was measured 4 h poststimulation, revealing a detectable response to all three fatty acids (Fig. 7b,c and Extended Data Fig. 8e). In addition, flow cytometry analysis revealed an increase in $Ca^{2+}$ levels following compound administration, which was diminished in *OR51E2*−/− cells (Fig. 7d–f). These results were replicated in THP1-derived macrophages). Notably, $Ca^{2+}$ levels increased upon exposure to palmitic acid, whereas they were reduced in the absence of OR51E2 (Extended Data Fig. 8f,g). Administration of ionomycin, which was utilized as a negative control, did not alter $Ca^{2+}$ levels in

**Fig. 5 | Pharmacological inhibition of olfactory receptor in a prostate cancer model. a**, FACS analysis of LGP (*n* = 3), OLFR644-KO (*n* = 3) and VMN2R29-KO (*n* = 3) macrophages exposed to 13-cRA for 4 h with and without *Pten*−/− *Trp53*−/− conditioned media. Events are plotted as ratio versus conditioned macrophages. **b,c**, Experimental scheme (**b**) and tumor growth (**c**): mice were injected intravenously with macrophages pretreated for 4 h with DMSO (*n* = 8) or 13-cRA (*n* = 9). Ctrl mice were untreated. **c**, Tumor growth expressed as a percentage of the initial volume. **d**, Representative FACS plot of GFP− and GFP+ tumor-infiltrating macrophages. **e,f**, FACS analysis: percentage of cells gated on GFP+F4/80+CD11b+ cells (Macs + DMSO or 13-cRA, *n* = 8) (**e**); percentages of CD39+ cells gated on CD8+ T cells and FoxP3+CD25+ $T_{reg}$ cells gated on CD4+ T cells (Untr., Macs + DMSO or 13-cRA, *n* = 9) (**f**). **g**, Response of splenocytes to mitomycin-C-killed tumor cells was examined ex vivo using tumor cell restimulation assays. IFNγ production in response to stimulation was assessed through ELISA or FACS analysis after a 72-h incubation period (Untr. *n* = 10, Macs + DMSO *n* = 8, Macs + 13-cRA *n* = 10). **h**, Tumor growth expressed as a percentage of the initial volume. Mice were injected intravenously with LGP-Macs or *Olfr644*−/− Macs pretreated for 4 h with DMSO or 13-cRA. **i,j**, FACS analysis: percentages of cells gated on GFP+F4/80+CD11b+ cells (LGP *n* = 6, LGP+13-cRA *n* = 6, OLFR644 *n* = 8, OLFR644 + 13-cRA *n* = 6) (**i**); and percentages of CD39+ cells gated on CD8+ T cells and $T_{reg}$ cells gated on CD4+ T cells (LGP *n* = 6, LGP + 13-cRA *n* = 6, OLFR644 *n* = 8, OLFR644 + 13-cRA *n* = 6) (**j**). Statistical analyses were performed using two-tailed unpaired Student's *t*-test. Values are presented as the mean ± s.e.m. All replicates represent biological replicates.

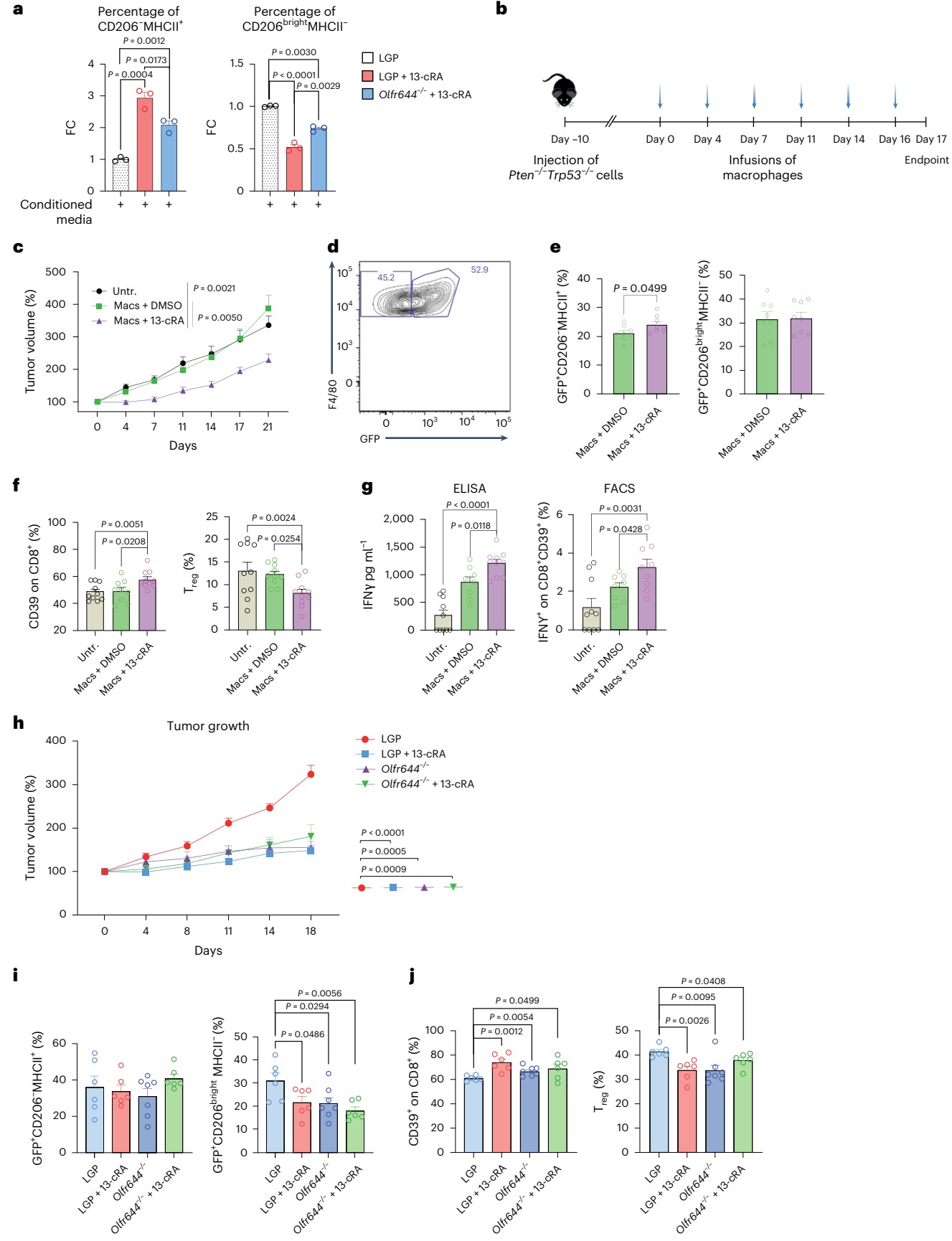

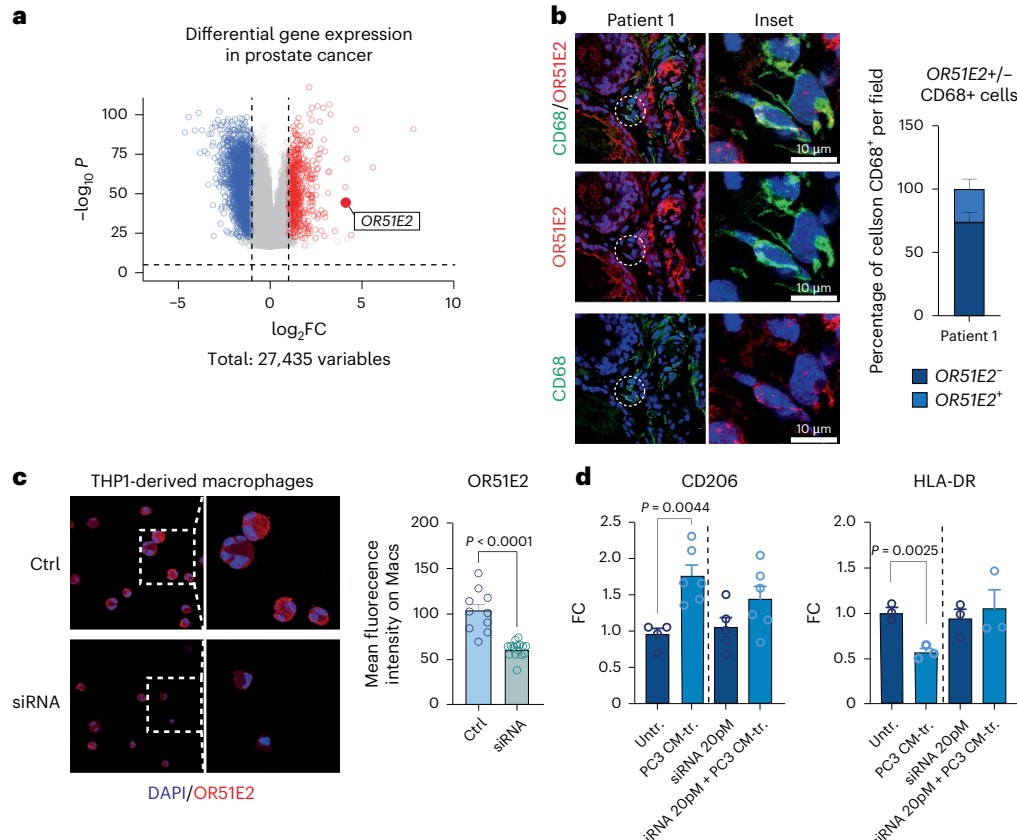

**Fig. 6 | Analysis of OR51E2 on human macrophages. a**, Volcano plot showing differentially expressed genes in prostate cancer tissues compared to normal tissue from The Cancer Genome Atlas. Genes are colored according to their $\log_2$FC value (blue, $\log_2$FC $\leq$ −0.5; red, $\log_2$FC $\geq$ 0.5). Data were analyzed using GEPIA2. **b**, Representative confocal immunofluorescence images and quantification of human prostate cancer tissues (patient 1) showing expression of OR51E2 (red) in CD68⁺ macrophages (green). Nuclei were counterstained with DAPI (blue). Images were acquired with an SP8-II confocal microscope (Leica). Scale bar, 10 μm; number of fields = 5. **c**, Representative confocal immunofluorescence images and quantification of OR52E2 expression in human

Thp1 cells with and without genetic deletion of *OR51E2*. Each dot represents one cell. $n \geq 10$. **d**, Bar graphs showing percentages of CD206⁺ and HLA-DR⁺ macrophages gated on CD68⁺ upon exposure to conditioned media from PC3 tumor cells, in the presence or absence of partial genetic deletion of *OR51E2*. Results are expressed as the FC of the CM-tr. over the Untr. group (CD206: Untr. $n = 4$, short interfering RNA (siRNA) $n = 5$, CM-tr. or siRNA + CM-tr. $n = 6$; HLA-DR: Untr. $n = 3$, siRNA $n = 3$, CM-tr. or siRNA + CM-tr. $n = 3$). Statistical analyses were performed using two-tailed unpaired Student's $t$-test. Values are presented as the mean ± s.e.m.

either THP1 cells or primary macrophages (Extended Data 8h,i). In addition, we performed immunofluorescence analysis to visualize lipid deposition using BODIP FL C16, a fluorescently labeled palmitic acid analog, in OR51E2-proficient and OR51E2-deficient primary macrophages. We observed reductions in palmitic acid deposition in OR51E2-deficient cells at 10 min and 1 h postexposure (Fig. 7g,h). Bulk mRNA-seq revealed that in macrophages, palmitic acid induced upregulation of biological pathways associated with wound healing and downregulation of inflammation-related pathways, such as antigen presentation and T cell activation (Extended Data Fig. 9a,b). Accordingly, it increased levels of CD204, CD206 and ARG1 in human tumor-conditioned macrophages, while decreasing HLA-DR levels. These changes were abolished in OR51E2-deficient macrophages (Fig. 7i). Similar results were obtained in THP1 cells (Extended Data Fig. 9c–f).

To explore the abundance of palmitic acid in human tumor sections, we employed a targeted mass-spectrometry-based spatial lipidomic approach to analyze prostate cancer biopsies. Spatial analysis revealed a higher abundance of palmitic acid in tumoral tissue compared to adjacent normal tissue (Fig. 8a,b). In addition, neoplastic glands exhibited extensive CD68⁺ macrophage infiltration, in close contact with palmitic acid (Fig. 8a, right panel and insets). To characterize the cellular composition and spatial organization

of prostate cancer biopsies, we developed a multiplexed antibody panel (Supplementary Table 3) and used imaging mass cytometry to acquire 18 high-dimensional histopathology images. These comprised images of nine advanced tumor areas and nine adjacent nontumoral tissues. Segmentation and cell type assignment using a pixel-level classification system[32] (Fig. 8c) identified epithelial cells, endothelial cells, fibroblasts and various immune cell subsets, including TAMs. Macrophages were further categorized into pro-inflammatory (CD206⁻MHCII⁺CD68⁺ M1-like) and tumor-promoting (CD206⁺MHCII⁻CD68⁺ M2-like) phenotypes based on the expression of MHC-II and CD206, respectively. Absolute counts of each cell type within individual images were quantified, enabling direct comparisons between tumoral and nontumoral regions. Our analysis revealed increased abundance of most cell types in tumor areas, including conventional T cells, $T_{reg}$ cells, and both CD206⁻MHCII⁺CD68⁺ M1-like and CD206⁺MHCII⁻CD68⁺ M2-like macrophages (Extended Data Fig. 9g). Notably, spatial interaction analysis demonstrated a greater propensity of Pan-CK⁺ epithelial cancer cells to interact with M2-like macrophages in palmitic-rich tumor regions compared to Pan-CK⁺ epithelial cells in the palmitic-deprived areas (Fig. 8d,e). Furthermore, we observed greater homotypic interactions between macrophages of different subtypes in nontumoral areas relative to tumor regions, suggesting reprogramming of TAM activation and altered interaction

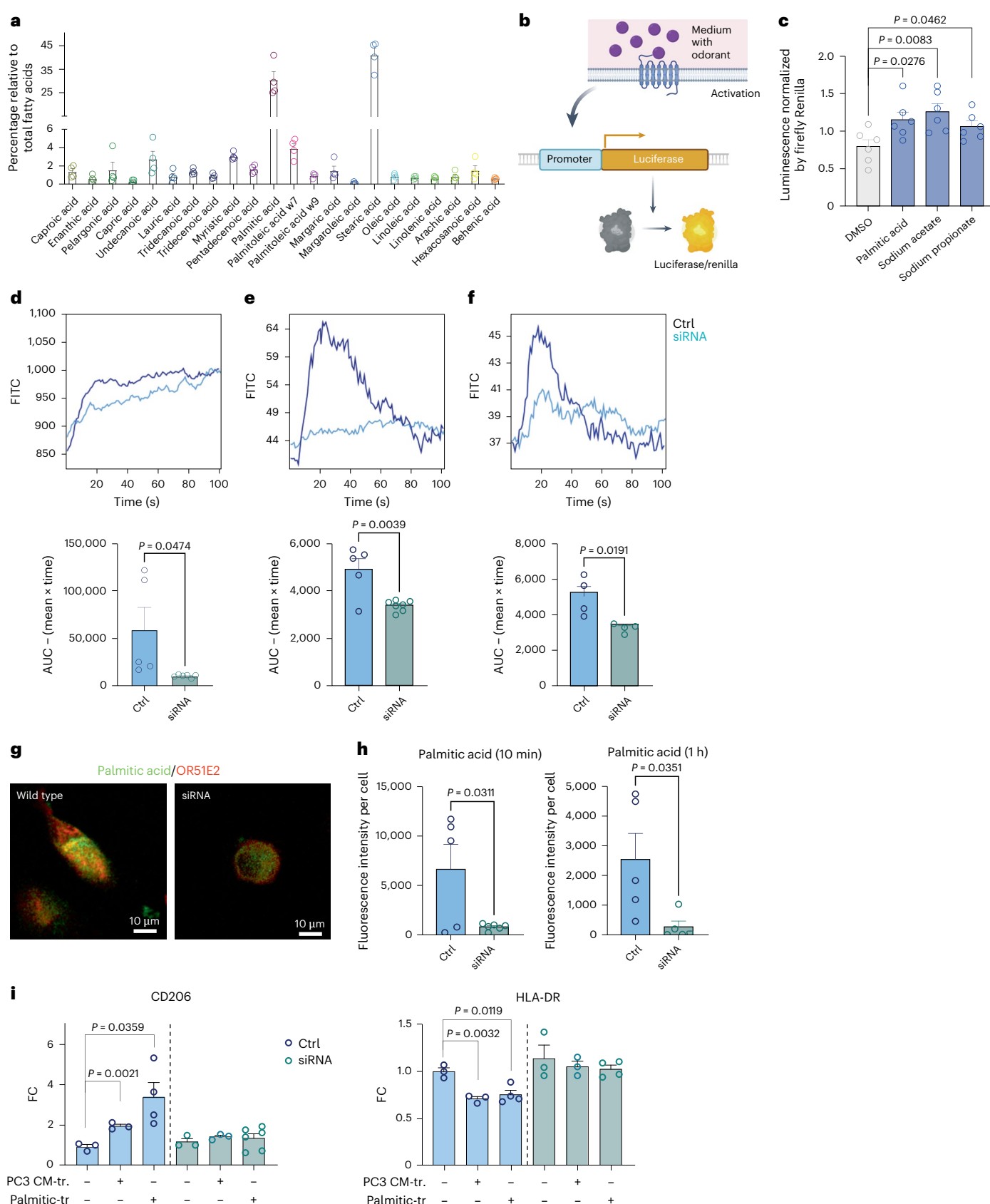

patterns in the tumor microenvironment (Fig. 8e). These results support the hypothesis that palmitic acid in the tumor microenvironment may shape macrophage activation toward a protumoral phenotype. Heterotypic interactions between immune cells were globally increased

in tumor regions enriched with palmitic acid, reflecting reorganization of the cellular niche within the cancerous tissues (Fig. 8d,e and Extended Data Fig. 9g). To explore the heterogeneity of TAMs in prostate tumors, we performed spatial transcriptomics using the

**Fig. 7 | Effect of palmitic acid on human macrophages. a**, Lipidomic analysis of conditioned media from PC3 prostate cancer cells. Results are expressed as the percentage of each fatty acid relative to the total fatty acids detected ($n = 4$). **b,c**, Luciferase reporter gene assay. Experimental scheme (**b**). Cells were transfected with 20 ng per well of plasmids encoding an olfactory receptor, 5 ng per well of RTP1S, 10 ng per well of CRE–luciferase and 5 ng per well of pRL-SV40. Twenty-four hours later, cells were stimulated by incubation with 50 μM palmitic acid, sodium acetate or sodium propionate. Four hours after stimulation, luminescence was measured. Bar plot showing all luminescence values divided by Renilla luciferase activity to control for transfection efficiency in a given well (**c**). Each comparison was performed in three technical triplicates ($n = 6$). **d–f**, Ca$^{2+}$ flux in primary macrophages in the absence or presence of partial genetic deletion of OR51E2-KO treated with palmitic acid (100 μM) (Ctrl $n = 5$,

siRNA $n = 6$) (**d**), acetate (Ctrl $n = 5$, siRNA $n = 7$) (**e**) or propionate (Ctrl $n = 4$, siRNA $n = 4$) (**f**). **g**, Immunofluorescence showing palmitic acid (green) and OR51E2 (red) on primary macrophages in the presence or absence of RNA KO of OR51E2. Palmitic acid was administered to the cells for 10 min or 1 h before quantification. **h**, Results are expressed as fluorescence intensity per cell (Ctrl $n = 5$, siRNA 10 min $n = 6$, siRNA 1 h $n = 5$). **i**, Expression of CD206 and HLA-DR by FACS on wild-type or OR52E1-KO THP1 cells. Cells were conditioned with either PC3-conditioned media or palmitic acid (100 μM) (CD206: Untr. or siRNA $n = 3$, CM-tr. or siRNA + CM-tr. $n = 3$; palmitic-tr. or siRNA + palmitic-tr. $n = 4$ or 6; HLA-DR: Untr. or siRNA $n = 3$, CM-tr. or siRNA + CM-tr. $n = 3$; palmitic-tr. or siRNA + palmitic-tr. $n = 4$). Statistical analyses were performed using two-tailed unpaired Student's $t$-test. Values are presented as the mean ± s.e.m. Schematic in **b** created using BioRender.com.

Visium Cell Assist assay on prostate cancer biopsies and integrated data with spatial lipidomics on the same samples. Unbiased clustering analysis identified four distinct tissue areas, of which two corresponded to palmitic-deprived nontumoral tissues and two corresponded to palmitic-enriched tumoral tissues (Fig. 8f,g and Extended Data Fig. 9i). Analysis of gene expression revealed enrichment of the lipid-laden TAM and angiogenic TAM signatures within palmitic-enriched regions (Fig. 8h–j and Extended Data Fig. 9j); this was indicative of a protumoral phenotype of TAMs in these areas and thus supported the role of palmitic acid in shaping macrophage function in tumors.

Overall, these results support the role of palmitic acid as an endogenous ligand for OR51E2 and demonstrate its role in polarizing human macrophages toward an immunosuppressive phenotype.

## Discussion

The abundance of macrophages at the tumor bed, as well as their plasticity, can be exploited against cancer. Among therapies targeting macrophages, inhibition of CSF1R has emerged as a particularly promising avenue[33]. However, in most solid tumors, CSF1R inhibitors as single agents or in combination with chemotherapy have demonstrated limited activity[1]. In this context, there is an unmet need for safe and effective anticancer therapies directed against TAMs. To gain a deeper understanding of the cross-talk between cancer and TAMs and identify new targets involved in this process, we conducted a comprehensive genome-wide unbiased CRISPR screening of primary macrophages exposed to the tumor microenvironment. This approach revealed a multitude of known drivers associated with both proinflammatory and protumoral phenotypes, providing positive validation of the screening methodology. We focused on negative regulators and validated the roles of the top-ranked genes in other hormone-dependent tumors, recapitulating the observations in prostate cancer and thereby expanding the potential of the genetic screening.

Importantly, the screening highlighted the involvement of olfactory and vomeronasal receptors in shaping macrophage behavior. Chemosensor receptors were first described in odorant tissue, localized in the cilia of olfactory sensory neurons[34–37]. Recent reports have

expanded our understanding of olfactory receptors beyond their traditional presence in olfactory tissues, revealing pleiotropic expression on immune cells and macrophages among other cell types[2,9,38]. Our data suggest that the function of macrophage-expressed chemosensors may be diverse and contingent on the specific chemosensor expressed and the type of ligand it recognizes. A compelling hypothesis is that macrophages perceive the multitude of the factors in the tumor milieu through the complexity of their chemosensor receptors. A thorough analysis of the chemosensor expression patterns in macrophages during cancer and inflammation could yield valuable insights. We initially explored chemosensory receptors in mice, in which these receptors are present in large numbers and differ from those in humans, with vomeronasal receptors functionally absent from human macrophages. Our findings suggest that other chemosensory receptors, such as OR51E2, may have a compensatory role in regulation of macrophage function. OR51E2 was among the most differentially expressed olfactory receptors in human prostate cancer, as demonstrated by analyses of data from The Cancer Genome Atlas. Its strong modulation in the human context led us to investigate its interaction with palmitic acid, which may have important implications for tumor biology and potential therapeutic strategies.

We identified common transcriptional signature in macrophages that were genetically deleted with respect to different chemosensors. This strongly suggests that activation of chemosensor receptors initiates a common signaling pathway, providing information to the mechanistic underpinnings of olfactory engagement in macrophages. This is of interest as, canonically, olfactory receptors in sensory organs couple with stimulatory G proteins, whereas vomeronasal receptors couple with Go subtypes. Further investigation is needed to clarify the downstream factors involved in the activity of chemosensor receptors in TAMs.

In the human setting, we demonstrated that OR51E2 is expressed on TAMs and confers protumoral functions on macrophages once engaged by palmitic acid. The association between lipids, particularly fatty acids, and their involvement in supporting tumor growth is well established; they not only serve as fuel for the rapid proliferation of cancer cells but also play a crucial part in modulating immune responses within the tumor microenvironment[39,40]. However, our

**Fig. 8 | Palmitic acid accumulates in tumor regions and modulates macrophage phenotype. a**, Representative images of patient biopsies analyzed by hematoxylin and eosin (H&E; T, neoplastic tissue; A, adjacent normal tissue); spatial analysis of the distribution of palmitic acid by mass spectrometry imaging; and mosaic immunofluorescence showing CD68 (green), pancytokeratin (Pan-CK, red) and nuclei (DAPI, blue). **b**, Quantification of palmitic acid in patient biopsies, comparing tumoral (Tum.) versus nontumoral (No tum.) areas within each patient. $n = 4$ patients. Two-tailed paired Student's $t$-test was used for the statistical analysis. **c**, H&E and spatial analysis of the distribution of palmitic acid by mass spectrometry imaging in patient number 2. **d**, Segmented images (inset) showing nontumor (left) and tumor (right) regions. **e**, Heat map showing significant pairwise cell–cell interaction (red) or avoidance

(blue) across the nontumor and tumor regions. **f**, UMAP identifying two nontumoral and two tumor clusters. **g**, Spatial distribution of the cluster shown in **f**. **h**, Spatial distribution of inflammatory TAM (Inflam-TAM), TAM defined by Cl3 interferon-related genes (IFN-TAM), lipid-laden TAM (LA-TAM), angiogenic TAM (Angio-TAM) and Cl6 regulatory TAM (Reg-TAM) signatures in patient number 2; image represents the enrichment of each signature and its spatial distribution. **i**, Distribution of each signature from **h** in the four identified areas: each dot represents a gene, and the size of the dot is representative of the expression level (exp.) of the gene. **j**, Bubble plot representing the expression of selected genes in the four identified areas. Unless otherwise specified, statistical analyses were performed using two-tailed unpaired Student's $t$-test. Values are presented as the mean ± s.e.m. Max., maximum; Min., minimum; T$_H$ cell, T helper cell.

findings introduce an important perspective by elucidating a potential new role for lipids, specifically palmitic acid, in shaping the behavior of macrophages. It will be of interest to investigate whether palmitic acid acts as a ligand of olfactory receptors or can be engulfed by macrophages following interaction with these receptors. The unexpected connections among olfactory receptors, palmitic acid and the protumoral functions of macrophages open avenues for further

exploration, prompting a broader evaluation of the interplay between lipids and the immune system in the context of cancer.

In conclusion, our research provides insights into the interaction between TAMs and cancer, offering potential targets to enhance antitumor responses. The role of olfactory receptors in tumor contexts holds substantial promise and warrants further exploration in future research.

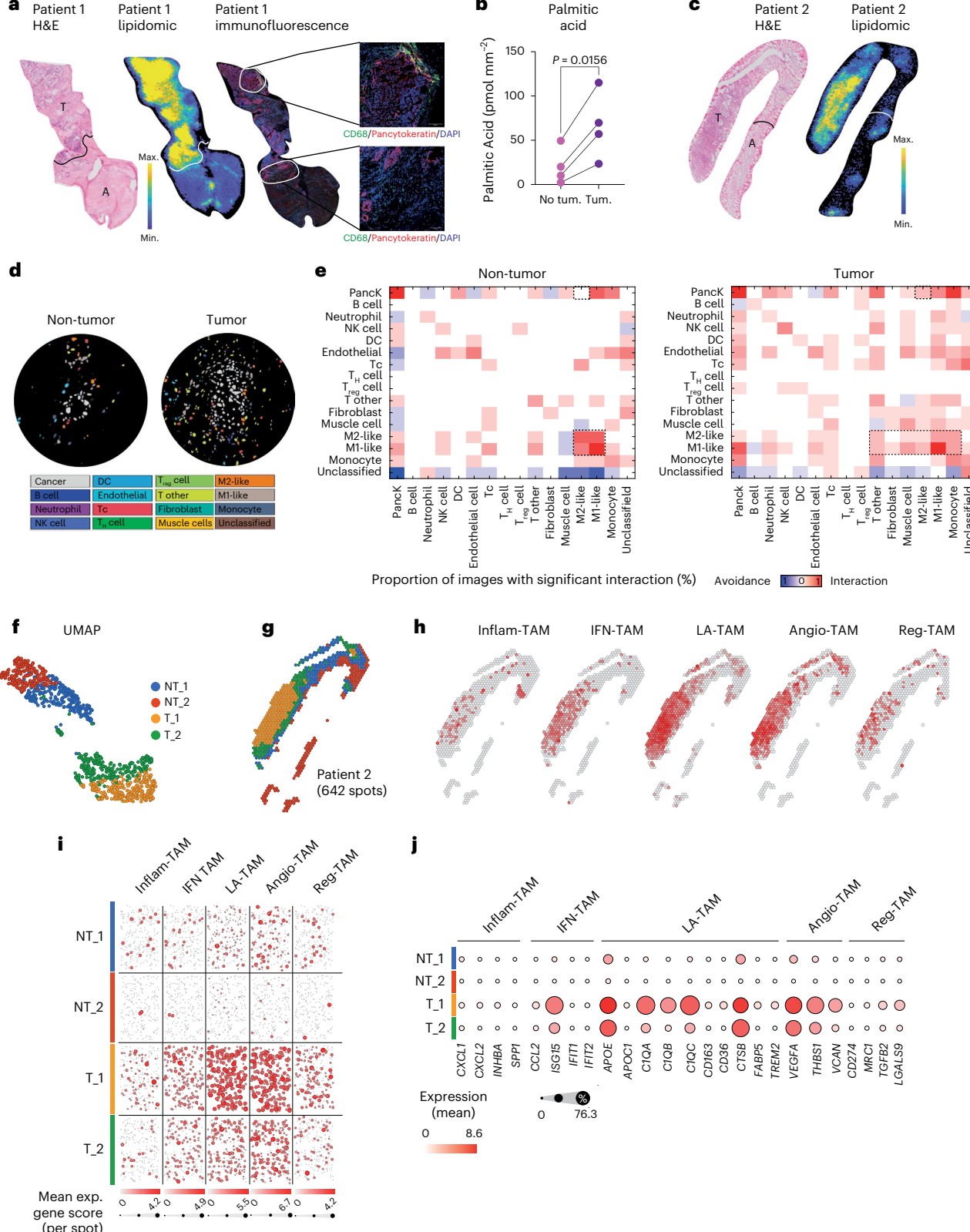

## Online content

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

[1]Department of Biomedical Sciences, Humanitas University, Milan, Italy. [2]IRCCS Humanitas Research Hospital, Tumor Microenvironment Unit, Milan, Italy. [3]IRCCS Humanitas Research Hospital, Translational Immunology Lab, Milan, Italy. [4]IRCCS Humanitas Research Hospital, Medical Genetics and RNA Biology Lab, Milan, Italy. [5]Human Technopole, Milan, Italy. [6]Institute of Genetics and Biomedical Research, UoS of Milan, National Research Council, Milan, Italy. [7]IRCCS Humanitas Research Hospital, Genomics Unit, Milan, Italy. [8]IRCCS Humanitas Research Hospital, Flow Cytometry Core, Milan, Italy. [9]IRCCS Humanitas Research Hospital, Experimental Immunopathology Unit, Milan, Italy. [10]Institute of Oncology Research, Faculty of Biomedical Sciences, USI, Bellinzona, Switzerland. [11]Rosalind and Morris Goodman Cancer Institute, Department of Human Genetics, McGill University, Montreal, Quebec, Canada. [12]IRCCS Humanitas Research Hospital, Department of Pathology, Milan, Italy. [13]IRCCS Humanitas Research Hospital, Urology Unit, Milan, Italy. [14]IRCCS Humanitas Research Hospital, Metabolomic Unit, Milan, Italy. [15]Institute of Neuroscience, National Research Council of Italy (CNR) c/o Humanitas Mirasole S.p.A, Rozzano, Italy. [16]IRCCS Humanitas Research Hospital, Cellular and Molecular Oncoimmunology Unit, Milan, Italy. [17]Department of Molecular Genetics and Microbiology, Duke University School of Medicine, Durham, NC, USA. [18]Oncology Institute of Southern Switzerland, Ente Ospedaliero Cantonale, Bellinzona, Switzerland. ✉e-mail: diletta.di_mitri@humanitasresearch.it

## Methods

### Cell culture and treatments

The *Pten*[−/−] *Trp53*[−/−] cell line was provided by R. A. DePinho. *Pten*[−/−] *Trp53*[−/−], PC3 and HEK293T cells were grown in DMEM with high glucose (4,500 mg l[−1]; Sigma-Aldrich) supplemented with 10% heat-inactivated fetal bovine serum (FBS; Sigma-Aldrich), penicillin–streptomycin solution (penicillin G 10 U ml[−1] + streptomycin 0.1 mg ml[−1]; Euroclone), 2 mM L-glutamine (Euroclone) and 1 mM sodium pyruvate solution (Euroclone) at 37 °C in 5% $CO_2$. To make conditioned media, $7 \times 10^5$ cells were seeded in a T75 flask, and media were collected after 72 h. The L929 murine fibroblast cell line was cultured with the described complete media with North American FBS, and supernatant was collected after 4 days. Media for macrophage differentiation was produced with the following proportions: 30% L929 supernatant, 20% NA FBS, 1% penicillin–streptomycin solution, 2 mM L-glutamine (1%), 0.5 mM sodium pyruvate (0.5%), 10 µM β-mercaptoethanol (0.1%; Gibco). BMDMs were differentiated in vitro. Briefly, bone marrow precursors were flushed from long bones of C57BL/6 male mice or Rosa26-Cas9 male mice and cultured in complete media with L929 supernatant for 7 days. When conditioned, on day 7, media were replaced by conditioned media diluted 1:1 in complete L929 media for 48 h before sample processing. The media were then replaced and macrophage supernatant collected after a further 48 h. THP1 cells were grown in RPMI (4,500 mg l[−1]; Sigma-Aldrich) supplemented with 10% heat-inactivated FBS (Sigma-Aldrich), penicillin–streptomycin solution (penicillin G 10 U ml[−1] + streptomycin 0.1 mg ml[−1]; Euroclone), 2 mM L-glutamine (Euroclone) and 1 mM sodium pyruvate solution (Euroclone) at 37 °C in 5% $CO_2$. Macrophage-derived THP1 cells were obtained by administering phorbol 12-myristate 13-acetate (30 ng ml[−1]) for 24 h. Media were changed and cells used for experiments after a further 24 h.

### CRISPR knockdown

Viral particles were obtained by transfecting HEK293T cells with sgRNA plasmids VSVG and PAX using $CaCl_2$ overnight. Media were then replaced with macrophage-suitable culture media. After 24 h, virus-containing supernatant was filtered with a 0.45-µm filter and given to Cas9-expressing macrophages. After a second round of infection the following day, macrophages were exposed to puromycin selection (4 µg ml[−1]) for 72 h. At this step RNA and protein lysates were obtained. When conditioned, macrophages were exposed to conditioned media for 48 h before harvesting.

### sgRNA cloning and virus production

sgRNAs were either purchased as part of the lentiGuide-Puro vector from GeneScript (U142SEJ040_3) or manually designed using CRISPRscan.org (Giraldez laboratory, Yale University). Designed sequences were purchased from Sigma-Aldrich and cloned into the lentiGuide-Puro vector following the protocol of the Zhang laboratory. All the sequences used are reported in the 'Oligonucleotides tables' in the Supplementary Information. The vector was digested with BsmBI (Fermentas) for 30 min at 37 °C. The digested plasmid (~11 kb) was gel purified, and the oligo pair was annealed. Then, a ligation reaction was carried out with Quick Ligase (New England Biolabs). The obtained sgRNA–lentiGuide-Puro vector was transformed in Stbl3 bacteria, and the amplified plasmid was purified with Qiagen Midi Prep Kit. Viral particles were obtained by transfecting HEK293T cells with the sgRNA plasmid, VSVG and PAX using $CaCl_2$ overnight. Media were then replaced with macrophage-suitable culture media. After 24 h, virus-containing supernatant was filtered with a 0.45-µm filter and given to Cas9-expressing macrophages. After a second round of infection the following day, macrophages were exposed to puromycin selection (4 µg ml[−1]) for 72 h.

### GeCKO v2 Library B

To perform the genome-scale KO screening, we followed the protocol provided by the Zhang laboratory. The mouse GeCKO v2

Library B (Addgene) was used: this library consists of 62,804 sgRNAs constructs, with three sgRNAs targeting each of the 20,661 genes of the mouse genome. These constructs are included in lentiGuide-Puro plasmid 35. The library was introduced in Lucigen Endura competent bacterial cells (no. 60242) through electroporation. The transformation efficiency was determined, and transformed cells were plated with a spreader on prewarmed agar plates containing ampicillin for cell selection. After overnight growth, scraped colonies were purified for plasmid DNA with a Plasmid Maxi prep kit (Qiagen). To determine sgRNA distribution, the pooled sgRNA library was amplified by PCR by using Kapa HiFi high-fidelity polymerase (Kapa Biosystems) and the primer mix indicated by the Zhang protocol. The amplified library was purified with a QIAquick PCR purification kit (Qiagen) and run on a 2% agarose gel, and the product was extracted with a QIAquick Gel extraction kit (Qiagen) before being quantified and subjected to NGS. An admissible percentage of ~70% perfectly matching guides was detected. To include the library in a viral vector and produce the virus, we used HEK293T cells. The pMD2.G (VSVG) and psPAX plasmids (Addgene) needed to enable viral replication were obtained by transforming Stbl3 competent bacterial cells (Invitrogen) through heat shock and purifying the product with a Qiagen Midi or Maxi prep kit. HEK293T cells were transfected with the library and viral components with Lipofectamine 3000 (Invitrogen) following the relevant guidelines. The lentivirus produced was harvested 48 h after transfection and titrated. A multiplicity of infection of 0.7 was used. Primary macrophages expressing Cas9 (from Rosa26-Cas9 knock-in mice) were infected with the library and selected with puromycin (Sigma-Aldrich) for 72 h. Uninfected macrophages were used to check effective cell death. After selection, cells were detached, stained and sorted for MHCII[bright]CD206[−] and CD206[bright]MHCII[−] populations with a FACSAria III. The genomic DNA from each population was harvested with a Zymo Quick-gDNA kit (Zymo Research), and PCR was performed to amplify the sgRNA for NGS. The experiment was repeated twice. Statistical analyses and comparisons of the NGS output were performed with MAGeCK.

### T cell proliferation assay

Spleens were taken from C57BL/6 mice and dissociated through a 40-µm strainer with the end of a syringe. After centrifugation for 5 min at 400*g*, the supernatant was removed, and 1 ml of ACK was added for 1 min. ACK was then diluted with complete DMEM, a further centrifugation step was performed, and cells were resuspended in 10 ml of media. Cells were counted to enable us to seed $2 \times 10^5$ cells per well. The cells needed were centrifuged, resuspended in Cell Trace solution (Thermo Fisher Scientific) and incubated at 37 °C for 20 min. After incubation, cells were centrifuged, washed with 1 ml of media, then centrifuged again, pooled with the Dynabeads and seeded in a 96-well round-bottomed plate. Then, 150 µl of specific media derived from macrophages were added to the wells. After 72 h, T cells were activated with 50 ng ml[−1] phorbol 12-myristate 13-acetate and 1 µg ml[−1] ionomycin for 6 h. After 90 min, protein transport inhibitor brefeldin (1,000×; BioLegend) was added. Cells were then separated from the beads with a magnet (DynaMag, Invitrogen) and stained for FACS analysis.

### Wound healing assay

*Pten*[−/−] *Trp53*[−/−] cells ($3 \times 10^5$) were seeded in each well of a 12-well plate. After 24 h, culture media were removed, and a scratch was made with a p200 tip following the central axis of each well. Wells were gently washed to remove debris, and different media for each condition were administered to cells. The plate was then positioned in the acquisition chamber of a DMi8 microscope for live imaging experiments. Three images from different spots of the same well were acquired every 30 min for 6 h.

### Immunofluorescence

Frozen tissue sections (8 µm) were first rehydrated and then permeabilized and blocked with 0.1% Triton X-100 (Sigma-Aldrich), 2% bovine

serum albumin (BSA), and 5% normal goat serum in PBS with calcium and magnesium (PBS+/+) for 30 min in a dark incubation chamber. Sections were washed for 5 min in PBS+/+ with Tween-20 0.05% (washing buffer) and subsequently incubated with primary antibodies in washing buffer for 1 h at room temperature. After three washes in washing buffer, sections were incubated with fluorophore-conjugated secondary antibodies (1:1,000) at room temperature for 1 h in the dark. Finally, nuclei were counterstained with DAPI. Images were captured with a SP8-II confocal microscope (Leica). The acquired images were analyzed with Fiji (ImageJ) software.

## Western immunoblotting

Cells were detached and lysed with RIPA buffer with addition of inhibitors of proteases and phosphatases (Thermo Fisher Scientific). Protein lysates were collected, quantified with a DC Protein Kit (Bio-Rad), boiled at 95 °C for 5 min and prepared for gel electrophoresis. Sodium dodecyl sulfate polyacrylamide gel electrophoresis was performed by loading 30 µg of proteins on 10% bis-acrylamide gels, in parallel with Precision Plus Protein Dual Color Standards (Bio-Rad) for molecular weight estimation. Proteins were transferred onto nitrocellulose or 0.45 µm polyvinylidene fluoride membranes, then blocked with BSA (5% in TBS-Tween 0.1%) and incubated with primary antibodies. Antibodies for pSTAT6 (1:1,000; Cell Signaling), STAT6 (1:1,000; Cell Signaling) and β-actin (1:1,000; Abcam) were used. After washing with TBS-Tween, membranes were incubated with HRP-conjugated anti-rabbit secondary antibody (1:1,000; R&D). Membranes were washed again, then incubated with ECL solution (Bio-Rad). Signal acquisition was performed with ImageLab software.

## qPCR

Total RNA was extracted using RLT lysis buffer or TRIzol reagent (Thermo Fisher Scientific), following the manufacturer's recommendations. RNA was further purified using an RNeasy Mini RNA isolation kit or the TRIzol protocol for RNA isolation. Complementary DNA (cDNA) was synthesized using 1 µg of total RNA by reverse transcription using a High Capacity cDNA Reverse Transcription Kit. Quantitative real-time PCR was performed using SYBR Green PCR Master Mix using a QuantStudio 7 Flex Real-Time PCR System. All the primers used are reported in the 'Oligonucleotides tables' in the Supplementary Information.

## RNA sequencing and bioinformatic analysis

All samples were sequenced on an Illumina NextSeq500 platform, generating an average of 15 million 75-bp single-end reads per sample. After sequencing, quality control checks were performed to assess the data quality and remove low-quality reads or artifacts. The quality-filtered reads were aligned to the human genome (GRCm38) using the STAR aligner with default parameters (v.2.6.1d). The aligned reads were then used to obtain gene-based read counts using the featureCounts module (v.1.6.4) and Ensembl GRCm38 annotation. Raw read counts were normalized using the trimmed mean of log-ratio values method. Genes with counts per million mapped reads greater than 1 in at least two libraries were considered for further analysis. The edgeR package (v.3.26.5) in the R statistical software was used to perform differential gene expression analysis. $P$ values were adjusted using the Benjamini–Hochberg method, and genes were considered to be differentially expressed when the false discovery rate was less than 0.05 and the expression change greater than 1 $\log_2$FC. A volcano plot was generated using the EnhancedVolcano R package (v.1.18). For gene signature identification (RNA sequencing), GSEA was performed using GSEA software (v.3.0) from the Broad Institute of MIT. The gene list was ranked based on $\log_2$FC. GSEA was conducted in preranked mode with the scoring scheme set to 'classic' and using 1,000 permutations. The gene signature used for analysis was retrieved from the Molecular Signatures Database (v.6.2).

## scRNA-seq

Prostates from mice were processed as previously described. CD45+ cells were sorted by flow cytometry. Afterward, CD45+ cells from each sample were loaded into one channel of a Single Cell Chip A using a Single Cell 3′ v2 Reagent Kit (10x Genomics) for generation of gel bead emulsion into the Chromium system. Following capture and lysis, cDNA was synthesized and amplified for 14 cycles according to the manufacturer's protocol (10x Genomics). Then, 50 ng of the amplified cDNA was used for each sample to construct Illumina sequencing libraries. Sequencing was performed on the Illumina NextSeq 500 sequencing platform following the 10x Genomics instructions for read generation. A sequencing depth of at least 20,000 reads per cell was obtained for each sample.

## Data processing

Raw BCL files were analyzed using Cell Ranger (v.7.2) with default settings. The files were demultiplexed and converted to FASTQ format, followed by alignment using Cell Ranger (v.7.2), which employs the STAR aligner, to generate the gene expression matrix. Reads were aligned to the mm10 (mouse) reference genome (2024-A version, 10x Genomics). Confidently mapped reads with valid barcodes and unique molecular identifiers (UMIs) were selected, resulting in a gene expression matrix containing UMI counts for each gene across individual cells. These gene count matrices were imported into the R environment (v.4.4.2) and analyzed with Seurat (v.3). Genes expressed in fewer than three cells were removed, and cells with less than 1,000 UMI counts or fewer than 200 genes (for mouse samples) were excluded. In addition, cells exhibiting a mitochondrial-to-endogenous gene expression ratio greater than 0.5 (for mouse samples) were filtered out. The raw expression matrix was normalized with a $\log_2$ transformation via the NormalizeData function. The data were scaled using ScaleData while regressing out mitochondrial gene percentages and cell cycle effects; the latter were computed using the CellCycleScoring function. Highly variable genes (top 2,000) were identified using the FindVariableFeatures function with the vst method. Principal component analysis was performed with the RunPCA function, using default parameters.

## Graph-based clustering, differential gene expression analysis and trajectory analysis

Graph-based clustering and differential gene expression analysis were performed as previously reported. Macrophage-related clusters were isolated from $Pten^-Trp53^-$ prostate mouse model datasets using Seurat. Clusters 0, 2, 4, 5, 6, 9 and 12 were selected for further analysis. To ensure consistency, the active assay was reset to 'RNA', and slots such as scale.data, meta.features and var.features were cleared. Data normalization was performed using the NormalizeData function, followed by variable feature identification and dimensionality reduction based on principal component analysis using the top 15 components. Uniform manifold approximation and projection embeddings were generated, and clustering was performed at a resolution of 0.5. The subclustered macrophage data were converted into a cell_data_set object for trajectory analysis with Monocle3, as previously described.

## Visualization of TAM subtype signatures

Signatures for seven TAM subtypes (proliferating TAMs, Cl1 resident-tissue macrophages, inflammatory TAMs, Cl2 angiogenic TAMs, Cl6 regulatory TAMs, lipid-laden TAMs, and TAMs defined by Cl3 interferon-related genes) were compiled from the literature (Table 1 in ref. [41]). Human gene symbols were converted into mouse orthologs using the BiomaRt R package. Gene rankings for each cell were computed with the AUCell_buildRankings function using count data extracted from the Seurat object with the GetAssayData function. Area under the curve (AUC) scores for each TAM signature were calculated using the AUCell_calcAUC function. Thresholds for identifying signature-positive cells were explored and defined using the

AUCell_explore Thresholds function. Cells exceeding these thresholds were labeled as 'signature positive', and this annotation was added as a metadata column to the Seurat object. Uniform manifold approximation and projection visualizations were generated with the DimPlot function, displaying cells by their signature status. Each TAM subtype was highlighted using distinct color schemes. This analysis provided a detailed visualization of TAM subtype signatures within macrophage subclusters from the *Pten⁻Trp53⁻* prostate mouse model, offering insights into their diversity and functional roles within the tumor microenvironment.

## Mice

All mice were maintained under specific-pathogen-free conditions at the Humanitas Clinical and Research Institute, and experiments were performed according to national guidelines approved by the Italian Health Ministry. Procedures involving animal handling and care conformed to protocols approved by the Humanitas Clinical and Research Center, in compliance with national and international law and policies. C57BL/6 mice were provided by Charles River. The maximal tumor diameter permitted by our ethics committee is 1 cm. This maximal tumor size was not exceeded. Mice were fed a standard chow diet and were randomized to the treatment groups.

## In vivo experiments

For the orthotopic experiment, 9-week-old male mice underwent surgery to inject *Pten⁻/⁻ Trp53⁻/⁻* cells, and, when specified, LGP, *Olfr644⁻/⁻* or *Vmn2r29⁻/⁻* macrophages were coinjected into one of the anterior lobes of the prostate. *Pten^{pc−/−} Trp53^{pc−/−}* mice were obtained by crossing *Pten^{lx/lx}Trp53^{lx/lx}* mice to probasin–Cre under the control of the androgen-responsive probasin promoter. Mice were sacrificed at 16 weeks old. For the allograft experiment, $3 \times 10^6$ *Pten⁻/⁻ Trp53⁻/⁻* epithelial cells were injected subcutaneously in the flanks of male 8-week-old mice. Ten days after injection, mice were randomized to the treatment groups. Tumor growth was monitored every other day by measuring tumor size with a caliper, and the volume was calculated using the following formula: tumor volume $= \pi W^2 L/6$, where $w$ is the tumor width and $l$ is the length. For intravein injection, bone marrow was used as a source for macrophages as described above. Macrophages were subjected to knockout of the selected genes using CRISPR. Alternatively, 13-cRA (50 μM) was used to treat macrophages 4 h before injection. Macrophages ($2 \times 10^6$) were infused twice a week for 3 weeks for a total of six injections. For CD8 depletion, mice were intraperitoneally treated with 200 mg of specific monoclonal antibodies (rat anti-CD8a, clone YTS 169.4; rat isotype control, clone LTF-2) 7 days after inoculation with *Pten⁻/⁻ Trp53⁻/⁻* cells and with 100 mg twice a week for the entire duration of the experiment. No statistical methods were used to predetermine sample size, but our sample sizes were similar to those reported in previous publications[42]. No data points were excluded from the analyses.

## Tumor infiltrate analysis by FACS

For analysis of tumor-infiltrating leukocytes, tumors were collected, cut into small pieces and digested with collagenase I (1 mg ml⁻¹ for mouse tissue and 0.5 mg ml⁻¹ for human tissue) for 45 min at 37 °C on a rocking platform. After a quick digestion in 2.5% trypsin and DNase I, single-cell suspensions were obtained by mechanical dissociation through a syringe needle (18G) and subsequent filtration on a 40-μm cell strainer. The composition of tumor infiltrate was determined by flow cytometry. Samples were analyzed with a FACSymphony A5 Cell Analyzer.

## Flow cytometry

Primary macrophages were detached from plates with accutase solution (Thermo Fisher Scientific), and nonspecific antibody binding was prevented by incubating cells with an Fc block (TruStain FcX anti-CD16/32, clone 93). Cells were then stained with LIVE/DEAD Fixable Viability Dye eFluor 780; BioLegend) for 20 min at 4 °C, followed by staining with

the following antibody mix: F4/80-BV421, CD206-APC, MHC-II-BV40, Ly6G-BUV786, CD11b-PECF594, CD115-BV711 and CD39-PeCy7 (BD Biosciences and BioLegend) for 30 min at room temperature. For ARG1 detection, after extracellular staining, samples were fixed and permeabilized (Intracellular Fixation & Permeabilization Buffer Set; eBioscience) and stained with ARG1-AF700 antibody. Each antibody had previously been titrated to identify the optimal working dilution. Cells were then fixed in 1% paraformaldehyde and acquired using a BD FACSymphony system. For T cells, extracellular staining was performed with CD3-BV650, CD8-BUV650 and CD4-BUV496, whereas intracellular staining used IFNγ-APC. The composition of tumor infiltrate was determined by flow cytometry. Samples were acquired using a BD FACSymphony A5 Cell Analyzer, and data were analyzed using FlowJo software.

## Ex vivo splenocyte restimulation assays

Harvested spleens were flushed through 70-μm BD Falcon cell strainers with complete T cell media (RPMI, 4,500 mg l⁻¹; Sigma-Aldrich) supplemented with 10% heat-inactivated FBS (Sigma-Aldrich), penicillin–streptomycin solution (penicillin G 10 U ml⁻¹ + streptomycin 0.1 mg ml⁻¹; Euroclone), 2 mM L-glutamine (Euroclone) and 1 mM sodium pyruvate solution (Euroclone) at 37 °C in 5% $CO_2$ and 0.1% β-mercaptoethanol). Red blood cells were lysed using RBC lysis buffer (Sigma-Aldrich) and resuspended in complete T cell media. Splenocytes were resuspended in T cell media to a final concentration of $5 \times 10^6$ cells ml⁻¹. *Pten⁻/⁻ Trp53⁻/⁻* cells were growth-arrested using mitomycin C (Roche) at a final concentration of 100 mg ml⁻¹ in a humidified incubator at 37 °C with 5% $CO_2$ for 2 h. Cells were washed twice with PBS and resuspended in T cell media at a final concentration of $5 \times 10^5$ cells ml⁻¹. Then, 100-μl aliquots of splenocyte suspensions were cocultured with 100 μl of mitomycin-C-treated cells in 96-well plates. Plates were incubated at 37 °C with 5% $CO_2$ for 3 days; then, the suspensions were centrifuged at 350g for 5 min, and the supernatants were collected. The concentration of IFNγ was determined using a murine-specific IFNγ ELISA kit (R&D Systems).

## Short interfering RNA

THP1 cells were differentiated into macrophages as described above. Then, $1 \times 10^5$ cells were seeded in a six-well plate and transfected with 50 pM short interfering RNA for OR51E2 using Lipofectamine 3000 (Thermo Fisher) for 48 h. Cells were then analyzed by qPCR or immunofluorescence.

## Intracellular calcium influx

Intracellular calcium influx was measured with Fluo-4, NW (Thermo Fisher). Briefly, 150,000 cells were plated on a 96-well plate and loaded with 2 μM Fluo-4. Cells were incubated in a humidified atmosphere of 5% $CO_2$ and 95% air at 37 °C for 30 min and then stored at room temperature for a further 30 min. Fluorescence was acquired using a BD FACSymphony system. Samples were acquired for 30 s to set the background. Then, a specific stimulus (B-ionone, 100 μM; palmitic acid, 100 μM; concentrate conditioned media, ionomycin, 2 μM) was added, and cells were acquired for a further 120 s. Results were analyzed using the 'kinetics' function in FlowJo. Then, the following formulas were used: AUC = AUC − (mean × 120); peak = peak − mean.

## Human samples

Male patients affected by prostate cancer were enrolled at Humanitas Clinical and Research Centre, Rozzano, Milan, Italy. All patients provided written informed consent. The protocol was approved by the Ethical Committee of Humanitas Clinical and Research Hospital. Patients did not receive compensation.

## Luciferase reporter gene assay

The Dual-Glo Luciferase Assay System (Promega) was used to measure receptor responses. Hana3A cells were plated on 96-well plates.

Twenty-four hours after plating, cells were transfected with 20 ng per well of plasmids encoding an olfactory receptor, 5 ng per well of RTP1S, 10 ng per well of CRE–luciferase and 5 ng per well of pRL-SV40. Furthermore, 24 h later, cells were stimulated by incubation with 50 µM compound (DMSO, palmitic acid, sodium acetate or sodium propionate) diluted in Minimum Essential Medium Eagle (Sigma-Aldrich) at 37 °C and 5% $CO_2$ to allow for CRE–luciferase expression. Four hours after stimulation, luminescence was measured using a Cytation 5 microplate reader. All luminescence values were divided by Renilla luciferase activity to control for transfection efficiency in a given well. Each comparison was performed in three technical triplicates.

### Lipidomic analysis

The lipidomic analysis was performed by Theoreo Srl. To obtain the fatty acid profile, lipids were converted into their corresponding methyl esters. One milliliter of the sample was mixed with 19 ml of extraction and transesterification solution composed of 17 ml of methanol, 1 ml of acetyl chloride and 1 ml of internal standard solution (containing 10 µg of 23:0 methyl ester). The tubes were capped and heated at 100 °C for 60 min. The tubes were then allowed to cool to room temperature. Hexane (7.5 ml) was added, and the tubes were vortexed for 30 s at 350$g$. The upper organic phase was collected with a glass Pasteur pipette. The combined hexane solution was evaporated to dryness under nitrogen, and the residue was then redissolved in 100 µl of hexane, transferred to capped gas chromatography vials and flushed with nitrogen. Two microliters of this solution were injected into the GC-MS-2010SE system, a gas chromatograph coupled to a single quadrupole mass spectrometer (Shimadzu Corp.) to obtain the fatty acid profile. Chromatographic separation was achieved with a 30-m fused silica Zebron ZB-Wax capillary gas chromatography column with an internal diameter of 0.25 mm and a film thickness of 0.25 µm, manufactured by Phenomenex, using helium as the carrier gas. The initial oven temperature of 80 °C was maintained for 2 min and then increased at a rate of 5 °C min$^{-1}$ to 170 °C, then at 2 °C min$^{-1}$ to 200 °C and at 20 °C min$^{-1}$ to the final temperature of 230 °C, with an additional 6.5 min hold time. The gas flow rate was set to achieve a constant linear velocity of 40 cm s$^{-1}$.

### Mass spectrometry imaging

For mass spectrometry imaging analysis of palmitic acid distribution, frozen tissues were cut in 10-µm-thick sections using a cryomicrotome (Leica Microsystems) at −20 °C and mounted on glass slides coated with indium tin oxide by standard thaw-mounting techniques. The slides were dried in a vacuum drier at room temperature for at least 3 h. Deuterated palmic acid at different concentrations was spotted on a supplementary tissue slice on each slide to build a calibration curve for quantification of palmitic acid in tissue sections. Each indium tin oxide slide was then sprayed with 1,5-diaminonaphthalene dissolved at a concentration of 10 mg ml$^{-1}$ in 70% $CH_3CN$ 70% using a SunCollect MALDI Sprayer (SunChrom Wissenschaftliche Geräte GmbH) with nitrogen at 2.5 bar and the following parameters: $z$ axis, 25 mm; number of layers, 10; spray speed, 600 mm min$^{-1}$; line distance, 2 mm; variable flow rate, 10, 20, 30, 40 or 6 × 60 µl min$^{-1}$. An Orbitrap Exploris 120 mass spectrometer (Thermo Fisher Scientific) coupled with an AP-MALDI-ng-UHR ion source (MassTech Inc.) was used to control the source with Target-ng software (MassTech Inc.). A laser energy of 3.5%, 3 kV voltage was applied to the plate, and a capillary temperature of 300 °C and 70% RF lens were used. Constant speed raster motion was used, with 30 µm spatial resolution and plate velocity dependent on scan time. Acquisitions were performed in full scan mode with negative polarity, $m/z$ 200–300 mass range, and resolution of 120K. Automatic gain control 1 was set to 100% with 100 ms maximum injection time. Data were exported and converted to imzML file format using the MT imzML Converter (ng) Installer Package (v.1.4.1) and imported into MSiReader v.1.00. The palmitic acid ion signal (tolerance, 2.5 ppm) was normalized in each pixel to the total ion chromatogram signal. For standard curve generation and quantitation, the $m/z$ intensity data for each region of interest drawn on each calibration spot were exported to Excel using the MSiReader ROI tool.

### Hyperion

Frozen sections were washed in PBS+/+ for 10 min and then fixed with 4% paraformaldehyde for 5 min at room temperature in a dark chamber. Subsequently, sections were incubated with the antibody mix, diluted in PBS+/+ with 2% BSA, 5% normal mouse serum (Biosera), normal rat serum (Sigma-Aldrich), rabbit normal serum (Dako) or goat normal serum (Sigma-Aldrich) serum, and 0.3% Triton X (Sigma) overnight at 4 °C. After incubation, sections were washed four times, for 5 min each time, in PBS+/+ with 0.05% Tween-20 (Merck). For nuclear staining, tissues were then incubated with 0.6 µM Ir191/193 (Standard BioTools) in PBS for 30 min at room temperature. After incubation, tissue sections were washed three times, for 3 min each time, in PBS+/+ with 0.05% Tween-20. Finally, sections were washed in ultrapure $H_2O$ to remove leftover salt and air dried. Images were acquired with a Hyperion Imaging System (Standard BioTools), according to the manufacturer's instructions. All cells contained within the imaging mass cytometry images were segmented as previously described[32]. To address phenotyping challenges in highly multiplexed imaging, we used a previously described supervised hierarchical pipeline for cell classification (REF Walsh Nature). This approach integrates canonical lineage markers, staining quality, population abundance and cell maturation. $k$-means clustering and generalized Gaussian models were used to segment multilevel image stacks based on staining intensity, identifying marker presence at specific locations. Each marker was evaluated across six levels, with final masks manually curated for accuracy. Finally, we performed a permutation-test-based analysis of spatial single-cell interactions to identify significant pairwise interaction or avoidance between cells. Interacting cells were defined as those within six pixels.

### Visium CytAssist Spatial Gene Expression for Fresh Frozen

Fresh frozen tumor samples were prepared according to the manufacturer's instructions (10x Genomics, tissue preparation guide, CG000636). Methanol fixation, hematoxylin and eosin staining, imaging and destaining of fresh frozen tissue were performed according to the protocol (10x Genomics, CG000614). Bright-field histological images were acquired using an Axio Scan.Z1 (ZEISS). Libraries were prepared using Visium CytAssist Spatial Gene Expression Reagent Kits following the manufacturer's protocol (10x Genomics, CG000495) and sequenced with P3 reagents (100 cycles) on a NextSeq 2000 system (Illumina) at a minimum sequencing depth of 50,000 read pairs per spatial spot. Sequencing was performed with the recommended protocol (read 1, 28 cycles; i7 index read, 10 cycles; i5 index read, 10 cycles; read 2, 50 cycles), yielding 600 million sequenced reads for each sample. Raw FASTQ files and histology images were processed and analyzed using the ParteK flow software.

### Statistical analysis

Statistical analyses were performed using two-tailed unpaired or paired Student's $t$-test, as specified. Values are presented as the mean ± s.e.m. Data were analyzed using GraphPad Prism 9 software.

### Reporting summary

Further information on research design is available in the Nature Portfolio Reporting Summary linked to this article.

## Data availability

scRNA-seq data and bulk mRNA-seq data from mice and CRISPR screening and spatial transcriptomic data from human tissues are available via Zenodo at https://doi.org/10.5281/zenodo.15309075 (ref. 43) and https://doi.org/10.5281/zenodo.10908256 (ref. 44). Data are accessible on request. Source data are provided with this paper.

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

## Acknowledgements

We thank A. Doni and M. Erreni (Humanitas Research Hospital) for help with the Hyperion imaging system. This work was supported by Associazione Italiana per la Ricerca sul Cancro (AIRC IG 2023N. 29492 to D.D.M., Bridge grant-ID 27673 to D.D.M., AIRC 5×1000 2019-ID22757 to D.D.M.) and Minsal (Gr-2016-02363531, GR-2019-1236851 to D.D.M.). D.D.M. also received funds from Transcan3 (JTC2021) and FIS (Fondo Italiano per la Scienza) FIS00002519. F.B. receives funds from Swiss Cancer Research (KFS-5502-02-2022). G.M. is supported by a fellowship from Associazione Italiana per la Ricerca sul Cancro (AIRC-ID 22588). F.P. is supported by a fellowship from Pezcoller Foundation. E.L. is a CRI Lloyd J. Old STAR (CRI award 3914) and is supported by the Associazione Italiana per la Ricerca sul Cancro (AIRC IG 2022–ID 27391 and AIRC 5×1000 program UniCanVax 22757) and by EU funding within the MUR PNRR Italian Network of Excellence for Advanced Diagnosis (INNOVA, project no. PNC-E3-2022-23683266 PNC-HLS-DA).

## Author contributions

Conceptualization: G.M. and D.D.M. Methodology: G.M., D.D.M., G.S., F.B. and H.M. Investigation: G.M., N.M., M.I., M.P., F.P., S.P., M.C., E.P., J.C., G.B., C.P., G.D.S., C.C., D.M., E.D., G.M., G.S., F.B., L.M., E.K. and H.M. Visualization: G.M. and D.D.M. Funding acquisition: G.M., D.D.M. and F.P. Project administration: D.D.M. Supervision: D.D.M., E.L., R.A., C.P., P.C., M.L., P.C., E.B., F.B. and L.W. Writing: G.M. and D.D.M.

## Competing interests

E.L. received research funding from Bristol Myers Squibb unrelated to this study and consulting fees from Swarm Therapeutics, Menarini, Amgen, Pfizer and BioLegend. F.B. has received institutional research funds from ADC Therapeutics, Bayer AG, BeiGene, Floratek Pharma, Helsinn, HTG Molecular Diagnostics, Ideogen AG, Idorsia Pharmaceuticals Ltd., Immagene, ImmunoGen, Menarini Ricerche, Nordic Nanovector ASA, Oncternal Therapeutics and Spexis AG; consultancy fees from BIMINI Biotech, Floratek Pharma, Helsinn, Immagene, Menarini and Vrise Therapeutics; advisory board fees to the institution from Novartis. F.B. has also provided expert statements to HTG Molecular Diagnostics and has received travel grants from Amgen, Astra Zeneca and iOnctura. H.M. has received royalties from Chemcom, research grants from Givaudan and consultant fees from Kao. The remaining authors declare no competing interests.

## Additional information

**Extended data** is available for this paper at https://doi.org/10.1038/s41590-025-02191-x.

**Correspondence and requests for materials** should be addressed to Diletta Di Mitri.

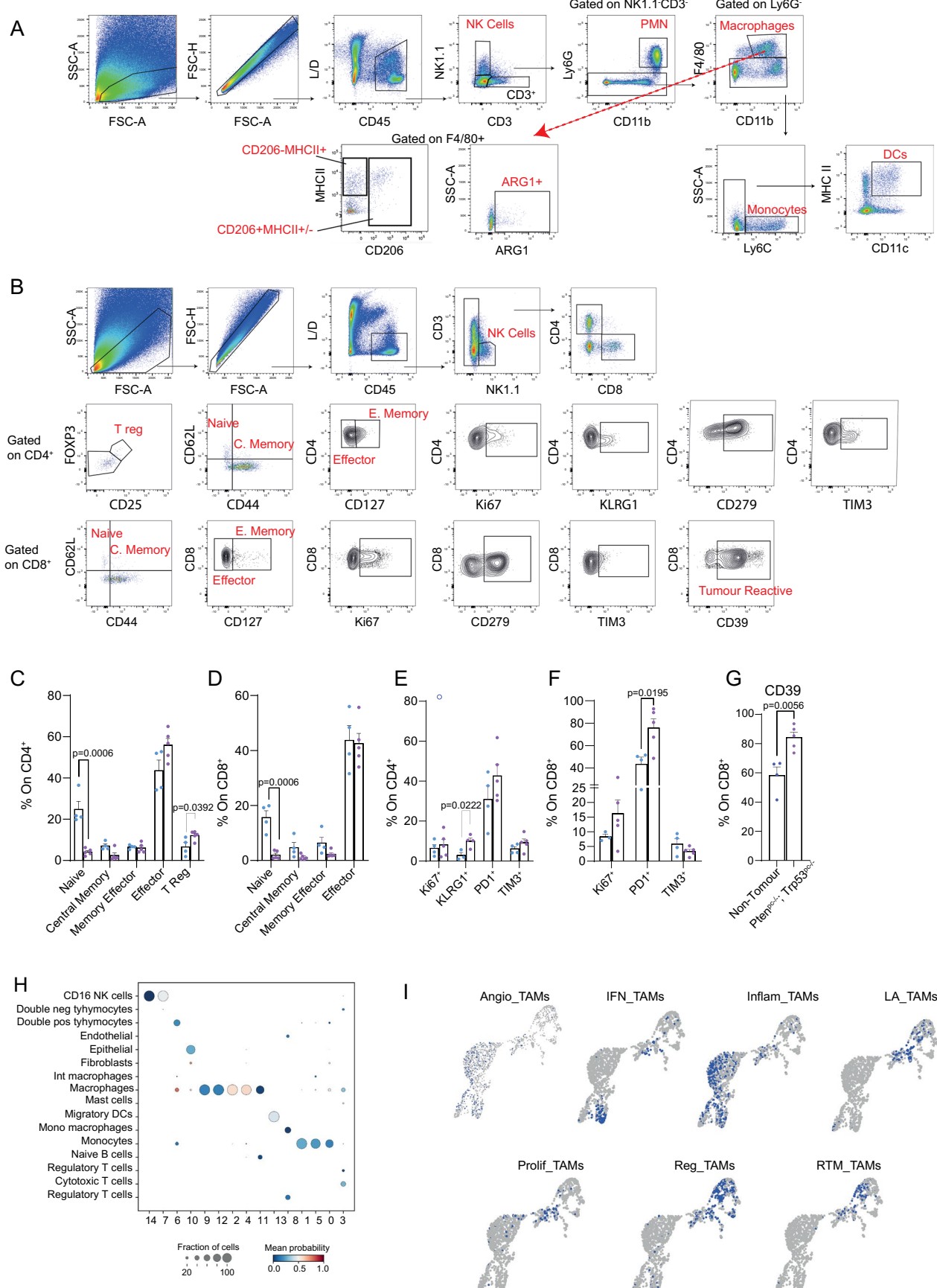

**Extended Data Fig. 1 | See next page for caption.**

**Extended Data Fig. 1 | Analysis of the tumor immune infiltrate in transgenic Pten^pc-/-; Trp53^pc-/- model. a-b**) Gating strategy used to define the myeloid and lymphoid sub-populations. Representative FACS plot from the Pten^pc-/-; Trp53^pc-/- model. **c**) Events are gated on CD45^+ cells. **d-e**)% of naive, central memory, effector, effector memory and Treg cells gated on the (**d**) CD4^+ subset and the (**e**) CD8+ subset. **f-g**) % of cells expressing functional markers gated on CD4+ and CD8+ cells. **g**) % of CD39+ cells gated on CD8+ cells. **h**) Cell Typist classification: each cluster is annotated with its predicted cell type. **i**) UCell score distribution in UMAP space for seven gene signatures. Statistical analyses were performed using two-tailed unpaired Student's t test. Values are presented as mean ± SEM.

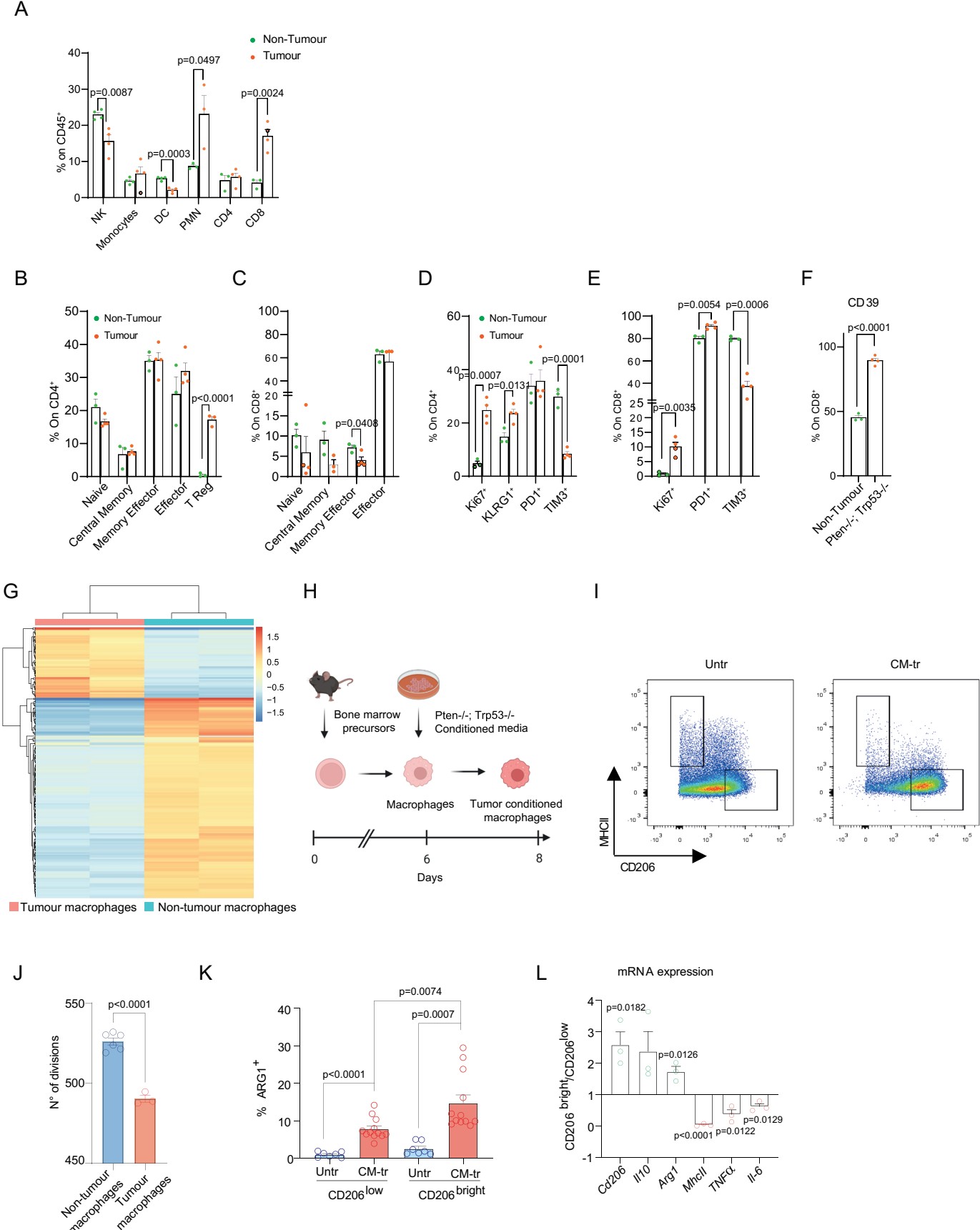

**Extended Data Fig. 2 | See next page for caption.**

**Extended Data Fig. 2 | Analysis of the tumor immune infiltrate in transgenic Pten-/-; Trp53-/- orthotopic model. a**) Bar graphs showing results of the FACS analysis on the Pten-/-; Trp53-/- orthotopic model. Same strategy as shown in Extended Data Fig. 1. Events are gated on CD45+ cells. **b-c**) % of naive, central memory, effector, effector memory and Treg cells gated on the (**b**) CD4+ subset and the (**c**) CD8+ subset. **d-e**) % of cells expressing functional markers gated on CD4+ and CD8+ cells. **f**) % of CD39+ cells gated on CD8+ cells. **g**) Heat map illustrating top 200 differentially expressed genes from bulk mRNA-Seq on sorted macrophages from the Pten-/-; Trp53-/- orthotopic model. **h**) Experimental scheme of in vitro conditioning of macrophages. **i**) Representative FACS plot showing non-conditioned macrophages (Untreated=Untr, left) and macrophages exposed to conditioned media from Pten-/-; Trp53-/- cells (Conditioned =CM-tr, right). **j**) Proliferation of CD8 + T cells exposed to supernatant derived from ex vivo macrophages. Macrophages were sorted from either tumor (n = 3) or healthy tissues (n = 6) and cultured for two days to collect conditioned supernatant. CD8 + T cells were subsequently exposed to these supernatants to assess their proliferation: bar graph represents the number of divisions. **k**) % of ARG1+ cells on CD206 + FACS-sorted macrophages (Untr n = 7, CM-tr n = 12. L) RT-qPCR gene expression analysis on CD206+ and CD206- FACS sorted macrophages (n = 3). Statistical analyses were performed using two-tailed unpaired Student's t test. Values are presented as mean ± SEM. Schematic in **h** created using BioRender.com.

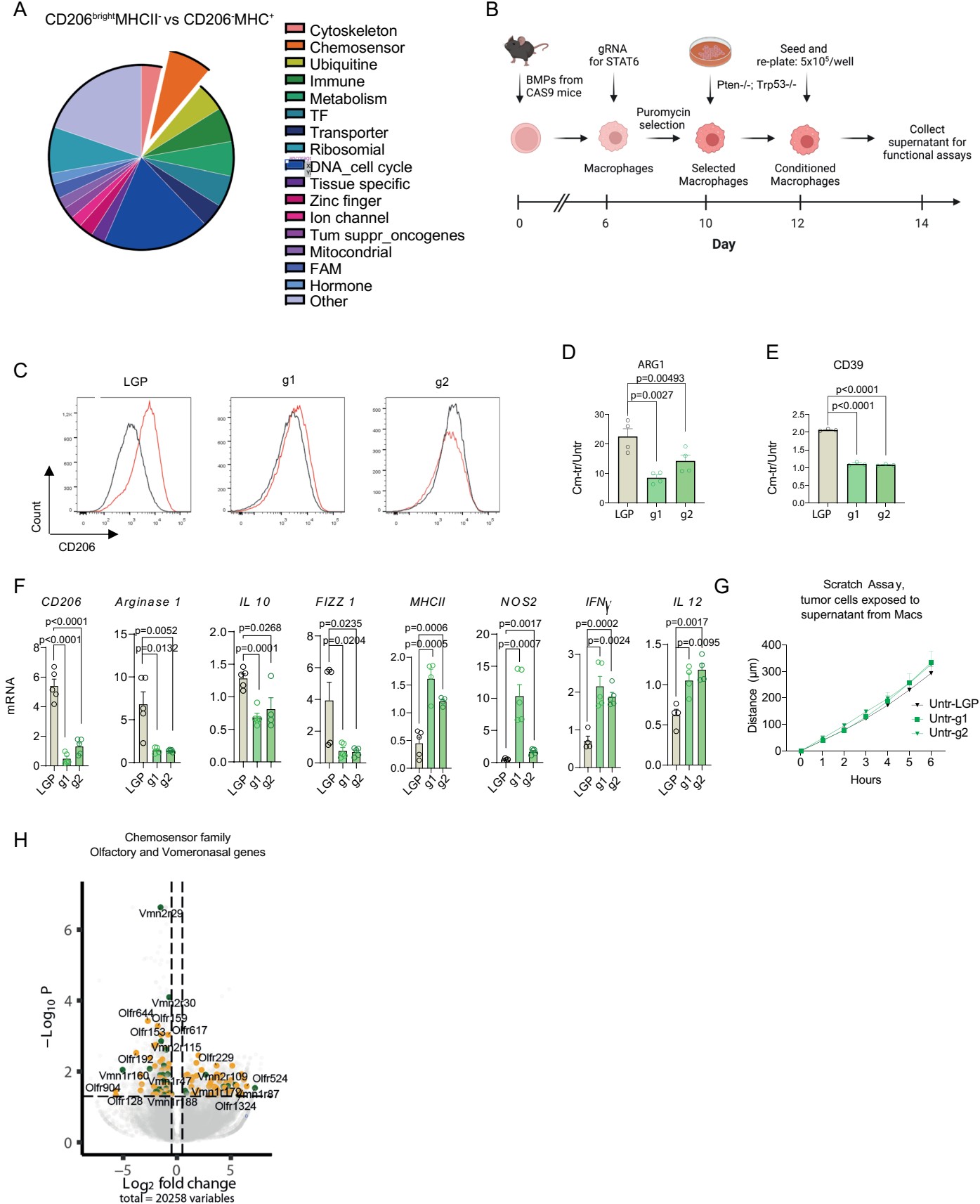

**Extended Data Fig. 3 | See next page for caption.**

**Extended Data Fig. 3 | Stat6 genetic deletion in TAMs. a**) Macrophages transduced with the CRISPR-Cas9 GeCKOv2 library were sorted based on MHC II and CD206 expression. Negative regulators were ranked based on pvalue (<=-0.005) and the first 200 genes were grouped into functional families. The pie chart shows the percentage distribution of the identified genes. **b**) Experimental scheme of in vitro Stat6 genetic deletion on primary bone marrow derived macrophages. Two independent sgRNA (g1 and g2) have been employed. **c**) Representative FACS plot showing the expression of CD206 and MHCII on LGP (CTRL) and Stat6 KO (g1 and g2) macrophages. **d-e**) FACS analysis of control (LGP) and STAT6 silenced (g1 and g2) macrophages upon exposure to Pten-/-; Trp53-/- conditioned media: (**d**) % of ARG1+ (LGP n = 4, g1 n = 4, g2 n = 4) and (**e**)

CD39+ cells (LGP n = 3, g1 n = 3, g2 n = 3) gated on F4/80 + CD11b+ cells. **f**) RT-qPCR gene expression analysis on LGP or Stat6 KO macrophages exposed to Pten-/-; Trp53-/- media. Bar graphs show the fold change of treated cells versus untreated for each condition **g**) Scratch assay: graph and curves showing the distance (um) covered by tumor cells over time after exposure to supernatant from untreated macrophages (n = 9/group). **h**) Volcano plot showing chemosensor genes related to the differentially enriched sgRNA guides from CD206-MHCII+ vs CD206brightMHCII- cells. Olfactory receptors are shown in orange, Vomeronasal receptors are shown in green. Statistical analyses were performed using two-tailed unpaired Student's t test. Values are presented as mean ± SEM. Schematic in **b** created using BioRender.com.

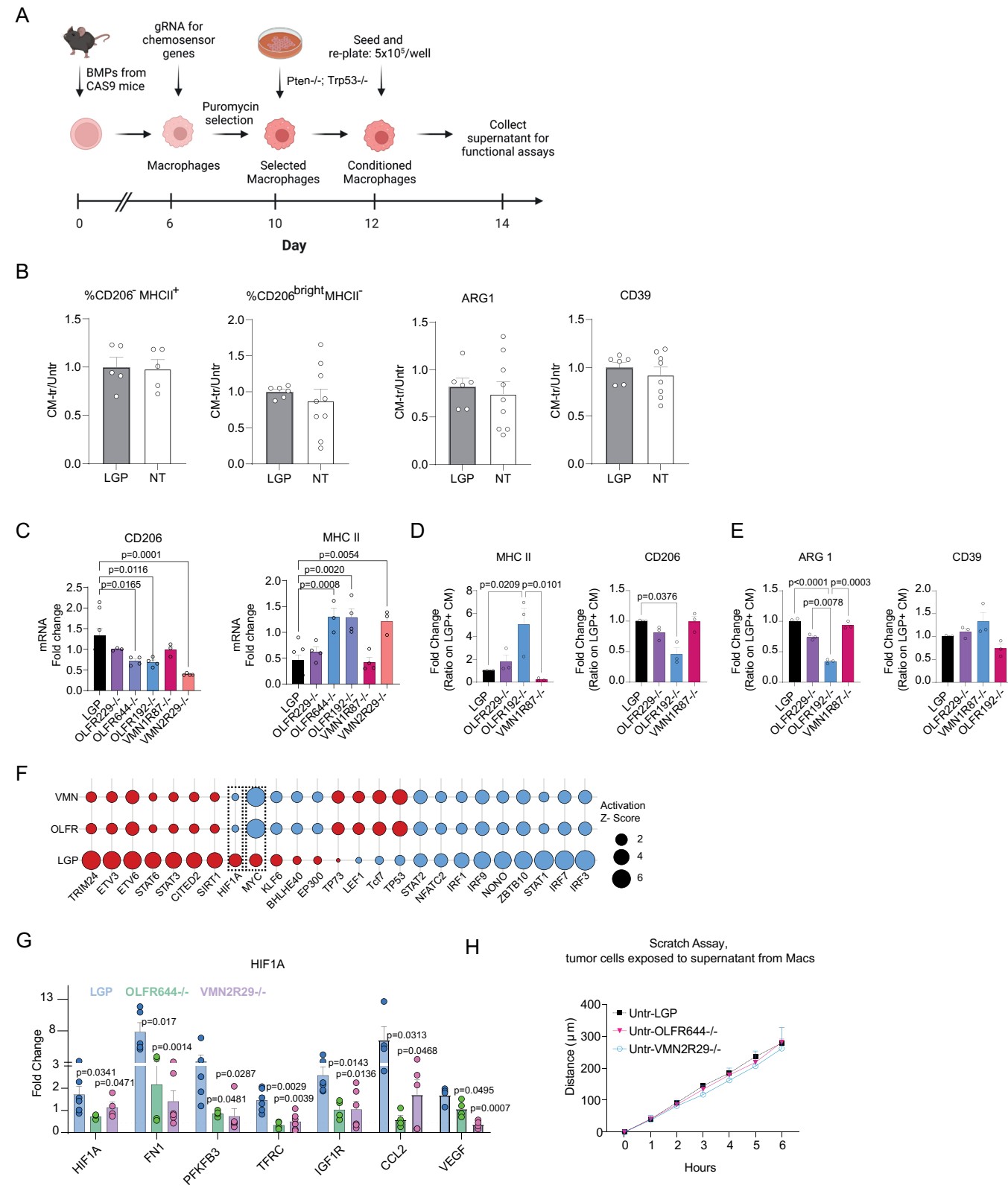

Extended Data Fig. 4 | See next page for caption.

**Extended Data Fig. 4 | Chemosensor genetic deletion re-educate TAMs.**
**a**) Experimental scheme of OLFR644 and VMN2R29 genetic deletion on
primary bone marrow derived macrophages. **b**) FACS analysis of LGP and NT
macrophages exposed to Pten-/-; Trp53-/- media. NT= non- targeting guides.
(CD206-MHCII + LGP n = 5, NT n = 5; CD206BrightMHCII- LGP n = 6, NT n = 9;
ARG LGP n = 6, NT n = 9; CD39 LGP n = 6, NT n = 8). **c**) RT-qPCR gene expression
analysis on LGP or chemosensor KO macrophages. Bar graphs show the fold
change of treated cells versus untreated for each condition. (CD206: LGP
n = 4, OLFR229-/- n = 3, OLFR644-/- n = 4, OLFR192-/- n = 4, VMN1R87-/- n = 3,
VMN2R29-/- n = 4; MHCII: LGP n = 4, OLFR229-/- n = 4, OLFR644-/- n = 3, OLFR192-/-
n = 4, VMN1R87-/- n = 4, VMN2R29-/- n = 3). **d-e**) FACS analysis on macrophages
in absence (LGP n = 4) or presence of OLFR229 deletion (n = 3), OLFR192 deletion
(n = 3) and VMN1R87 deletion (n = 3). Events are gated on F4/80 + CD11b+ cells.
Bar graphs show the ratio between conditions: (**d**) % of CD206-MHCII+ and %
of CD206brightMHCII- (**e**) % of ARG1+ and % of CD39+ cells. (**d**) and (**e**) one-way
ANOVA test with Tukey's multiple comparisons test was used. **f**) Bulk mRNA-seq
was performed on non-conditioned LGP, Olfr644-/-, and Vmn2r29-/- Macs and
LGP, Olfr644-/-, and Vmn2r29-/- Macs exposed to conditioned media from Pten-/-;
Trp53-/-cells. The balloon plot shows predicted upstream regulators identified
using Ingenuity Pathway Analysis (IPA) by comparing macrophages exposed
to conditioned medium and their respective untreated controls. In the LGP
condition, genes were ranked based on their pvalue and the top 25 genes were
selected. Each balloon represents an upstream regulator, with size indicating
the activation Z-score and color reflecting its predicted activation state (red =
activated, blue = inhibited). **g**) RT-qPCR gene expression analysis on Hif1a
downstream genes performed on LGP or chemosensor- KO macrophages. Bar
graphs show the fold change of treated cells versus untreated for each condition.
**h**) Scratch assay: graph and curves showing the distance (um) covered by tumor
cells over time after exposure to supernatant from untreated macrophages.
When not specified, statistical analyses were performed using two-tailed
unpaired Student's t test. Values are presented as mean ± SEM. Schematic in **a**
created using BioRender.com.

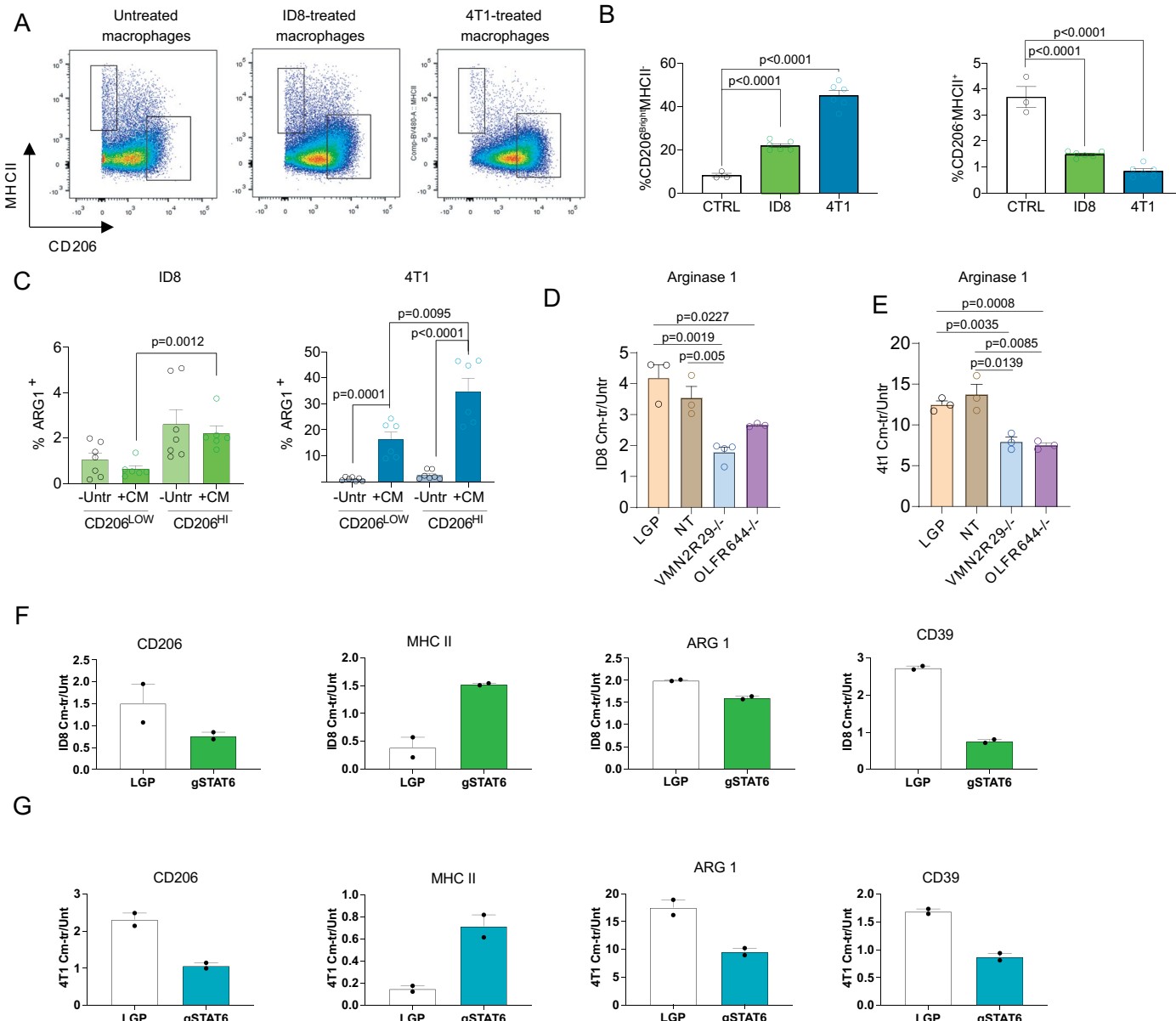

**Extended Data Fig. 5 | Role of chemosensors in macrophages in models of ovarian and breast cancer. a-b**) FACS plot and quantification of macrophages conditioned with conditioned media from ID8 or 4T1 cancer cells. (**b**) % of cells gated on F4/80+ cells (Ctrl n = 3, ID8 n = 6, 4T1 n = 6). **c**) % ARG1+ cells in CD206low and CD206bright FACS-sorted macrophages (Untr n = 7, Untr+CM n = 6). **d-e**) % ARG1+ cells gated on F4/80+ cells (ID8: LGP n = 3, NT n = 3,

VMN2R29-/- n = 4, OLFR644-/- n = 3, 4T1: LGP n = 3, NT n = 3, VMN2R29-/- n = 3, OLFR644-/- n = 3). **f-g**) FACS analysis of LGP or Stat6 KO macrophages exposed to ID8 (**f**) or 4T1 (**g**) conditioned media (n = 2). Statistical analyses were performed using two-tailed unpaired Student's t test. Values are presented as mean ± SEM (*, P < 0.05; **, P < 0.01; ***, P < 0.001).

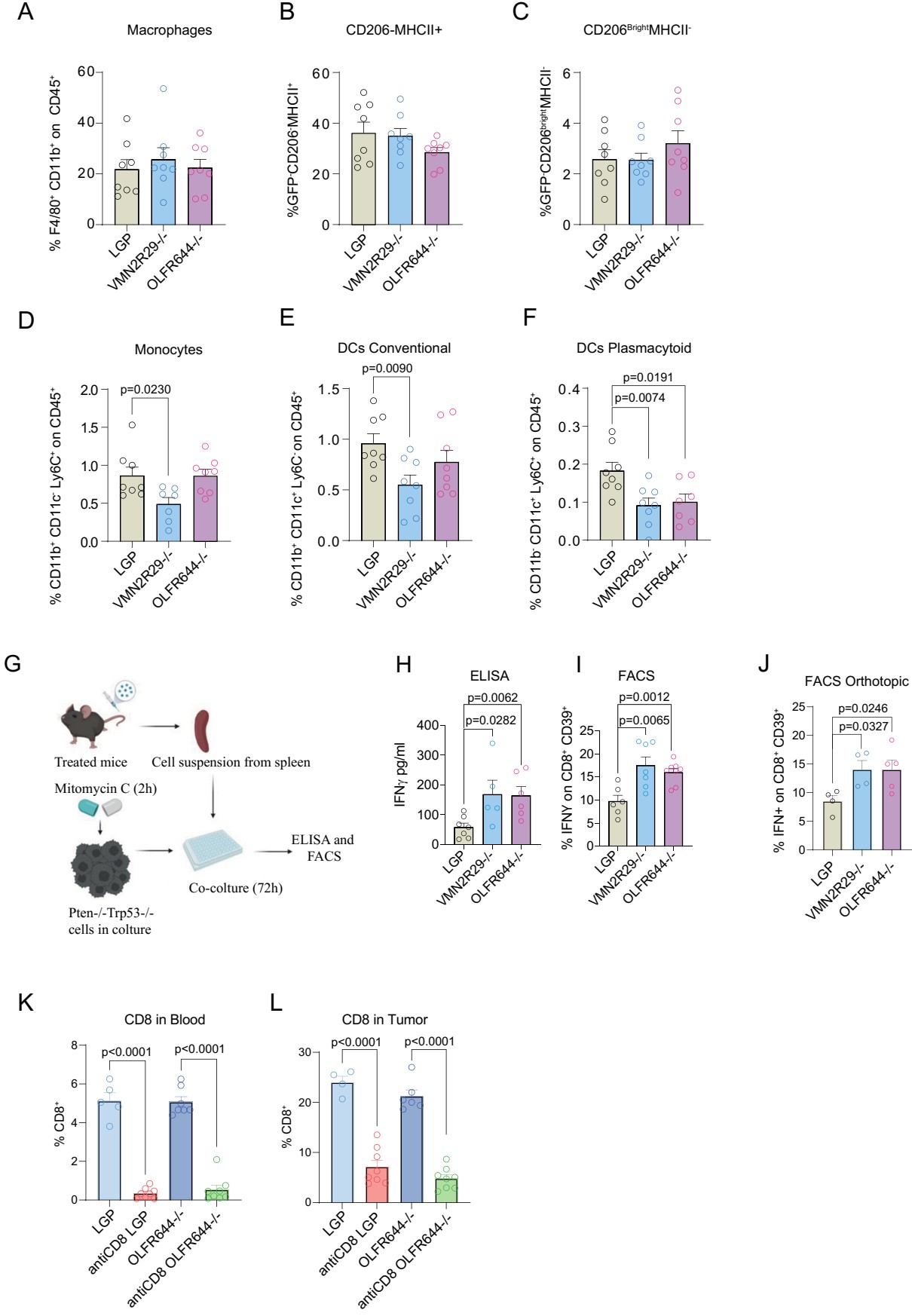

**Extended Data Fig. 6 | See next page for caption.**

**Extended Data Fig. 6 | Composition of the TME in mice injected with either OLFR644-/- or VMN2R29-/- macrophages. a-f**) FACS analysis to determine the immune infiltrate of tumors upon injection with LGP (n = 8), OLFR644-/- (n = 8) and VMN2R29-/- (n = 8) macrophages: a) Quantification of macrophages gated on CD45+ cells. **b-c**) % cells gated on GFP-F4/80 + CD11b+ cells. (**d-f**) % cells gated on CD45+ cells. **g-i**) The response of splenocytes to mitomycin **c** killed tumor cells was examined ex vivo using tumor cell restimulation assays. Interferon production in response to stimulation was assessed through ELISA (LGP n = 7,

OLFR644 n = 5 and VMN2R29 n = 6) (**h**) or FACS analysis (LGP n = 6, OLFR644 n = 7 and VMN2R29 n = 8. (**i**) after a 72-hour incubation period. **j**)FACS analysis of interferon production in splenocytes from mice contextually injected orthotopically with Pten-/-; Trp53-/- and with LGP-Macs, OLFR644-/-Macs or VMN2R29-/- Macs. **k-l**) FACS analysis of CD8 + T lymphocytes to confirm the CD8 depletion in mice treated with antiCD8 antibody or isotype in (**k**) blood or (**l**) tumor. Events are gated on CD3+ cells. Statistical analyses were performed using two-tailed unpaired Student's t test. Values are presented as mean ± SEM.

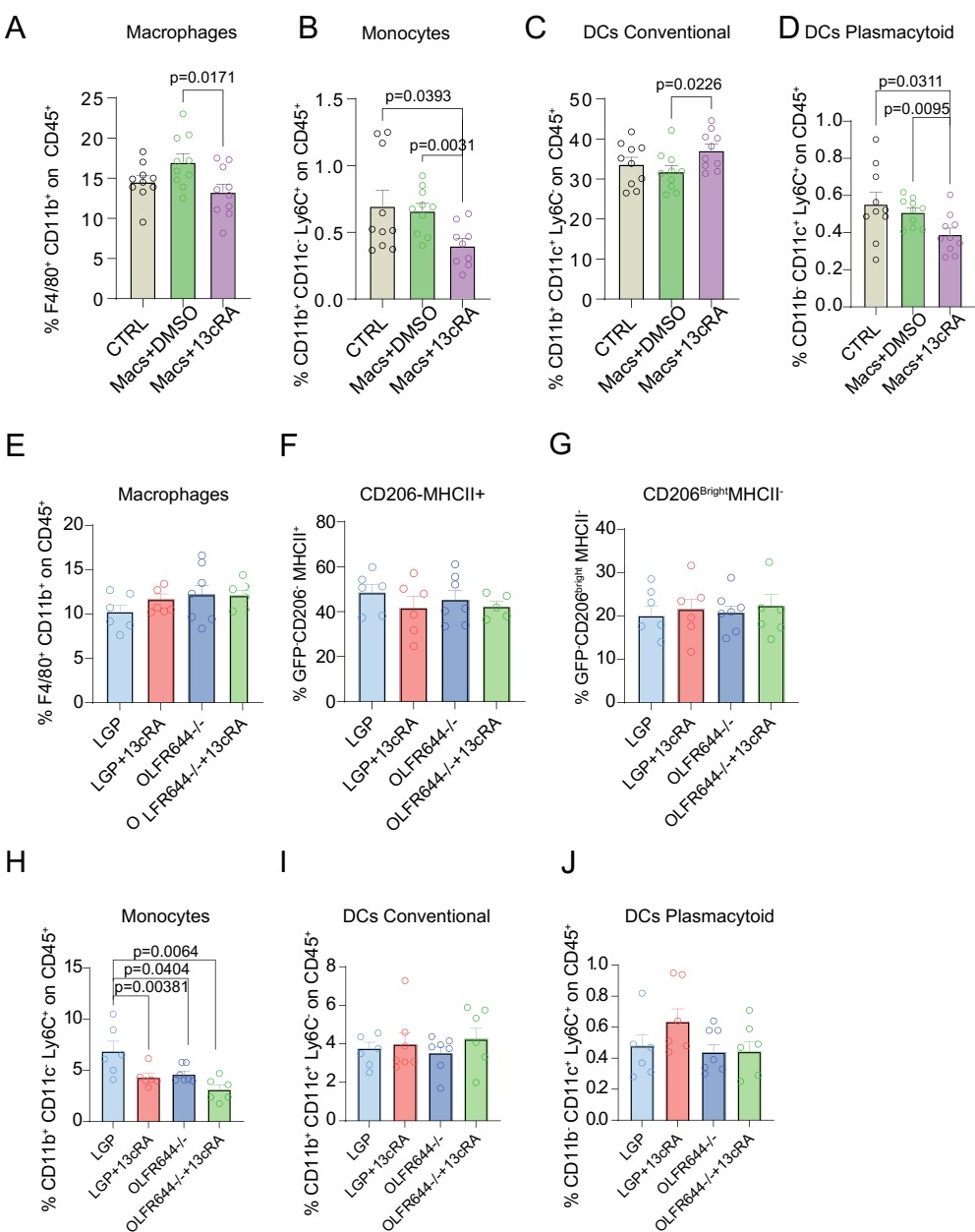

**Extended Data Fig. 7 | Composition of the TME in mice injected with macrophages exposed to 13-cRA. a-d)** FACS analysis to determine the immune infiltrate of tumors upon injection with macrophages pre-treated for 4 hours with DMSO or 13-cRA: (**a**) Quantification of macrophages gated on CD45+ cells. (**b**) % of monocytes gated on CD45+ cells. (**c-d**) % of dendritic cells gated on CD45+ cells (**a-d** Untr, Macs+DMSO or 13cRA n = 10). **e-j)** FACS analysis to determine the immune infiltrate of tumors upon injection with OLFR644-/-

and VMN2R29-/- macrophages pre-treated for 4 hours with DMSO or 13-cRA: **e)** Quantification of macrophages gated on CD45+ cells. **f-g)** % cells gated on GFP-F4/80 + CD11b+ cells. (**h-j**) % cells gated on CD45+ cells. (E-J: LGP n = 6, LGP+13cRA n = 6, OLFR644 n = 7, OLFR644 + 13cRA n = 6). Statistical analyses were performed using two-tailed unpaired Student's t test. Values are presented as mean ± SEM.

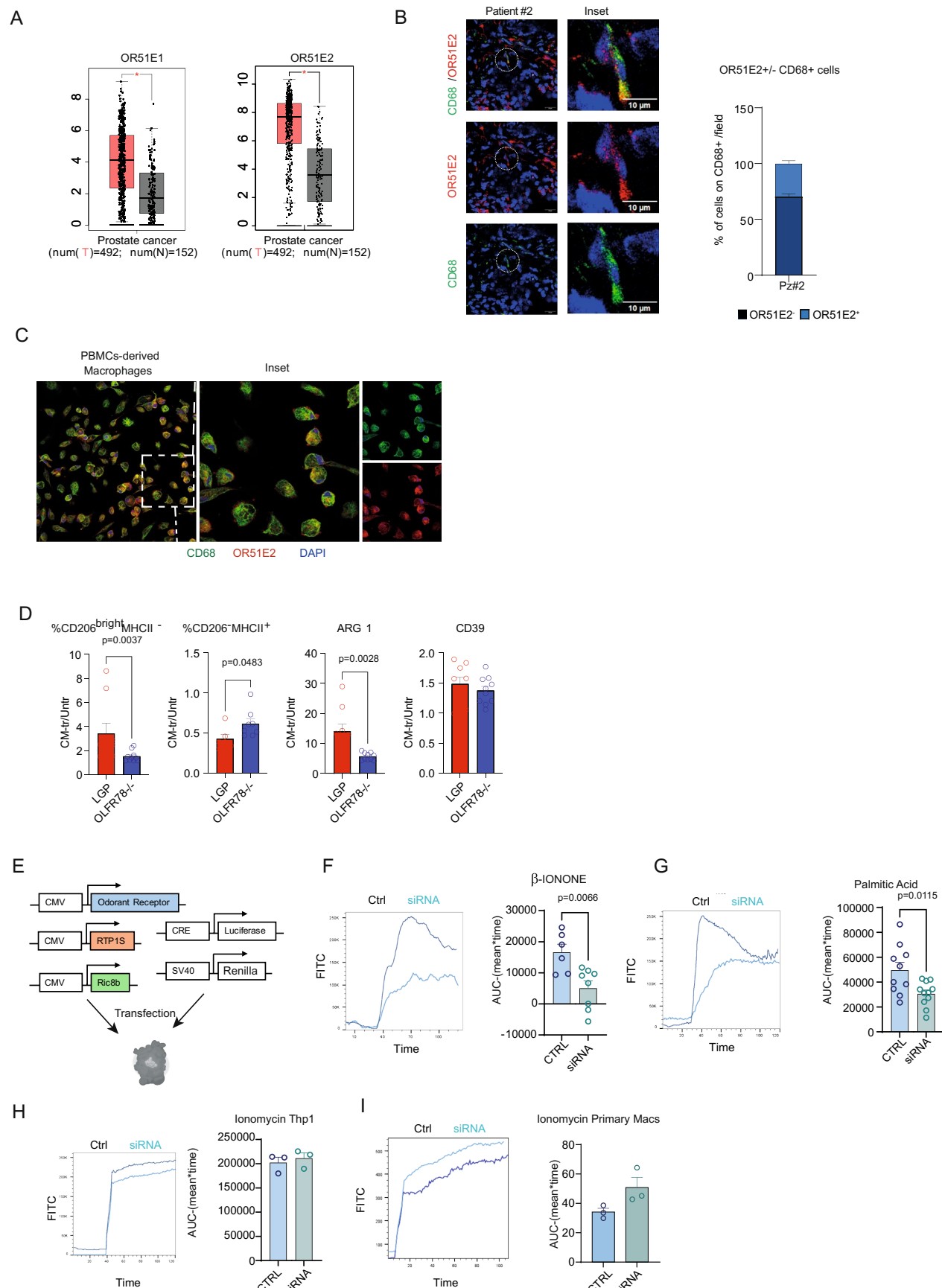

**Extended Data Fig. 8 | See next page for caption.**

**Extended Data Fig. 8 | Expression of OR51E2 by human macrophages. a**) OR51E1 and OR51E2 expression in PCa human tissues respect to normal tissue. TGSA data were analysed using GEPIA2 (n tum= 492, n Normal= 152) and presented as Min to Max box-and-whisker plot, the box extends from the 25th to 75th percentiles and the whiskers reach the sample maximum and minimum values, the median is indicated at center line. **b**) Representative confocal immunofluorescence images and quantification of human PCa tissues (patient#2) showing the expression of OR51E2 (red) in CD68+ macrophages (green). Images were acquired with an SP8-II confocal microscope (Leica). Scale bar: 10um. **c**) Representative confocal immunofluorescence images showing PBMC- derived macrophages (CD68 + , green) expressing OR51E2 (red). Nuclei were counterstained with DAPI (blue). Images were acquired with an SP8- II confocal microscope (Leica) with a 40× objective. Images on the right are 2× digital zoom. **d**) Flow cytometry analysis to assess the impact of gene silencing on murine bone marrow derived macrophages. Control macrophages (LGP) were compared to OLFR78-KO (Or51e2-KO) macrophages after exposure to Pten-/-; Trp53-/- conditioned media. Cells were gated on F4/80 + CD11b+ cells. **e**) Schematic representation of the plasmid encoding OR, RTP1S, CRE-luciferase and 5 pRL-SV40. **f-g**) Ca2+ flux in THP1 cells in absence or presence of partial genetic deletion of OR51E2 KO treated with (**f**) β-ionone (100 μM) (Ctrl n = 6, treated n = 8), (**g**) palmitic acid (100 μM) (Ctrl n = 10, treated n = 11), or (**h**) ionomycin (2 μM) (Ctrl n = 3, treated n = 3). **i**) Ca2+ flux in primary macrophages in absence or presence of partial genetic deletion of OR51E2 KO treated with ionomycin (2 μM) (Ctrl n = 3, treated n = 3). Values are presented as mean ± SEM.

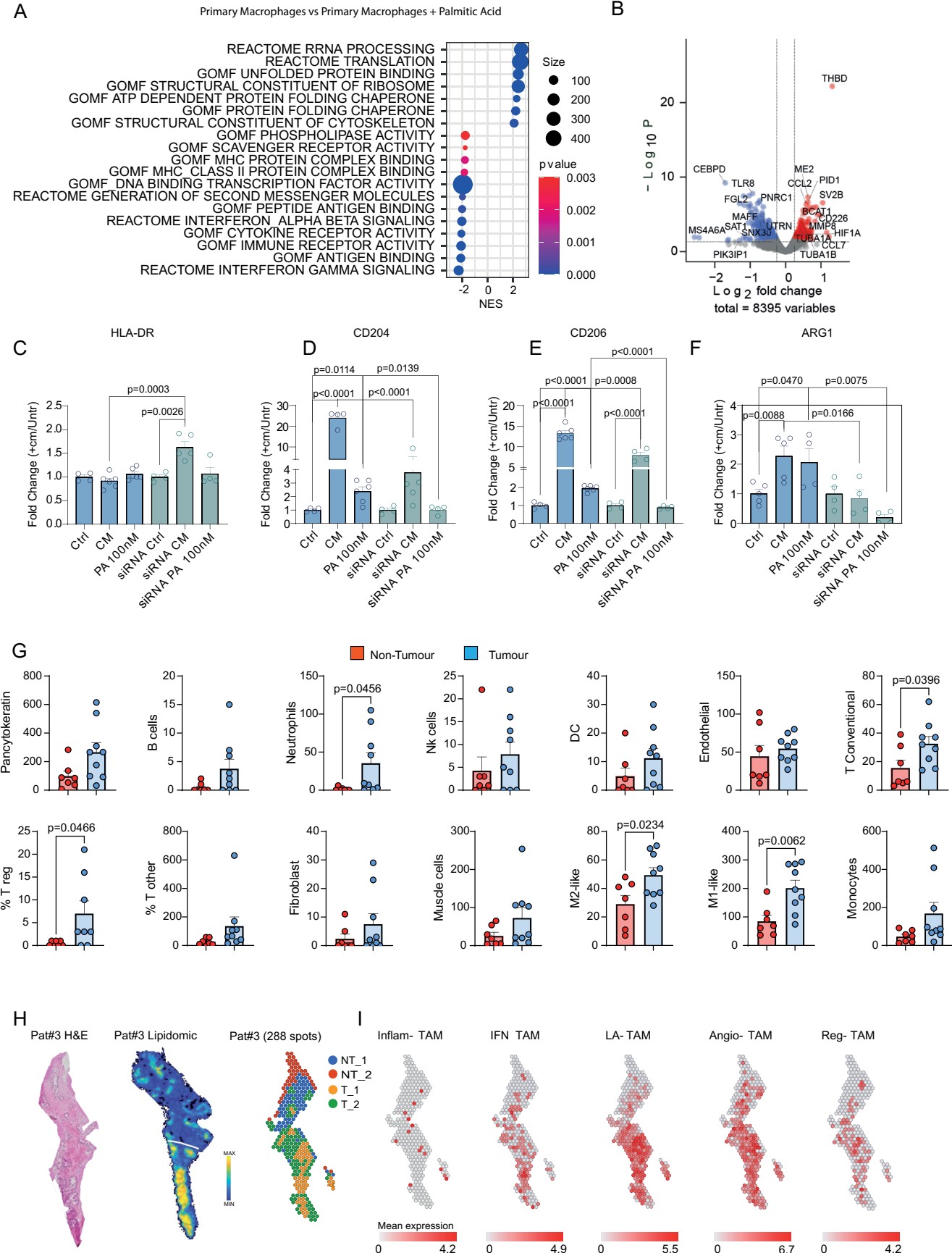

**Extended Data Fig. 9 | See next page for caption.**

**Extended Data Fig. 9 | Expression of OR51E2 by human macrophages.**
**a**) Gene set enrichment analysis (GSEA) showing up and downregulated biological pathways in BPMCs derived macrophages exposed or not to Palmitic acid. The size of each dot indicates the number of enriched genes relative to the pathway of interest. The fraction of genes represents the proportion of genes significantly enriched out of the total number of genes of the pathway. **b**) Volcano plot showing differential expressed genes in palmitic acid-exposed macrophages compared to untreated macrophages. Genes are colored according to their log2 fold change value (Blue <=−0.5, red >=+0.5). **c-f**) FACS analysis of primary macrophages exposed to the conditioned media from PC3 tumor cells or to palmitic acid (100 nM) in presence or absence of partial genetic deletion of

OR51E2. Bar graphs showing the % of (**c**) HLA-DR+ (Ctrl n = 4, CM n = 6, PA n = 6, siRNA ctrl n = 4, siRNA CM n = 5, siRNA PA n = 4), (**d**) CD204+ (Ctrl n = 4, CM n = 4, PA n = 6, siRNA ctrl n = 4, siRNA CM n = 5, siRNA PA n = 4), (**e**) CD206+ (Ctrl n = 4, CM n = 6, PA n = 6, siRNA ctrl n = 4, siRNA CM n = 4, siRNA PA n = 4) and (**f**) Arginase1+ (Ctrl n = 5, CM n = 5, PA n = 4, siRNA ctrl n = 4, siRNA CM n = 4, siRNA PA n = 4) gated on CD68+ macrophages. **g**) Absolute count of different cell subsets in non-tumor (n = 7) and tumor (n = 9) regions. **h**) H&E and spatial analysis of the distribution of palmitic acid by mass spectrometry imaging in patient number 3. **i**) Spatial distribution of Inflamm-TAMs, IFN TAMs, LA-TAMs, Angio-TAMs and Reg-TAMs signature in patient number #3. Statistical analyses were performed using two-tailed unpaired Student's t test. Values are presented as mean ± SEM.

|---|---|

# Reporting Summary

## Statistics

For all statistical analyses, confirm that the following items are present in the figure legend, table legend, main text, or Methods section.

| n/a | Confirmed | |
|---|---|---|
| ☐ | ☒ | The exact sample size (*n*) for each experimental group/condition, given as a discrete number and unit of measurement |
| ☐ | ☒ | A statement on whether measurements were taken from distinct samples or whether the same sample was measured repeatedly |
| ☐ | ☒ | The statistical test(s) used AND whether they are one- or two-sided<br>*Only common tests should be described solely by name; describe more complex techniques in the Methods section.* |
| ☒ | ☐ | A description of all covariates tested |
| ☐ | ☒ | A description of any assumptions or corrections, such as tests of normality and adjustment for multiple comparisons |
| ☐ | ☒ | A full description of the statistical parameters including central tendency (e.g. means) or other basic estimates (e.g. regression coefficient) AND variation (e.g. standard deviation) or associated estimates of uncertainty (e.g. confidence intervals) |
| ☐ | ☒ | For null hypothesis testing, the test statistic (e.g. *F*, *t*, *r*) with confidence intervals, effect sizes, degrees of freedom and *P* value noted<br>*Give P values as exact values whenever suitable.* |
| ☒ | ☐ | For Bayesian analysis, information on the choice of priors and Markov chain Monte Carlo settings |
| ☒ | ☐ | For hierarchical and complex designs, identification of the appropriate level for tests and full reporting of outcomes |
| ☐ | ☒ | Estimates of effect sizes (e.g. Cohen's *d*, Pearson's *r*), indicating how they were calculated |

*Our web collection on statistics for biologists contains articles on many of the points above.*

## Software and code

Policy information about availability of computer code

| Data collection | Gepia2 software was utilized to analyze gene expression data from the TGCA dataset. |
|---|---|
| Data analysis | RNA-seq and bioinformatics analysis were performed to analyze the gene expression profiles of the samples. The following steps were carried out:<br><br>Sequencing and Quality Control:<br>All samples were sequenced on an Illumina NextSeq500 platform, generating an average of 15 million 75-bp single-end reads per sample. After sequencing, quality control checks were performed to assess the data quality and remove low-quality reads or artifacts.<br><br>Alignment and Read Counting:<br>The quality-filtered reads were aligned to the human genome (GRCm38) using the STAR aligner with default parameters (version 2.6.1). The aligned reads were then used to obtain gene-based read counts using the featureCounts module (version 1.6.4) and the Ensembl GRCm38 annotation.<br><br>Normalization and Differential Gene Expression Analysis:<br>To compare expression levels across samples, raw read counts were normalized using the TMM (trimmed mean of log-ratio values) method. Genes with counts per million (CPM) mapped reads greater than 1 in at least 2 libraries were considered for further analysis. The edgeR package (version 3.26.5) in R statistical software was used to perform differential gene expression analysis.<br><br>CRISPR-Cas9 Analysis:<br>To analyze the DNA sequencing data derived from the CRISPR-Cas9 library, the software MAGECK (Model-based Analysis of Genome-wide |

CRISPR-Cas9 Knockout) was utilized.

For manuscripts utilizing custom algorithms or software that are central to the research but not yet described in published literature, software must be made available to editors and reviewers. We strongly encourage code deposition in a community repository (e.g. GitHub). See the Nature Portfolio guidelines for submitting code & software for further information.

## Data

Policy information about availability of data

All manuscripts must include a data availability statement. This statement should provide the following information, where applicable:
- Accession codes, unique identifiers, or web links for publicly available datasets
- A description of any restrictions on data availability
- For clinical datasets or third party data, please ensure that the statement adheres to our policy

A data avaiability statment has been included

## Research involving human participants, their data, or biological material

Policy information about studies with human participants or human data. See also policy information about sex, gender (identity/presentation), and sexual orientation and race, ethnicity and racism.

| | |
|---|---|
| Reporting on sex and gender | Enrolled patients were all male considering the nature of the study (prostate cancer) |
| Reporting on race, ethnicity, or other socially relevant groupings | No socially relevant groupping was applied to the study |
| Population characteristics | Prostate cancer patients that received biopsy for their clinical practice, in absence of prior therapy. |
| Recruitment | Tissue samples were retrieved from routine biopsies of patients diagnoses with PCa |
| Ethics oversight | The protocol was approved by the Ethical Commitee of Humanitas Clinical and research Hospital |

Note that full information on the approval of the study protocol must also be provided in the manuscript.

# Field-specific reporting

Please select the one below that is the best fit for your research. If you are not sure, read the appropriate sections before making your selection.

☒ Life sciences  ☐ Behavioural & social sciences  ☐ Ecological, evolutionary & environmental sciences

For a reference copy of the document with all sections, see nature.com/documents/nr-reporting-summary-flat.pdf

# Life sciences study design

All studies must disclose on these points even when the disclosure is negative.

| | |
|---|---|
| Sample size | In most experiments sample size was calculated with a simple size calculator, basing on preliminary data |
| Data exclusions | No data were excluded |
| Replication | Experiments were repeated and the number of each experiment is reported in the figure legend |
| Randomization | Samples, mice and human samples were randomly assigned to experimental groups |
| Blinding | Measurement of the tumour size were performed in blind |

# Reporting for specific materials, systems and methods

We require information from authors about some types of materials, experimental systems and methods used in many studies. Here, indicate whether each material, system or method listed is relevant to your study. If you are not sure if a list item applies to your research, read the appropriate section before selecting a response.

## Materials & experimental systems

| n/a | Involved in the study | |
|---|---|---|
| ☐ | ☒ | Antibodies |
| ☐ | ☒ | Eukaryotic cell lines |
| ☒ | ☐ | Palaeontology and archaeology |
| ☐ | ☒ | Animals and other organisms |
| ☒ | ☐ | Clinical data |
| ☒ | ☐ | Dual use research of concern |
| ☒ | ☐ | Plants |

## Methods

| n/a | Involved in the study | |
|---|---|---|
| ☒ | ☐ | ChIP-seq |
| ☐ | ☒ | Flow cytometry |
| ☒ | ☐ | MRI-based neuroimaging |

# Antibodies

Antibodies used

Antibody Fluorochrome Source Identifier Application Dilution
B-actin unconjugated Abcam ab213262 Western Blot 1:1000
STAT6 unconjugated Cell Signaling 9362 Western Blot 1:1000
anti-rabbit secondary antibody HRP-conjugated Bio-Techne HAF008 Western Blot 1:1000
anti-mouse secondary antibody HRP-conjugated Bio-Techne HAF007 Western Blot 1:1000
CD45 BUV563 BD 612924 FACS 1 uL in 100
KLRG1 PerCPCy5.5 BD 563595 FACS 0.63 uL in 100
CD39 PECy7 Biolegend 143806 FACS 0.31 uL in 100
CD4 BUV496 BD 612952 FACS 1.25 uL in 100
CD8 BUV805 BD 612898 FACS 0.63 uL in 100
NK1.1 BUV395 BD 564144 FACS 0.63 uL in 100
CD45R BV570 Biolegend 103237 FACS 1.25 uL in 100
CD127 PECy5 Biolegend 135016 FACS 2.5 uL in 100
CD62L BV510 Biolegend 104441 FACS 1.25 uL in 100
CD44 PEDazzle594 Biolegend 103056 FACS 0.63 uL in 100
CD25 AF488 Biolegend 102017 FACS 0.63 uL in 100
PD1 (CD279) BV605 Biolegend 135220 FACS 2.5 uL in 100
TIM3 BV711 Biolegend 119727 FACS 1.25 uL in 100
Ki67 AF647 Biolegend 151206 FACS 0.5 uL in 100
FOXP3 BV421 Biolegend 126419 FACS 1.25 uL in 100
Ly6G BV785 Biolegend 127645 FACS 0.15 uL in 100
Ly6C APC Cy7 Biolegend 128026 FACS 0.15 uL in 100
MHC II BV480 BD 566086 FACS 0.15 uL in 100
CD206 AF647 Biolegend 141712 FACS 1.25 uL in 100
CD11b PECF594 BD 562287 FACS 0.15 uL in 100
F480 BV421 Biolegend 123132 FACS 0.63 uL in 100
CD11c PerCPCy5.5 Biolegend 117328 FACS 1.25 uL in 100
CD115 BV711 Biolegend 135515 FACS 1.25 uL in 100
CD3 BV650 Biolegend 100229 FACS 1.25 uL in 100
Arginase-1 AF700 LifeTechnologies 56-3697-80 FACS 1.25 uL in 100
IFNgamma APC Biolegend 505810 FACS 0.63 uL in 100
CD45.1 BV650 BD 563754 FACS 1 uL in 100
CD45.2 PeCy7 Biolegend 109830 FACS 1 uL in 100
CD8 BV605 Biolegend 100744 FACS 1 uL in 100
CD103 PE Biolegend 156904 FACS 2.5 uL in 100

Antibody Metal Source Identifier Application Dilution
KI67 142Nd ABCAM ab279657 IMC 1:100
CD20 176YB BD 555677 IMC 1:100
CD31 153 Eu ABCAM ab226157 IMC 1:100
CD14 145Nd ABCAM ab209971 IMC 1:100
NKp46 156Gd Biotechne AF1850 IMC 1:100
Foxp3 166Er Biolegend 320202 IMC 1:100
HLA-DR 174Yb eBioscience 14-9956-82 IMC 1:200
CD45 152Sm Cell Signalling 47937SF IMC 1:200
CD8 167Er Cell Signalling #90257 IMC 1:100
PD1 165Ho Abcam ab186928 IMC 1:200
MARCO 158Gd LsBio LS-B15577 IMC 1:50
Pan-Cytokeratin 148Nd Abcam ab264485 IMC 1:100
CD68 159Tb Abcam ab233172 IMC 1:100
COLLAGEN TYPE I 169Tm Standard Biotools 3169023D IMC 1:400
CD16 146Nd Abcam ab256582 IMC 1:100
PDL1 150Nd Cell Signalling #85164 IMC 1:50
CD11c 154Sm Abcam ab264107 IMC 1:100
CD3 170Er Standard Biotools 3170019D IMC 1:50
CD66b 161Dy Biolegend 392902 IMC 1:100
CD39 155 Gd Biolegend 328202 IMC 1:100
aSMA 141Pr Abcam ab240654 IMC 1:100

CD206 168 Er Abcam ab254471 IMC 1:100
CD163 147Sm Standard Biotools 3147021D IMC 1:100
CD74 144nD Cell Signalling #95154 IMC 1:100
CD11b 149sM Abcam ab209970 IMC 1:25

| | |
|---|---|
| Validation | All antibodies used in the study are commercially available. |
| | All antibodies used in the study were titrated before use. |
| | All antibodies have been validated by the commercial manufacturers. Validation data are available on each manufacturer's website. |

## Eukaryotic cell lines

Policy information about cell lines and Sex and Gender in Research

| | |
|---|---|
| Cell line source(s) | The Pten-/- Trp53-/- cell line was provided by R.A. DePinho. |
| | PC3, HEK293T, L929, 4T1, ID8, and THP1 cells are commercially available and were purchased from ATCC. |
| Authentication | None of the cell lines used in this work were authenticated. |
| Mycoplasma contamination | All the cell lines used were tested and were Mycoplasma-free. |
| Commonly misidentified lines (See ICLAC register) | No commonly misidentified lines were used. (See ICLAC register for reference). |

## Animals and other research organisms

Policy information about studies involving animals; ARRIVE guidelines recommended for reporting animal research, and Sex and Gender in Research

| | |
|---|---|
| Laboratory animals | Male mice were used in the study. |
| Wild animals | This study did not involve wild animals. |
| Reporting on sex | Only male mice were used considering the nature of the study (prostate cancer) |
| Field-collected samples | No field-collected samples were used in the study |
| Ethics oversight | Procedures involving animal handling and care conformed to protocols approved by the Humanitas Clinical and Research Centre and Italian Minister of Health, in compliance with national and international law and policies |

Note that full information on the approval of the study protocol must also be provided in the manuscript.

## Plants

| | |
|---|---|
| Seed stocks | *Report on the source of all seed stocks or other plant material used. If applicable, state the seed stock centre and catalogue number. If plant specimens were collected from the field, describe the collection location, date and sampling procedures.* |
| Novel plant genotypes | *Describe the methods by which all novel plant genotypes were produced. This includes those generated by transgenic approaches, gene editing, chemical/radiation-based mutagenesis and hybridization. For transgenic lines, describe the transformation method, the number of independent lines analyzed and the generation upon which experiments were performed. For gene-edited lines, describe the editor used, the endogenous sequence targeted for editing, the targeting guide RNA sequence (if applicable) and how the editor was applied.* |
| Authentication | *Describe any authentication procedures for each seed stock used or novel genotype generated. Describe any experiments used to assess the effect of a mutation and, where applicable, how potential secondary effects (e.g. second site T-DNA insertions, mosiacism, off-target gene editing) were examined.* |

# Flow Cytometry

## Plots

Confirm that:

☐ The axis labels state the marker and fluorochrome used (e.g. CD4-FITC).

☒ The axis scales are clearly visible. Include numbers along axes only for bottom left plot of group (a 'group' is an analysis of identical markers).

☒ All plots are contour plots with outliers or pseudocolor plots.

☒ A numerical value for number of cells or percentage (with statistics) is provided.

## Methodology

| | |
|---|---|
| Sample preparation | Primary macrophages were detached from the plate with Accutase Solution (Thermo Fisher Scientific), and nonspecific antibody binding was prevented by incubating cells with an Fc block (TruStain FcX anti-CD16/32, clone 93). Cells were then stained with LIVE/DEAD Fixable Viability Dye eFluor 780 (BioLegend) for 20 min at 4°C, followed by staining with the antibody mix for 30 min at room temperature. |
| | For ARG1 detection, after extracellular staining, samples were fixed and permeabilized (Intracellular Fixation & Permeabilization Buffer Set; eBioscience) and stained for 30 min at 4°C. Cells were then fixed in 1% PFA.For analysis of tumour-infiltrating leukocytes, tumours were collected, cut into small pieces, and digested with Collagenase I (1 mg/mL for mouse tissue and 0.5 mg/mL for human tissue) for 45 min at 37°C on a rocking platform. After quick digestion in 2.5% Trypsin and DNase I, single-cell suspension was obtained by mechanical dissociation through a syringe needle (18G) and subsequent filtration on a 40-μm cell strainer. |
| | Cells were stained with LIVE/DEAD Fixable Viability Dye (BioLegend) for 20 min at 4°C, followed by staining with the antibody mix for 30 min at room temperature. After extracellular staining, samples were fixed and permeabilized (Intracellular Fixation & Permeabilization Buffer Set; eBioscience) and stained with intracellular mix for 30 min at 4°C. Cells were then fixed in 1% PFA. |
| Instrument | Samples were acquired using a BD FACSymphony™ A5 Cell Analyzer. |
| Software | Data were analyzed using FlowJo software. |
| Cell population abundance | Cells frequency is shown in the relevant gates. After sorting, a small aliquot of sorted cells was used to determine purity (>90%). |
| Gating strategy | The gating strategie are shown in relative extended data. |

☒ Tick this box to confirm that a figure exemplifying the gating strategy is provided in the Supplementary Information.

