## [Peer Review File · Nature Immunology]

Chemosensor receptors are lipid-detecting regulators of macrophage functions in cancer

Corresponding Author: Dr DILETTA DI MITRI

Version 0:

Decision Letter:

3rd May 2024

Dear Dr. DI MITRI,

We have now finished reviewing your manuscript entitled "CRISPR Genome-Wide screening reveals chemosensor receptors as lipid-detecting regulators of macrophage functions in cancer", reference number NI-A37606.

Although the editors thought that the manuscript was interesting enough to send out for in-depth review, and reviewer 3 (tech ref for chemosensor biology) is keen, overall the reviewers are not in favor of publishing the paper in Nature Immunology because of concerns regarding the methods, further requirements for validation and comments about the preliminary nature of the data.

We are therefore returning the reviews to you with the hope that you find them useful when you prepare the paper for another journal. If you can address these concerns, however, you would be welcome to appeal this decision in the future if new data are supportive.

Although we cannot publish your paper, it may be appropriate for another journal in the Nature Portfolio. If you wish to explore the journals and transfer your manuscript please use our manuscript transfer portal. You will not have to re-supply manuscript metadata and files, unless you wish to make modifications. For more information, please see our [manuscript transfer FAQ](http://www.nature.com/authors/author_resources/transfer_manuscripts.html?WT.mc_id=EMI_NPG_1511_AUTHORTRANSF&WT.ec_id=AUTHOR) page.

We realize that this is disappointing. I hope that you continue to consider Nature Immunology for your results most significant for the immunology community and wish you well in your future investigations.

Sincerely,

Nick Bernard, PhD
Senior Editor
Nature Immunology

Reviewers' comments:

Reviewer #1 (Remarks to the Author):

NAT IMMUNO NI-A37606

Marelli et al: CRISPR Genome-Wide screening reveals chemosensor receptors as lipid-detecting regulators of macrophage functions in cancer

Summary

In this study, Marelli et al. employ a genome-wide CRISPR screen in primary murine bone-marrow derived macrophages (BMDMs) subsequently to these cells being exposed to a prostate cancer cell line derived conditioned media, and in order to identify regulators of immunosuppression. In a genetically engineered mouse model and an orthotopic model of prostate

cancer, the authors first confirm the skewing of tumor-infiltrating macrophages toward an immunosuppressive phenotype in the prostate cancer tumor microenvironment (TME). The results of the CRISPR/Cas9 screen identified previously known factors genes enriched in pro-tumorigenic macrophages compared to inflammatory populations, such as Stat6, supporting the relevance of the chosen screening approach. Genes involved in repolarizing otherwise pro-tumorigenic macrophages include chemosensors of the olfactory receptor (OLFR) and vomeronasal receptor (VMNR) families. The authors validated the top ranked candidates of these families, namely OLFR644 and VMN2R29, using genetic deletion in subcutaneous tumor transplant experiments, showing that loss of these genes conferred a pro-inflammatory phenotype in tumor-associated macrophages. The results further indicate that injection of OLFR644- or VMN2R29-deficient macrophages leads to a partial remodeling of the tumor microenvironment (TME) and to tumor stasis. Finally, using spatial lipidomics analyses, the authors suggest that tumor cell-derived palmitic acid is a plausible ligand driving the activation of OLFRs in prostate cancer.

This study applies a complex, genome-wide CRISPR genetic screening approach in primary murine immune cells, showcasing the value of unbiased screens to decipher macrophage plasticity *in vitro*. To date, CRISPR screening remains rarely applied to primary murine immune cells, particularly of myeloid origin, due to low success rates in generating a genetically diverse pool of transduced cells that can be sustained for the adequate amount of time required in such large-scale investigations. The results of this study point towards novel targets that may play important roles in macrophage polarization and response to tumor-derived signals. However, the design and set-up of the presented CRISPR screen are rather limited in terms of the characteristics chosen to define the two populations studied, and the rationale behind the choice of target genes is not clear. Although some of the identified targets are novel and potentially interesting as cancer immunology candidates regulating macrophage plasticity, the authors spend significant time and space in this manuscript repeating previously known findings, such as 1) the rather extensive description of the TME in prostate cancer^{1, 2}, 2) the potential of macrophages to acquire an immunosuppressive state in cancer³, 3) validating known regulators of inflammation; all of which limits the novelty of this study. In its current state, the study provides too limited *in vivo* and translational validation of the novel targets identified in the CRISPR/Cas9 screen and little functional bases for their mechanistic engagement in the TME of prostate cancers, as laid out in the comments below.

1. The authors show that the immune composition of *Ptenpc^{-/-};Trp53pc^{-/-}* tumors greatly differ from that of healthy prostate tissue. The described changes have previously been shown in *Pten/Trp53* prostate cancer models and the immune landscape of healthy prostates and prostate tumors has been demonstrated on a single-cell level in human patients^{1,2}. In light of this, the description of the TME should be only summarized in the manuscript.

2. The authors state that tumor-associated macrophages display a pro-tumorigenic phenotype as demonstrated by the expression of CD206 and Arginase1. There is mounting evidence that these markers only have limited value to properly define immunosuppressive and inflammatory macrophages, a complex qualification that has greatly been adapted recently⁴. Therefore, the authors should delve further into what characteristics these cells present using both complex panel of markers and functional assays.

Similarly, for the *in vitro* system, the characterization of conditioned media-educated macrophages and the evidence supporting their immunosuppressive functions are not sufficient. The authors should illustrate how the immunosuppressive phenotype and functions observed *in vitro* correspond to *in vivo* TAMs. For instance, using bulk or scRNAseq.

3. The set-up of the CRISPR screen is not sufficiently described and important details such as the number of cells and the coverage used are missing. Additionally, it is not clear which MOI was chosen for the transduction as the reported numbers do not correspond to each other between the figure and method section (Figure 2A and method section "GeCKO v2 Library B"). Additionally, the size of the sorted populations seems rather small and it is unclear whether the conditioned media-induced phenotypic shift was sufficient for appropriate comparison of the different groups to the control condition. It is unclear why the authors chose to compare the two sorted populations to each other over comparing them to an (input) control sample (e.g. control media). Moreover, the authors should clarify as to why only negative regulators are presented in this study (figure 2D), which could be interpreted as a shortcoming of the screen efficient identification of target genes.

4. Stat6 is a well-known player in driving an immunosuppressive phenotype in macrophages⁵. Therefore, the vast validation of genetic deletion of Stat6 in BMDMs and its effects on macrophage polarization is not novel and, therefore, can be summarized in the manuscript.

5. The CRISPR screen resulted in ample significant targets and the rationale of why the authors chose to validate specifically OLFR and VMNR targets is unclear. Additionally, it is not sufficiently convincing how specific the chosen targets are in light of the subsequent findings that genetic deletion of other members of the protein families can yield similar results (ext. data fig. 3C). A more refined screen design would help narrow down relevant targets. Alternatively, a smaller scale and more hypothesis-driven screen may be more relevant.

6. The *in vivo* validation studies were only conducted in a subcutaneous mouse model of prostate cancer. As subcutaneous models do not faithfully recapitulate the tumor kinetics and tumor microenvironment compared to other models, alternative approaches for *in vivo* validation should be undertaken. For instance, the intravenous infusion of macrophages chosen to validate OLFR and VMNR, which currently involves treating macrophages *ex vivo* prior to injecting them, should be complemented with either orthotopic injections of tumor cells and treated macrophages or using monocyte-specific knockout of these targets in prostate cancer GEMM or orthotopic tumors.

7. The authors show that pharmacological inhibition of Olfr644 *in vitro* with subsequent infusion into tumor-bearing mice results in significant reduction of tumor volume. However, no therapeutic intervention is performed using these inhibitors in

vivo, representing a major limitation with regards to therapeutic applicability. A pharmacological approach should be tested systemically and potential off-target effects should be evaluated.

8. It is unclear why the authors chose to validate OR51E2 in the human setting instead of the human homologue of Olfr644, especially as the authors describe that different OLFR and VMNR members possess divergent roles in the context of inflammation (e.g., Olfr229 and Vmn1R87 promote inflammation whereas Olfr192 promotes immunosuppression).

9. The evidence suggesting palmitic acid as a ligand for stimulating OR51E2 is not sufficiently elaborated upon and should be further validated. The underlying mechanism of palmitic acid driving an immunosuppressive phenotype is very interesting, but requires significantly more functional insights. Additionally, the spatial analysis of palmitic acid within the TME is underexplored. The authors should examine whether palmitic acid can be found in close proximity with macrophages causing their phenotypic rewiring in a quantitative manner, and use ex vivo/in vitro assays to examine the interplay between cells expressing/releasing palmitate and TAMs responding to it.

References

1. Mejía-Hernández, J.O. et al. Modelling aggressive prostate cancers of young men in immune-competent mice, driven by isogenic Trp53 alterations and Pten loss. *Cell Death & Disease* 13, 777 (2022).
2. Tuong, Z.K. et al. Resolving the immune landscape of human prostate at a single-cell level in health and cancer. *Cell Reports* 37, 110132 (2021).
3. de Visser, K.E. & Joyce, J.A. The evolving tumor microenvironment: From cancer initiation to metastatic outgrowth. *Cancer Cell* 41, 374-403 (2023).
4. Bill, R. et al. *CXCL9:SPP1* macrophage polarity identifies a network of cellular programs that control human cancers. *Science* 381, 515-524 (2023).
5. Czimmerer, Z. et al. The Transcription Factor STAT6 Mediates Direct Repression of Inflammatory Enhancers and Limits Activation of Alternatively Polarized Macrophages. *Immunity* 48, 75-90.e76 (2018).

Reviewer #2 (Remarks to the Author):

The finding that olfactory receptors may be important for regulating the TAM phenotype is novel and exciting. However, several important methodological concerns are formulated below:

- 1) Several remarks on Figures 1B-G.
 - the authors take a shortcut by simply stating that the results in the orthotopic model are 'similar'. However, there are clear differences at the level of CD8⁺ T cells and TAMs
 - Fig 1C tumour: how can the sum of the % of both TAM populations be more than 100%? Eg one of the datapoints for the CD206⁺ populations is around 75%. One would expect a counterpart for CD206⁻ of around 25%. But all datapoints for CD206⁻ are above 40%. That is simply impossible if the datapoints come from the same mice (as it should), suggesting that these data are not derived from the same experimental mice, which is not correct. Please explain.
 - Conversely, the sum of CD206⁺ and CD206⁻ TAMs is sometimes very low, eg in Fig 1F non-tumour, but to some extent also tumour. What are the other F4/80⁺ populations?
- 2) Fig 1O-P makes use of 'TAM' supernatant, which actually is supernatant of BMDM treated with cancer cell CM. To what extent is the cancer cell CM still present in the 'TAM' supernatant? These assays would be much more informative with bona fide TAMs, isolated from tumors versus healthy tissue macrophages
- 3) Fig 2F. The lack of CD206 expression rather seems to be due to a rise in the background staining (isotype control? This is not indicated in the legend), but not really a decrease in the CD206-specific staining. The data shown can not exclude the possibility that the STAT6 downregulation rather upregulates a specific FcR binding, reducing the % positivity for the markers under study.
- 4) The intravenous transfer of BMDM is a very artificial system that does not recapitulate the in vivo situation during tumor growth. Mature macrophages are never in circulation, monocytes are. This may alter important aspects, such as extravasation and tumor penetration, intratumoral localization etc
- 5) The GFP signal appears like a continuum, rather than a clear discriminatory marker to discriminate endogenous and transferred macrophages. Hence, based on Figure 4C, it is impossible to say whether the authors really gated on transferred macrophages, or whether transferred macrophages are even there. A more discriminative approach is needed.
- 6) Fig 4D-E. The changes in the immune infiltrate (at least CD8⁺ and Treg) are rather modest to explain such clear differences in tumor growth. Are there effects on vessel density and maturation, extracellular matrix,...? The importance of

anti-tumor immunity should be assessed via the specific depletion of immune cell populations

7) Retinoic acids have a multitude of effects on macrophages, which likely go beyond the inhibition of one olfactory receptor. The effects of the KO in Fig 4I are indeed partial, leaving room for interpretation of what RA's other effects are.

8) THP-1-derived macrophages are not fully representative for human primary macrophages. The use of monocyte-derived macrophages is preferred.

9) The role of palmitic acid could be much more elaborated on, in an in vivo context. Is this also a ligand for the mouse orthologue? If so, mouse systems could be used to go much deeper into the role of this lipid in vivo

Reviewer #3 (Remarks to the Author):

The manuscript elucidates the unexpected role of chemosensory receptors in macrophage infiltration in cancer and the potential of re-educating tumor-associated macrophages (TAMs) for immunotherapy. Through genome-wide CRISPR screening, several olfactory and vomeronasal receptors were identified as major drivers of TAMs' tumor-supportive status in prostate cancers. Deletion of one olfactory or one vomeronasal chemoreceptor led to cancer regression and increased infiltration in vivo. Additionally, palmitic acid was found to enhance their pro-tumor phenotype in human prostate cancer tissues, and this effect requires OR51E2. These findings suggest chemoreceptors as new potential therapeutic targets for enhancing antitumor immunity. The work is potentially groundbreaking in the fields of cancer biology, immunology, and chemical senses. However, there are several reservations that the authors need to address:

1. In Figure 2, the authors show that sgRNAs for several chemosensory receptors were significantly over- or under-represented in the in vitro CRISPR screen. There is a need to demonstrate the specificity of chemosensory receptors involved in TAMs. The authors chose one OR and one Vmn2r as representatives, but later show that Or51e2 also has a similar role. Where is Olfr78 (Or51e2) CRISPR sgRNA in the volcano plot Fig 2D? As the manuscript stands, it is unclear how many chemosensory receptors are indeed involved. As there are many chemosensory receptors, it would not be feasible to test them all. However, the authors should provide examples of ORs and Vmn2rs that do not show similar effects.
2. In chemosensory organs, ORs and Vmn2rs couple with different G proteins and their canonical downstream signaling pathways are very different. However, the authors' data appear to show these receptors having the same role in TAMs. The authors should address these apparent discrepancies. Additionally, no functional Vmn2rs is present in humans, which the authors should discuss.
3. The three chemoreceptors are likely to bind distinct ligands, yet the role of these and potentially additional chemoreceptors in functioning TAMs appears to be equivalent. Yet, in the section involving Olfr78/Or51e2, the authors appear to propose a novel ligand. The author should clarify this.
4. The authors should note that OR51E2 and OR51E1 (they are called PSGRs) have been previously reported to be over-represented in prostate cancers.
5. Propionate and acetate are well-established ligands for OR51E2, whereas medium-chain fatty acids do not activate OR51E2. Therefore, palmitic acid as a novel ligand for OR51E2 is somewhat unexpected. The authors should demonstrate palmitic acid-mediated OR51E2 activation, perhaps in the heterologous system. In addition, the authors should use propionate or acetate as a benchmark in their experiments characterizing the role of OR51E2.
6. The nomenclature of ORs in humans vs. mice is confusing. The use of a human-centric nomenclature system, for example, Or51e2 instead of Olfr78, is recommended to minimize confusion.

Version 1:

Decision Letter:

Dear Dr Di Mitri,

Thank you for your letter asking us to reconsider our decision on your manuscript, "CRISPR Genome-Wide screening reveals chemosensor receptors as lipid-detecting regulators of macrophage functions in cancer".

Now that I have had a chance to discuss the matter carefully with my colleagues, I am happy to say that we would consider sending your manuscript back to external review. I'm sure, however, that you'll understand that we cannot predict the outcome of the review process.

However, we cannot send the present version to the reviewers as you have not uploaded the figures. A link to your figures is not sufficient.

Once you have made these revisions, please use the URL below to submit the revised manuscript with figures and a revised version of the life sciences reporting summary. It will be available to referees (and, potentially, statisticians) to aid in their evaluation if the manuscript goes back for peer review. A revised checklist is essential for re-review of the paper.

The Reporting Summary can be found here:

The Editorial Policy Checklist can be found here: <https://www.nature.com/documents/nr-editorial-policy-checklist.pdf>

Link Redacted

Please let us know how you wish to proceed and when we can expect your revised manuscript.

With kind regards,

Nick Bernard, PhD
Senior Editor
Nature Immunology

Version 2:

Decision Letter:

Our ref: NI-A37606B

10th Apr 2025

Dear Dr. Di Mitri,

Thank you for submitting your revised manuscript "CRISPR Genome-Wide screening reveals chemosensor receptors as lipid-detecting regulators of macrophage functions in cancer" (NI-A37606B). It has now been seen by the original referees 1 and 2. Reviewer 3 was not available to re-review the paper. Therefore we enrolled a new reviewer to comment on your response to reviewer 3 and to mediate over the discussion between yourself and reviewer 1 regarding the application of the CRISPR screen.

As you will see below, reviewer 4 sides with the authors on this matter and is satisfied by the response to reviewer 3 and therefore we'll be happy in principle to publish it in Nature Immunology, pending minor revisions to satisfy some minor comments from reviewer 4 and to comply with our editorial and formatting guidelines.

We will now perform detailed checks on your paper and will send you a checklist detailing our editorial and formatting requirements in about a week. Please do not upload the final materials and make any revisions until you receive this additional information from us.

If you had not uploaded a Word file for the current version of the manuscript, we will need one before beginning the editing process; please email that to immunology@us.nature.com at your earliest convenience.

Thank you again for your interest in Nature Immunology Please do not hesitate to contact me if you have any questions.

Sincerely,

Nick Bernard, PhD
Senior Editor
Nature Immunology

Reviewer #1 (Remarks to the Author):

The authors significantly improved the manuscript following the first round of revisions particularly strengthening their claims with the generation of additional data sets using (sc)RNAseq and spatial transcriptomic approaches. The authors strengthened the rationale of their study by expanding on the characterization of macrophage heterogeneity in mouse models of prostate cancer. Additionally, this allowed the authors to reinforce the relevance of the described TAM phenotypes in vitro that were used as a readout for the genetic screening. In the revised version, the in vivo validation of targets from the genetic screen is particularly improved, confirming the significance of the findings in this manuscript and their potential therapeutic impact.

However, the main limitation of the manuscript remains the design of the genetic screen that lacks guidance by a clear hypothesis, resulting in a rather broad and untargeted, lacking controlled approach. This may have complicated the choice of targets, potentially undermining the significance of the results and making it difficult to justify the choice of employing this approach.

This is also demonstrated by the lack of specificity of the selected target genes. An example that highlights this shortcoming is the identification of relevant olfactory receptors in human prostate cancer where the authors show that OR51E2 is highly upregulated with OLF78 being the corresponding mouse orthologue. However, OLF78 was not identified or chosen as one of the positive regulators in the genetic screen. This also resulted in the human cellular experiments being conducted

based on OR51E2 deletion which, technically, was not a chosen target from the screen. Therefore, it is unclear how the choice of validated targets was guided, and it raises the questions regarding whether more relevant targets could also be identified using a more simplified approach.

Nevertheless, the authors have significantly enriched the revised manuscript with valuable data and insights into macrophage functionality in prostate cancer. Additionally, the authors demonstrate strong validation in mouse models and human cellular systems, which is a clear strength of the manuscript. Yet, the technical execution and design of the CRISPR screen as well as the choice of targets could have been refined and revisited, as suggested in the first round of reviews, ultimately leaving some flaws in this study unaddressed.

Reviewer #2 (Remarks to the Author):

The authors sufficiently addressed my concerns

Reviewer #4 (Remarks to the Author):

Following review of the paper and both the reviewer comments and the authors rebuttal I am in favor of accepting the manuscript. The basis of the paper is taking an unbiased approach to identify new factors associated with tumor associated macrophage phenotypes and therefore their screen as presented makes sense. While it is unfortunate that OLF78 did not emerge as a top hit they did go on to show that individual removal of this did impact the phenotype and therefore there are a number of reasons it might not have emerged in the screen (for example the sgRNAs did not target it efficiently). Their subsequent screen hits such as chemosensory receptors OR51E2 are followed up and validated which argues for the power of the screening approach to identify novel regulators.

Responses to reviewer 3 comments are satisfactory.

In Figure 7 they use siRNA in primary cells. But in their legend they use the term "partial genetic deletion". This is not accurate and should be described as RNA knockdown.

One small additional point is in Figure 3 F they show a venn diagram. It would be preferable to have this be correctly sized to reflect the numbers. This would be more accurate and in keeping with the highly technical figures.

Revision Nature Immunology NI-A37606

Title: A genome-wide CRISPR screening reveals chemosensor receptors as lipid-detecting regulators of macrophage functions in cancer

To the Editor of Nature Immunology and the referees

We appreciated the referees' comments and feedback, which has helped us to strongly improve the manuscript. We have carefully addressed most of the points raised and made extensive revisions to enhance clarity, robustness, and overall scientific quality. We believe these changes have strengthened the manuscript and made it more comprehensive.

All modifications in the revised version are highlighted in yellow. Additionally, we have added the following panels to the original figures: Fig.1B-E, Fig.1H, Fig.1I, Fig.1N; Fig.4F-L; Fig.5H-J; Fig.7B-H; Fig.8B-J; Extended Data1H-I; Extended Data2J; Extended Data3A, Extended Data3H; Extended Data4D-G; Extended Data6J-L; Extended Data7E-J; Extended Data8E; Extended Data9A-I.

We also included two figures in this rebuttal showing graphs that were not part of the final manuscript.

Please find below our point-by-point response.

Referee 1

1. The authors show that the immune composition of *Pten^{pc}-/-*; *Trp53^{pc}-/-* tumors greatly differ from that of healthy prostate tissue. The described changes have previously been shown in *Pten/Trp53* prostate cancer models and the immune landscape of healthy prostates and prostate tumors has been demonstrated on a single-cell level in human patients^{1,2}. In light of this, the description of the TME should be only summarized in the manuscript.

Response:

We appreciate this suggestion and have revised the manuscript to provide a more concise summary of the immune composition while emphasizing the novelty of our findings. To add novelty and strengthen our analysis, we implemented scRNAseq profiling of the tumor microenvironment (TME) in *Pten^{pc}-/-*; *Trp53^{pc}-/-* transgenic mice, allowing us to refine the characterization of tumor-associated macrophages (TAMs) (Fig. 1C-E and Extended data H-I). By reclustering macrophages, we gained deeper insights into their heterogeneity and distinct functional states, adding a critical level of analysis on TAMs. As now reported in the manuscript, we re-clustered myeloid cells according to CellTypist annotation (Extended Data 1H). The resulting macrophage clusters were characterized based on their transcriptional profiles, drawing on phenotypes recently described²². Our analysis revealed distinct macrophage subpopulations, including proliferative C18 monocytes-macrophages (Mono-Macs), C14-5-7 lipid-laden macrophages (LA-Macs), C12 angiogenic TAMs (Angio Macs), C16 regulatory macrophages (Reg-Macs) and C11 resident-tissue macrophages (RTM-Macs) that express heat-shock proteins (previously described by Caronni et al.²² as exhausted cells). Other clusters were defined by C13 interferon-related genes (INF Macs), or by C10 Il-1 β expression as inflammatory macrophages (Inflam Macs) (Fig. 1D and Extended Data 1I). Trajectory inference identified a pseudotime progression, indicative of cell maturation and activation, starting from monocytes (C18) (Fig. 1E).

In addition, we now better highlighted the comparison between the *Pten^{pc}-/-*; *Trp53^{pc}-/-* genetic model and the orthotopic prostate cancer model, which has not been characterized in detail before. Notably, we observed specific immune differences between these models, that we now discussed. Our results indicate that the orthotopic model partially recapitulates the transgenic model, as macrophages in both systems share key phenotypic and functional similarities. We believe this

aspect adds novelty to our study, as it provides further validation of macrophage behavior in an additional tumor model. The revised text now incorporates these points.

2. The authors state that tumor-associated macrophages display a pro-tumorigenic phenotype as demonstrated by the expression of CD206 and Arginase1. There is mounting evidence that these markers only have limited value to properly define immunosuppressive and inflammatory macrophages, a complex qualification that has greatly been adapted recently 4. Therefore, the authors should delve further into what characteristics these cells present using both complex panel of markers and functional assays. Similarly, for the in vitro system, the characterization of conditioned media-educated macrophages and the evidence supporting their immunosuppressive functions are not sufficient. The authors should illustrate how the immunosuppressive phenotype and functions observed in vitro correspond to in vivo TAMs. For instance, using bulk or scRNAseq.

Response:

We appreciate the reviewer's comments and we agree on the need to properly define the features of TAMs. As a consequence, we have now expanded our characterization of TAMs to provide a more robust and detailed analysis. To functionally prove that TAMs are immunosuppressive, we sorted macrophages from the PCa model and tested in vitro their suppressive capacity against CD8+ T cells. Our results confirm that macrophages sorted from the tumors are able to suppress CD8+ T cell proliferation, thus recapitulating the *in vitro* findings (Extended Fig. 2J). In addition, as mentioned above, we implemented scRNAseq to profile *ex vivo* macrophage diversity in our *in vivo* PCa model. Also, to better dissect macrophage phenotype and function in vitro, we have incorporated additional macrophage polarization markers to refine by Flow cytometry the characterization of macrophages conditioned *in vitro* with tumor media (Fig. 1N). We then performed bulk RNA sequencing on *in vitro* tumor-conditioned macrophages, as suggested by the referee, which allowed us to uncover distinct transcriptional profiles (Fig. 1H-I). In accordance with our hypothesis, genes associated with macrophage immunosuppressive function, such as Arginase 1, Trem1 and Thbs1 (ref: PMID: 12496409, PMID: 37749092, PMID: 37651197), were upregulated in *in vitro* tumor-conditioned macrophages when compared to control (Fig. 1H-I). Importantly, the comparison between *in vivo* TAM populations and *in vitro*-conditioned macrophages showed strong similarities, thus reinforcing the relevance of our *in vitro* system in recapitulating the in vivo macrophage phenotype.

We think that these improvements significantly strengthen the characterization of TAMs. The revised manuscript now reflects these additions.

3. The set-up of the CRISPR screen is not sufficiently described and important details such as the number of cells and the coverage used are missing. Additionally, it is not clear which MOI was chosen for the transduction as the reported numbers do not correspond to each other between the figure and method section (Figure 2A and method section "GeCKO v2 Library B"). Additionally, the size of the sorted populations seems rather small and it is unclear whether the conditioned media-induced phenotypic shift was sufficient for appropriate comparison of the different groups to the control condition. It is unclear why the authors chose to compare the two sorted populations to each other over comparing them to an (input) control sample (e.g. control media). Moreover, the authors should clarify as to why only negative regulators are presented in this study (figure 2D), which could be interpreted as a shortcoming of the screen efficient identification of target genes.

Response:

We apologize for the missing and confounding information. We have revised the manuscript to clarify the methodological details of the CRISPR screen. We now explicitly state the number of cells used, the library coverage, and the chosen MOI (again we apologize for the mistake in the previous version) for transduction to ensure consistency between the methods section and the figure. These additions allow for a clearer interpretation of the experimental setup. Regarding the sorting strategy, we compared CD206^{bright}MHCII⁻ and CD206⁻MHCII⁺ macrophages to capture the two extreme functional states within the TAM spectrum—pro-tumoral and pro-inflammatory. This choice was driven by our aim to identify key regulators of macrophage polarization rather than assessing transcriptomic shifts between conditioned and unconditioned cells. While we acknowledge that an input control (such as control media-treated macrophages) could offer additional insights, our primary focus was to pinpoint genes differentially expressed between these two functionally distinct TAM populations. Rationale behind our choice is that there is broad scientific consensus that reprogramming tumor-associated macrophages (TAMs) from a pro-tumoral to an anti-tumoral/pro-inflammatory state is a more effective therapeutic approach than their depletion. Based on this rationale, we reasoned that comparing pro-tumoral CD206^{bright}MHCII⁻ versus anti-tumoral CD206⁻MHCII⁺ macrophages would allow us to identify key regulators that, when modulated, could induce a functional switch between these polarization states. Importantly, in our screening approach we actually performed all possible comparisons (here listed: 1- CD206^{bright}MHCII⁻ versus CD206⁻MHCII⁺; 2- CD206^{bright}MHCII⁻ versus CD206^{mid}MHCII⁻; 3- CD206^{bright}MHCII⁻ versus CD206⁻MHCII⁻; 4- CD206^{mid}MHCII⁻ versus CD206^{neg}MHCII⁻; 5- CD206^{neg}MHCII⁻ versus CD206⁻MHCII⁺; 6- CD206^{mid}MHCII⁻ versus CD206⁻MHCII⁺; also see Figure 2C of this rebuttal and Excel file 1) and we applied the sequencing strategy across all conditions. However, to maintain clarity and avoid overloading the manuscript with excessive data, we initially chose not to include these additional results. In response to the referee's request, we now provide in this rebuttal the complete results from all comparisons, including positive regulators, in Excel file 1. If the referee believes these data should be included in the main manuscript, we would be happy to do so. Meanwhile, we have incorporated in the revised manuscript a table showing the positive regulators identified in our screen for the comparison CD206^{bright}MHCII⁻ versus CD206⁻MHCII⁺ (Figure 2D and Table 2).

About numbers of cells, we have now provided all details in the manuscript. We maintained sufficient coverage (>500 cells for each sgRNA), as advised by Joung J et al (PMID: 28333914). The library is composed by 62,804 sgRNAs and after lentiviral infection we yielded the following number of infected and selected cells: 1) Experiment 1= 40x10⁶ cells, 2) Experiment 2= 35x10⁶ cells. Infected cells were then exposed to tumor conditioned media and sorted as reported in the manuscript. Number of sorted cells are the following (CD206⁺MHCII⁺ cells were discarded):

Exp1)

CD206^{low}MHCII⁻: 1.2x 10⁶cells

CD206^{mid}MHCII⁻: 7.2x10⁶ cells

CD206^{bright}MHCII⁻: 7.8x10⁶ cells

CD206⁻MHCII⁺: 500.000 cells

Exp2):

CD206^{low}MHCII⁻:961.557 cells

CD206^{mid}MHCII⁻: 5.668.239 cells

CD206^{bright}MHCII⁻: 5.977.499 cells

CD206⁻MHCII⁺: 177.224 cells

In addition to what discussed above, we believe that the independent validation of some of the targets, shown in the manuscript, confirmed the functional relevance of these targets in driving macrophage polarization.

4. Stat6 is a well-known player in driving an immunosuppressive phenotype in macrophages⁵. Therefore, the vast validation of genetic deletion of Stat6 in BMDMs and its effects on macrophage polarization is not novel and, therefore, can be summarized in the manuscript.

Response:

We now provided a more concise summary of the role of Stat6 deletion in macrophages.

5. The CRISPR screen resulted in ample significant targets and the rationale of why the authors chose to validate specifically OLFR and VMNR targets is unclear. Additionally, it is not sufficiently convincing how specific the chosen targets are in light of the subsequent findings that genetic deletion of other members of the protein families can yield similar results (ext. data fig. 3C). A more refined screen design would help narrow down relevant targets. Alternatively, a smaller scale and more hypothesis-driven screen may be more relevant.

Response:

We appreciate the referee's comments regarding target selection and specificity. While we acknowledge that our screen identified multiple potential modulators of macrophage polarization, it was not feasible for us to perform an additional refined screening due to the financial and logistical constraints associated with the CRISPR screening approach. In the manuscript, we prioritized a subset of genes for further validation based on both their biological relevance and novelty. Our choice to focus on chemosensor genes (OLFR and VMNR families) was guided by two key observations. First, these genes were highly represented among both negative and positive regulators, suggesting a functional role in macrophage biology. We have now included a Volcano plot showing all significant genes belonging to the chemosensor family from our screening (Extended Data 3H). Second, chemosensors have not been extensively studied in the context of tumor-associated macrophages, making this an unexplored avenue for understanding macrophage-tumor interactions and for identifying new therapeutic strategies to modulate macrophages in the tumor microenvironment.

Regarding the specificity of the chosen targets, we observed a degree of redundancy, which is not entirely unexpected given that chemosensor genes often exhibit overlapping or complementary functions. Indeed, a key message emerging from our screening strategy is that multiple chemosensory receptors contribute to macrophage activation, with some acting as positive regulators and others as negative regulators. We believe that macrophages may exploit the chemosensing to interpret the complexity of the tumor microenvironment.

Our screening identified OLFR644 and VMN2R29 as two of the most highly ranked and functionally relevant hits, which is why we prioritized them for further validation. To better understand the redundancy in chemosensor activity, we have now analyzed potential transcriptional pathways downstream of OLFR644 and VMN2R29, identifying shared signaling components that may account for their similar effects on macrophage polarization (Extended Data 4F-G). Our data suggest that in tumor-associated macrophages (TAMs), chemosensory receptors may converge on common regulatory mechanisms. To investigate the signaling pathways downstream of these receptors, we performed bulk mRNA sequencing of macrophages exposed to tumor-conditioned media and compared their transcriptional profiles following genetic deletion of OLFR644 and VMN2R29. Using Ingenuity Pathway Analysis, we identified key transcriptional regulators associated with the differentially regulated pathways. Our analysis revealed that transcription

factors such as HIF1 α and cMYC were inactivated or suppressed in both OLF644^{-/-} and VMN2R29^{-/-} macrophages (Extended Data 4F-G). Notably, HIF1 α and cMYC have been previously described as key modulators of pro-tumoral macrophage activation (PMID: 20841473 and PMID: 22067385). To further validate this, we performed qRT-PCR to assess the expression of known HIF1 α target genes. Our preliminary data indicate that HIF1 α signalling is indeed downregulated in the absence of these chemosensory receptors. These findings suggest the existence of common downstream pathways regulated by OLF644 and VMN2R29, despite their distinct canonical signalling mechanisms in chemosensory organs. While these data provide an initial indication of shared molecular circuits, further investigation will be necessary to fully elucidate the interplay between chemosensory receptors and transcriptional regulators in TAMs. Further studies will be necessary to explore these pathways and clarify the mechanistic basis of this redundancy.

6. The in vivo validation studies were only conducted in a subcutaneous mouse model of prostate cancer. As subcutaneous models do not faithfully recapitulate the tumor kinetics and tumor microenvironment compared to other models, alternative approaches for in vivo validation should be undertaken. For instance, the intravenous infusion of macrophages chosen to validate OLF644 and VMN2R29, which currently involves treating macrophages ex vivo prior to injecting them, should be complemented with either orthotopic injections of tumor cells and treated macrophages or using monocyte-specific knockout of these targets in prostate cancer GEMM or orthotopic tumors.

Response:

We appreciate the referee's suggestion and we agree on the importance of validating our findings in an additional prostate cancer model. To address this point, we have now followed the referee suggestions and we performed an orthotopic co-injection experiment, in which prostate cancer tumor cells were injected directly into the prostate together with macrophages, either wild-type (LGP) or knockout (KO) for chemosensor genes (OLF644 or VMN2R29). Our results confirmed the key role of chemosensor-expressing macrophages in supporting tumor growth. Specifically, when KO macrophages were co-injected with tumor cells, tumor growth was significantly reduced compared to the condition in which WT macrophages were present. Additionally, the tumor microenvironment was reshaped, with a notable increase in CD8⁺ T cell infiltration and decrease of CD4⁺ T regulatory cells (Fig. 4F-I). We believe that these findings reinforce our previous observations in the subcutaneous model and provide additional evidence that chemosensor-expressing macrophages contribute to tumor progression in an orthotopic setting. The revised manuscript now includes these results.

7. The authors show that pharmacological inhibition of OLF644 in vitro with subsequent infusion into tumor-bearing mice results in significant reduction of tumor volume. However, no therapeutic intervention is performed using these inhibitors in vivo, representing a major limitation with regards to therapeutic applicability. A pharmacological approach should be tested systemically and potential off-target effects should be evaluated.

Response:

We thank the referee for this comment regarding the need for systemic pharmacological validation. To address this, we have now performed a systemic administration experiment, in which tumor-bearing mice were treated with All-trans retinoic acid (13-cRA) via oral gavage injection (Fig. 1A-

B of this rebuttal). This preclinical study confirmed that systemic inhibition of OLF644 led to a significant reduction in tumor growth, however with only a partial reshaping of the tumor microenvironment (TME). Interestingly, the concomitant administration of 13-cRA and infusion of OLF644^{-/-} macrophages yielded similar results to 13-cRA alone, with no synergism, suggesting that this molecule acts at least partially through the OLF644. If the referee thinks it is useful, we can add these data to the manuscript. Please note that to further investigate the impact of 13-cRA on the OLF644 and on macrophages, we performed an experiment where OLF644 WT (LGP) macrophages and OLF644^{-/-} (KO) macrophages were exposed to 13-cRA and infused in tumor bearing mice. These arms were compared to mice infused with WT or OLF644^{-/-} (KO) untreated macrophages. No difference was observed between KO macrophages alone and KO macrophages treated with the 13-cRA, indicating that 13-cRA acts at least partially through the chemosensor receptor. The revised manuscript now includes these results (Fig. 5H-J). We believe that these findings strengthen the therapeutic relevance of our study and demonstrate that systemic administration of this inhibitor has direct anti-tumor effects in vivo.

8. It is unclear why the authors chose to validate OR51E2 in the human setting instead of the human homologue of OLF644, especially as the authors describe that different OLF and VMNR members possess divergent roles in the context of inflammation (e.g., Olfr229 and Vmn1R87 promote inflammation whereas Olfr192 promotes immunosuppression).

Response:

We thank the referee for this comment, which gives us the opportunity to clarify this point. The primary reason for the selection for the OR51E2 was the lack of a human homologue for OLF644. We were indeed unable to identify a direct human ortholog of this murine receptor. Instead, we prioritized OR51E2 because it is one of the most differentially expressed olfactory receptors in human prostate cancer compared to healthy tissue, as indicated by our analyses of publicly available datasets (TCGA). Given its strong modulation in the human context, we considered OR51E2 to be the most relevant candidate for further investigation. Specifically, we thought it would be important to identify a key ligand for human OR51E2, which is notably overexpressed in human prostate cancer samples. This led us to dedicate significant effort to detecting palmitic acid and investigating its role in human cancer, as we believe this could offer important insights into tumor biology and potential therapeutic impact. The revised manuscript clarifies this rationale in the discussion section.

9. The evidence suggesting palmitic acid as a ligand for stimulating OR51E2 is not sufficiently elaborated upon and should be further validated. The underlying mechanism of palmitic acid driving an immunosuppressive phenotype is very interesting, but requires significantly more functional insights. Additionally, the spatial analysis of palmitic acid within the TME is underexplored. The authors should examine whether palmitic acid can be found in close proximity with macrophages causing their phenotypic rewiring in a quantitative manner, and use ex vivo/in vitro assays to examine the interplay between cells expressing/releasing palmitate and TAMs responding to it.

Response:

We appreciate the referee's comments and have conducted additional analyses to strengthen our findings. To investigate the biological consequences of macrophage exposure to palmitic acid, we performed bulk mRNA sequencing on palmitic acid-treated macrophages (Extended Data 9A-B).

This analysis revealed a significant downregulation of pathways related to antigen presentation and immune functions, while pathways associated with RNA metabolism were upregulated. Notably, genes linked to macrophage pro-inflammatory activity, such as TLR8, were downregulated following palmitic acid treatment (Extended Data 9B). These findings suggest that palmitic acid suppresses pro-inflammatory macrophage functions while promoting their reprogramming. While these results reinforce the role of palmitic acid in shaping macrophage behaviour, further studies will be necessary to fully elucidate the molecular mechanisms underlying these effects.

To address the spatial distribution of palmitic acid within the TME, we performed Hyperion imaging mass cytometry (IMC) to map the localization of macrophages and other immune cells in tumor sections consecutive to those used for spatial lipidomics analysis of palmitic acid. These data were integrated with H&E staining and immunofluorescence, allowing us to correlate palmitic acid localization with immune cell distribution in the tumor microenvironment. Our analysis revealed an increased abundance of most cell types in tumor areas, including conventional T cells (T conv), regulatory T cells (Treg), and both pro-inflammatory (CD206-MHCII+CD68+ M1-like and CD206+MHCII-CD68+ M2-like macrophages) (Extended data 9G). Notably, spatial interaction analysis demonstrated a higher propensity for PanCyK+ epithelial cancer cells to interact with M2-like macrophages in palmitic-rich tumor regions compared to PanCyK+ epithelial cells in the palmitic deprived areas (Fig. 8D-E). Additionally, we performed Visium spatial transcriptomics on the same tumor sections to explore macrophage heterogeneity and identify transcriptional profiles associated with different macrophage subsets (Fig. 8F-J). This analysis provided a deeper understanding of how macrophage populations are spatially organized in relation to lipid accumulation and tumor architecture. Unbiased clustering analysis identified four distinct tissue areas, corresponding to palmitic-deprived non-tumoral tissues (Non-Tumor_1 and Non-Tumor_2) and palmitic-enriched tumoral tissues (Tumor_1 and Tumor_2) (Fig. 8F-G Extended Data 9H). Analysis of gene expression revealed an enrichment of the LA-TAMs and Angio-TAMs signatures within palmitic-enriched regions (Fig. 8H-J and Extended Data 9I), indicative of a pro-tumoral phenotype of TAMs in these areas, thus supporting the role of palmitic acid in shaping macrophage function in tumors.

We believe that these additional experiments provide strong evidence that palmitic acid is present in close proximity to tumor-associated macrophages and may thus contribute to their phenotypic reprogramming. The revised manuscript now includes these new findings.

Referee 2

The finding that olfactory receptors may be important for regulating the TAM phenotype is novel and exciting. However, several important methodological concerns are formulated below:

- 1) **Several remarks on Figures 1B-G.**
 - **the authors take a shortcut by simply stating that the results in the orthotopic model are 'similar'. However, there are clear differences at the level of CD8+ T cells and TAMs**
 - **Fig 1C tumour: how can the sum of the % of both TAM populations be more than 100%? Eg one of the datapoints for the CD206+ populations is around 75%. One would expect a counterpart for CD206- of around 25%. But all datapoints for CD206- are above 40%. That is simply impossible if the datapoints come from the same mice (as it should), suggesting that these data are not derived from the same experimental mice, which is not correct. Please explain.**
 - **Conversely, the sum of CD206+ and CD206- TAMs is sometimes very low, eg in Fig 1F non-tumour, but to some extent also tumour. What are the other F4/80+ populations?**

Response:

We thank the referee for these observations and we have now carefully re-evaluated our data to ensure accuracy and clarity. First, regarding the comparison between the genetically engineered model and the orthotopic model, we agree that while some immune features are shared, key differences exist—particularly in CD8⁺ T cell infiltration and TAM composition. We have now revised the text to explicitly describe these differences. Second, we acknowledge the mistake in the flow cytometry analysis that led to percentages exceeding 100%, we apologize for our mistake. This occurred because we unintentionally included MHCII expression in both the CD206⁺ and CD206⁻ populations, leading to an overestimation of certain fractions. We have now re-analyzed the flow cytometry data to correct this issue and ensure internal consistency, and the revised figures accurately reflect the true proportions of each population. Regarding the low sum of CD206⁺ and CD206⁻ TAMs in Figure 1F, we have further examined the F4/80⁺ compartment. The missing fraction corresponds to CD206⁻MHCII⁻ macrophages, which can be considered a "resting" population. However, to better characterize the macrophage subsets, we have now incorporated a scRNAseq analysis, which provides a more refined classification of macrophage heterogeneity and allows us to define additional macrophage states that were not fully captured by conventional flow cytometry gating. As reported in the revised manuscript, we implemented scRNAseq profiling of the tumor microenvironment (TME) in Pten^{pc-/-}; Trp53^{pc-/-} transgenic mice, allowing us to refine the characterization of tumor-associated macrophages (TAMs) (Fig. 1C-E and Extended data H-I). By reclustering macrophages, we gained deeper insights into their heterogeneity and distinct functional states, adding a critical level of analysis on TAMs beyond what has been previously reported. As now reported in the manuscript, we re-clustered myeloid cells according to CellTypist annotation (Extended Data 1H). The resulting macrophage clusters were characterized based on their transcriptional profiles, drawing on phenotypes recently described²². Our analysis revealed distinct macrophage subpopulations, including proliferative C18 monocytes-macrophages (Mono-Macs), C14-5-7 lipid-laden macrophages (LA-Macs), C12 angiogenic TAMs (Angio Macs), C16 regulatory macrophages (Reg-Macs) and C11 macrophages expressing heat-shock proteins (previously described by Caronni et al.²² as exhausted cells). Other clusters were defined by C13 interferon-related genes (INF Macs), or by C10 Il-1 β expression as inflammatory macrophages (Inflam Macs) (Fig. 1D and Extended Data 1I). Trajectory inference identified a pseudotime progression, indicative of cell maturation and activation (Fig. 1E).

We think that these revisions enhance the accuracy of our dataset and provide a clearer interpretation of macrophage diversity within the tumor microenvironment. The corrected figures and additional transcriptomic data are now included in the revised manuscript.

2) Fig 1O-P makes use of 'TAM' supernatant, which actually is supernatant of BMDM treated with cancer cell CM. To what extent is the cancer cell CM still present in the 'TAM' supernatant? These assays would be much more informative with bona fide TAMs, isolated from tumors versus healthy tissue macrophages.

Response:

We thank the referee for this point and we apologize for lack of clarity. We appreciate the referee's concern and would like to clarify that before collecting the TAM-conditioned supernatant, macrophages were thoroughly washed after 48 hours of exposure to cancer cell-conditioned medium (CM). This step ensures that the remaining supernatant used in our functional assays does not contain residual tumor-derived factors and exclusively reflects macrophage-secreted molecules. Also, we agree that using primary TAMs isolated from tumors would provide stronger physiological relevance. To address this point, we have now performed an additional experiment using supernatants from TAMs directly isolated from tumors and compared them to macrophages

isolated from healthy tissues. In details, to functionally prove that TAMs are immunosuppressive, we sorted macrophages from the PCa model and tested their suppressive capacity *in vitro* against CD8⁺ T cells. Our results confirm that macrophages sorted from the tumors effectively suppress CD8⁺ T cell proliferation, consistent with the *in vitro* findings (Extended Fig. 2J). As a control, we used peritoneal macrophages, since attempts to sort macrophages from healthy prostate tissue were unsuccessful due to their low abundance.

3) Fig 2F. The lack of CD206 expression rather seems to be due to a rise in the background staining (isotype control? This is not indicated in the legend), but not really a decrease in the CD206-specific staining. The data shown can not exclude the possibility that the STAT6 downregulation rather upregulates aspecific FcR binding, reducing the % positivity for the markers under study.

Response:

To address this point, we repeated the *in vitro* experiment using STAT6 KO cells and included an isotype control (anti-IgG2a, k). The results were consistent with the previous findings, and we believe this additional control improves clarity, ruling out the possibility of non-specific FcR binding and confirming the observed decrease in CD206-specific staining. We are reporting results from this additional experiment in Figure 2A-B of this rebuttal, but if the referee prefers we can include also these graphs in the revised manuscript.

4) The intravenous transfer of BMDM is a very artificial system that does not recapitulate the *in vivo* situation during tumor growth. Mature macrophages are never in circulation, monocytes are. This may alter important aspects, such extravasation and tumor penetration, intratumoral localization etc.

Response:

We appreciate the reviewer's comment. To address this concern, we now performed orthotopic injections of cancer cells and macrophages together. In details, we performed an orthotopic co-injection experiment, in which prostate cancer tumor cells were injected directly into the prostate together with macrophages, either wild-type (LGP) or knockout (KO) for chemosensor genes (OLFR644 or VMN2R29). Our results confirmed the key role of chemosensor-expressing macrophages in supporting tumor growth. Specifically, when KO macrophages were co-injected with tumor cells, tumor growth was significantly reduced compared to the condition in which WT macrophages were present. Additionally, the tumor microenvironment was reshaped, with a notable increase in CD8⁺ T cell infiltration and decrease of CD4⁺ T regulatory cells (Fig. 4F-I). We believe that these findings reinforce our previous observations in the subcutaneous model and provide additional evidence that chemosensor-expressing macrophages contribute to tumor progression in an orthotopic setting.

We believe this new experiment addresses the referee's request, and the revised manuscript now includes these results. This model better reflects the *in vivo* tumor environment, as it allows for the direct injection of macrophages into the tumor site. However, we would like to emphasize that macrophage cell therapy is currently under investigation in cancer treatment, making our model of macrophage injection highly relevant. Currently, two Phase 1 clinical trials are ongoing and involve the use of genetically modified macrophages as a therapeutic strategy: CAR-Macrophages for the Treatment of HER2-Overexpressing Solid Tumors (ClinicalTrials.gov: NCT04731432) and "Intraperitoneal MCY-M11 (Mesothelin-Targeting CAR) for the Treatment of Advanced Ovarian Cancer and Peritoneal Mesothelioma (ClinicalTrials.gov: NCT04660929)". Although

macrophages are not typically found circulating in the bloodstream, our approach mirrors clinical strategies exploring macrophage-based therapies for tumor targeting and immunomodulation.

5) The GFP signal appears like a continuum, rather than a clear discriminatory marker to discriminate endogenous and transferred macrophages. Hence, based on Figure 4C, it is impossible to say whether the authors really gated on transferred macrophages, or whether transferred macrophages are even there. A more discriminative approach is needed.

Response:

We thank the reviewer for raising this important point. We agree that GFP can be sometime problematic for FACS analysis *ex vivo* and therefore we repeated the experiment to enhance the robustness and clarity of our data. To improve the discrimination between endogenous and transferred macrophages, we repeated the preclinical trial using C57BL/6-CD45.2 donor mice and C57BL/6-CD45.1 recipient mice. We injected macrophages from C57BL/6-CD45.2 mice, in which GFP is expressed under the Ubiquitin promoter, in C57BL/6-CD45.1 recipient mice. As a result, the injected macrophages are GFP⁺CD45.2⁺. This allowed us to track the transferred macrophages by gating on CD45.2, providing a more specific marker. We compared the composition of the microenvironment and macrophage phenotype by gating on GFP⁺ macrophages and CD45.2⁺ macrophages, and we compared results. As shown in figure 1C-G of this rebuttal, results from GFP gated and CD45.2 gated cells are consistent. Importantly, we demonstrated that GFP and CD45.2 colocalize on infused macrophages, confirming the presence of transferred macrophages and validating GFP as a reporter for injected macrophages (Figure 1C of this rebuttal). We believe this approach addresses the concern regarding the use of GFP as a discriminatory marker and strengthens the reliability of our results. If the referee believe this is necessary, we can include these results in the revised manuscript.

6) Fig 4D-E. The changes in the immune infiltrate (at least CD8⁺ and Treg) are rather modest to explain such clear differences in tumor growth. Are there effects on vessel density and maturation, extracellular matrix,...? The importance of anti-tumor immunity should be assessed via the specific depletion of immune cell populations.

Response:

We appreciate the reviewer's comment. To further explore the role of immune cells in tumor growth, we repeated the trial using macrophages deficient for OLF644 and performed specific depletion of CD8⁺ T cells via monoclonal antibodies. The results showed that depletion of CD8⁺ T cells partially reversed the anti-tumoral effects of OLF644^{-/-} macrophages, suggesting a partial involvement of these cells in the observed effects. However, since the effect was not fully reversed, this implies that other immune cells, such as Tregs, which are consistently modulated in our trials, may also play a role in the tumor growth differences (Fig. 4J-L and Extended data 6K-L). Additionally, we investigated the potential impact of vessel density and extracellular matrix changes to better understand the tumor microenvironment dynamics. We performed immunofluorescence analysis on tumor sections from the trial with KO macrophages, but neither vessel density nor collagen abundance were significantly affected, potentially indicating that these two features are not involved in the anti-tumor effects of OLF644^{-/-} macrophages (Figure 2E-F of this rebuttal).

7) Retinoic acids have a multitude of effects on macrophages, which likely go beyond the inhibition of one olfactory receptor. The effects of the KO in Fig 4I are indeed partial, leaving

room for interpretation of what RA's other effects are.

Response:

We appreciate the reviewer's comment and agree that retinoic acids have a wide range of effects on macrophages beyond the inhibition of a single olfactory receptor. We have now set up two additional preclinical trials aimed at dissecting the impact of 13-cRA on macrophage function in our settings. First, we performed a systemic administration experiment, in which tumor-bearing mice were treated with 13-cRA via oral gavage injection (Fig. 1A-B of this rebuttal). This preclinical showed that the administration of 13-cRA led to a significant reduction in tumor growth, even if with only a partial reshaping of the tumor microenvironment (TME). Interestingly, the concomitant administration of 13-cRA and infusion of OLF644^{-/-} macrophages yielded similar results to 13-cRA alone, suggesting that this molecule acts at least partially through the OLF644. If the referee thinks it is useful, we can add these data to the manuscript. In addition, to further investigate the impact of 13-cRA on the OLF644 and on macrophages, we performed an experiment where WT (LGP) macrophages and OLF644^{-/-} (KO) macrophages were exposed to 13-cRA and infused in tumor bearing mice. These arms were compared to mice infused with WT or OLF644^{-/-} untreated macrophages. No difference was observed between KO macrophages alone and KO macrophages treated with the 13-cRA, indicating that 13-cRA acts at least partially through the chemosensor receptor.

We believe that these findings strengthen the therapeutic relevance of our study and give additional information on the impact of 13-cRA on macrophage function through OLF644 inhibition. The revised manuscript now includes these results (Fig. 5H-J).

8) THP-1-derived macrophages are not fully representative for human primary macrophages. The use of monocyte-derived macrophages is preferred.

Response:

We agree with the reviewer's comment and acknowledge the limitations of THP-1-derived macrophages in fully representing human primary macrophages. To strengthen our findings, we repeated key experiments using primary human macrophages obtained from peripheral blood monocytes of healthy donors. Monocytes were isolated from whole blood and differentiated into macrophages with M-CSF (Macrophage colony-stimulating factor) exposure for 1 week. In brief, we assessed the ability of palmitic acid and agonists of the OR51E2 (acetate and propionate) to induce Ca²⁺ flux in OR51E2-proficient and OR51E2-deficient primary macrophages. Flow cytometry analysis revealed an increase in Ca²⁺ levels following compound administration, which was abolished in OR51E2^{-/-} cells (Fig. 7D-F). Additionally, we performed an immunofluorescence analysis to visualize lipid uptake using BODIPY FL C16, a fluorescently labelled palmitic acid analog, in OR51E2-proficient and OR51E2-deficient primary macrophages and we observed a reduction in palmitic acid uptake in OR51E2-deficient cells (Fig 7G-H). Finally, we performed bulk mRNA sequencing of human primary macrophages exposed to palmitic acid. Palmitic acid induced in primary macrophages the upregulation of biological pathways associated with wound healing and downregulation of inflammation-related pathways, such as antigen presentation and T cell activation. These findings suggest activation toward an immunosuppressive phenotype upon palmitic acid exposure (Extended Data 9A-B). Accordingly, we observed increased levels of CD204, CD206, and Arginase 1 in human tumor-conditioned macrophages, while decreasing HLA-DR levels. These changes were abolished in OR51E2-deficient macrophages (Extended Data 9C-F). These results were consistent with our previous observations in THP1, and extended our analysis on the interaction between palmitic acid and the OR51E2. We believe that these results support the robustness of our conclusions.

9) The role of palmitic acid could be much more elaborated on, in an in vivo context. Is this also a ligand for the mouse orthologue? If so, mouse systems could be used to go much deeper into the role of this lipid in vivo.

Response:

We appreciate the reviewer's suggestion to further investigate the role of palmitic acid in vivo. Unfortunately, due to ethical reasons and the lack of a dedicated ethical authorization, we could not perform additional *in vivo* preclinical trials to test the impact of palmitic acid in tumor models. Nevertheless, we gained much deeper insights on human samples and performed an extensive analysis using Hyperion imaging mass cytometry (IMC) and Visium spatial transcriptomics to dissect the immune composition of prostate cancer biopsies, assessing palmitic acid presence in consecutive sections. In details, we performed Hyperion to map the localization of macrophages and other immune cells in tumor sections consecutive to those used for spatial lipidomics analysis of palmitic acid. These data were integrated with H&E staining and immunofluorescence, allowing us to correlate palmitic acid localization with immune cell distribution in the tumor microenvironment. Our analysis revealed an increased abundance of most cell types in tumor palmitic-enriched areas, including conventional T cells (T conv), regulatory T cells (Treg), and both pro-inflammatory (CD206⁻MHCII⁺CD68⁺ M1-like and CD206⁺MHCII⁻CD68⁺ M2-like macrophages) (Table 3 and Extended data 9G). Notably, spatial interaction analysis demonstrated a higher propensity for PanCyK⁺ epithelial cancer cells to interact with M2-like macrophages in palmitic-rich tumor regions compared to PanCyK⁺ epithelial cells in the palmitic deprived areas (Fig. 8D-E). Additionally, we performed Visium spatial transcriptomics on the same tumor sections to explore macrophage heterogeneity and identify transcriptional profiles associated with different macrophage subsets. This analysis provided a deeper understanding of how macrophage populations are spatially organized in relation to lipid accumulation and tumor architecture. Additionally, we explored the interaction between palmitic acid and OR51E2 through in vitro binding assays and immunofluorescence analysis. Briefly, to investigate the engagement of the olfactory receptor OR51E2 by ligands, we employed the Dual-Glo Luciferase Assay System. Specifically, we aimed to assess the receptor's binding with palmitic acid, using sodium acetate and sodium propionate as positive controls, as their interactions with OR51E2 have been previously described. Hana3A cells were transfected with OR51E2 and subsequently stimulated with palmitic acid, sodium acetate, or sodium propionate to induce CRE-luciferase expression. Luminescence was measured four hours post-stimulation, revealing a detectable response to all three fatty acids (Fig. 7B-C and Extended Data 8E). Next, we assessed the ability of these compounds to induce Ca²⁺ flux in OR51E2-proficient and OR51E2-deficient primary macrophages. Flow cytometry analysis revealed an increase in Ca²⁺ levels following compound administration, which was abolished in OR51E2^{-/-} cells (Fig. 7D-F). Finally, we performed an immunofluorescence analysis to visualize lipid uptake using BODIPY FL C16, a fluorescently labelled palmitic acid analog, in OR51E2-proficient and OR51E2-deficient primary macrophages and we observed a reduction in palmitic acid uptake in OR51E2-deficient cells (Fig 7G-H). All these new data are now included in the revised manuscript.

Reviewer #3 (Remarks to the Author):

The manuscript elucidates the unexpected role of chemosensory receptors in macrophage infiltration in cancer and the potential of re-educating tumor-associated macrophages

(TAMs) for immunotherapy. Through genome-wide CRISPR screening, several olfactory and vomeronasal receptors were identified as major drivers of TAMs' tumor-supportive status in prostate cancers. Deletion of one olfactory or one vomeronasal chemoreceptor led to cancer regression and increased infiltration in vivo. Additionally, palmitic acid was found to enhance their pro-tumor phenotype in human prostate cancer tissues, and this effect requires OR51E2. These findings suggest chemoreceptors as new potential therapeutic targets for enhancing antitumor immunity. The work is potentially groundbreaking in the fields of cancer biology, immunology, and chemical senses. However, there are several reservations that the authors need to address:

1. In Figure 2, the authors show that sgRNAs for several chemosensory receptors were significantly over- or under-represented in the in vitro CRISPR screen. There is a need to demonstrate the specificity of chemosensory receptors involved in TAMs. The authors chose one OR and one Vmn2r as representatives, but later show that Or51e2 also has a similar role. Where is Olf78 (Or51e2) CRISPR sgRNA in the volcano plot Fig 2D? As the manuscript stands, it is unclear how many chemosensory receptors are indeed involved. As there are many chemosensory receptors, it would not be feasible to test them all. However, the authors should provide examples of ORs and Vmn2rs that do not show similar effects.

Response:

We thank the reviewer for this comment. Indeed, a key message emerging from our screening strategy is that multiple chemosensory receptors contribute to macrophage activation, with some acting as positive regulators and others as negative regulators. This suggests that macrophages may sense and integrate environmental complexity through this sensory system. As the reviewer pointed out, our screen identified chemosensory receptors with both pro- and anti-tumoral roles. To clarify this further, we have now included a revised volcano plot highlighting chemosensory receptors that were significantly enriched in our CRISPR screen (Extended Data 3H). Additionally, we have indicated the position of OLF78 in the volcano plot, which did not reach statistical significance in our screening (Figure 2D of this rebuttal). However, to explore the functional relevance of OLF78, we used primary murine macrophages and we performed genetic deletion of OLF78. We found that in its absence, macrophages exposed to tumor cell-conditioned media failed to acquire a pro-tumoral phenotype. Specifically, we observed a reduction in the frequency of CD206^{bright}MHCII⁻ pro-tumoral macrophages and an increase in CD206⁻MHCII⁺ pro-inflammatory macrophages in OLF78^{-/-} cells exposed to tumor-conditioned media, compared to untreated controls. Moreover, Arginase 1 levels were decreased in the absence of Olf78 (Extended Data 8D). These results indicate that OLF78, the mouse homolog of human OR51E2, plays a similar role in macrophage polarization. We believe that these findings further support the role of chemosensory signaling in shaping macrophage function and validate the results from our CRISPR screen.

In addition, as suggested by the reviewer, we extended our analysis to other chemosensors involved in macrophage activation. Specifically, we examined OLF229 and VMN1R87 (chosen among the top-ranked positive regulators) and OLF192 (selected among the top-ranked negative regulators). As shown in Extended Data 4C-E, genetic deletion of the additional negative regulator OLF192 impaired macrophage polarization toward a pro-tumoral phenotype, as evidenced by a reduction in CD206^{bright}MHCII⁻ macrophages upon tumor conditioning and an increase in CD206⁻MHCII⁺ pro-inflammatory cells. Conversely, genetic deletion of the positive regulators OLF229 and VMN1R87 did not result in any detectable differences in macrophage phenotype. These findings are in line with our hypothesis and highlight a potential inherent limitation of this type of screening, where assessing the impact of positive regulators can be more challenging.

2. In chemosensory organs, ORs and Vmn2rs couple with different G proteins and their canonical downstream signaling pathways are very different. However, the authors' data appear to show these receptors having the same role in TAMs. The authors should address these apparent discrepancies. Additionally, no functional Vmn2rs is present in humans, which the authors should discuss.

Response:

We thank the reviewer for raising this important point. Our data suggest that in tumor-associated macrophages (TAMs), chemosensory receptors may converge on common regulatory mechanisms. To investigate the signaling pathways downstream of these receptors, we performed bulk mRNA sequencing of macrophages exposed to tumor-conditioned media and compared their transcriptional profiles following genetic deletion of OLFR644 and VMN2R29. Using Ingenuity Pathway Analysis, we identified key transcriptional regulators associated with the differentially regulated pathways. Our analysis revealed that transcription factors such as HIF1 α and cMYC, were inactivated or suppressed in both OLFR644^{-/-} and VMN2R29^{-/-} macrophages (Extended Data 4F-G). Notably, HIF1 α and cMYC have been previously described as key modulators of pro-tumoral macrophage activation (PMID: 20841473 and PMID: 22067385). To further explore this, we performed qRT-PCR to assess the expression of known HIF1 α target genes. Our preliminary data indicate that HIF1 α signaling is indeed downregulated in the absence of these chemosensory receptors. These findings suggest the existence of common downstream pathways regulated by OLFR644 and VMN2R29, despite their distinct canonical signaling mechanisms in chemosensory organs. While these data provide an initial indication of shared molecular circuits, further investigation will be necessary to fully elucidate the interplay between chemosensory receptors and transcriptional regulators in TAMs.

Regarding the absence of functional vomeronasal genes in humans, we acknowledge this important evolutionary difference. While human macrophages lack functional Vmn receptors, our findings suggest that other chemosensory receptors, such as OR51E2, may play a compensatory role in human macrophages. We now discuss this point in the discussion section of the revised manuscript, highlighting the potential for species-specific differences in chemosensory signaling while maintaining the concept that macrophages utilize these receptors to interpret the complex environment and modulate their activation states.

3. The three chemoreceptors are likely to bind distinct ligands, yet the role of these and potentially additional chemoreceptors in functioning TAMs appears to be equivalent. Yet, in the section involving Olfr78/Or51e2, the authors appear to propose a novel ligand. The author should clarify this.

Response:

We thank the reviewer for this comment. We acknowledge that different chemosensors may bind distinct ligands, and the situation is indeed complex. For this reason, we decided to focus on the human context, given its translational relevance. Specifically, we thought it would be important to identify a key ligand for human OR51E2, which is notably overexpressed in human prostate cancer samples. This led us to dedicate significant effort to detecting palmitic acid and investigating its role in human cancer, as we believe this could offer important insights into tumor biology and potential therapeutic impact. We now clarified this concept in the discussion section of the revised manuscript.

4. The authors should note that OR51E2 and OR51E1 (they are called PSGRs) have been previously reported to be over-represented in prostate cancers.

Response:

We thank the reviewer for this comment. We are aware that these two olfactory receptors have already been described in the literature. We have now clarified this point in the revised manuscript.

5. Propionate and acetate are well-established ligands for OR51E2, whereas medium-chain fatty acids do not activate OR51E2. Therefore, palmitic acid as a novel ligand for OR51E2 is somewhat unexpected. The authors should demonstrate palmitic acid-mediated OR51E2 activation, perhaps in the heterologous system. In addition, the authors should use propionate or acetate as a benchmark in their experiments characterizing the role of OR51E2.

Response:

We appreciate the reviewer's insightful comment. To address this concern, we have set up a series of new experiments to further investigate the role of palmitic acid as a potential ligand for OR51E2. These experiments include testing the activation of OR51E2 by palmitic acid, where we assessed receptor activation using calcium mobilization assays and luciferase reporter assays. Additionally, we compared the effects of palmitic acid with those of known ligands for OR51E2, such as propionate and acetate, as a benchmark in our experiments. These new data allow us to better understand the potential activation of OR51E2 by palmitic acid and its biological relevance in the context of human prostate cancer. In details, we explored the interaction between palmitic acid and OR51E2 through in vitro binding assays and immunofluorescence analysis. Briefly, to investigate the engagement of the olfactory receptor OR51E2 by ligands, we employed the Dual-Glo Luciferase Assay System. Specifically, we aimed to assess the receptor's binding with palmitic acid, using sodium acetate and sodium propionate as positive controls, as their interactions with OR51E2 have been previously described. Hana3A cells were transfected with OR51E2 and subsequently stimulated with palmitic acid, sodium acetate, or sodium propionate to induce CRE-luciferase expression. Luminescence was measured four hours post-stimulation, revealing a detectable response to all three fatty acids (Fig. 7B-C and Extended Data 8E). Next, we assessed the ability of these compounds to induce Ca^{2+} flux in OR51E2-proficient and OR51E2-deficient primary macrophages. Flow cytometry analysis revealed an increase in Ca^{2+} levels following compound administration, which was abolished in OR51E2^{-/-} cells (Fig. 7D-F). Finally, we performed an immunofluorescence analysis to visualize lipid uptake using BODIPY FL C16, a fluorescently labelled palmitic acid analog, in OR51E2-proficient and OR51E2-deficient primary macrophages and we observed a reduction in palmitic acid uptake in OR51E2-deficient cells (Fig 7G-H). All these new data are now included in the revised manuscript.

6. The nomenclature of ORs in humans vs. mice is confusing. The use of a human-centric nomenclature system, for example, Or51e2 instead of Olfr78, is recommended to minimize confusion.

Response:

We thank the referee for this comment. In response, we have now included the human nomenclature alongside the murine nomenclature where appropriate throughout the manuscript.

A
In vivo 13cRA systemic administration

B

C

D

E

F

G

Figure 1 of the rebuttal: A) Tumor growth is expressed as a percentage of the initial volume. Mice were subcutaneously injected with $Pten^{-/-}$; $Trp53^{-/-}$ tumor cells and received daily oral gavage with either vehicle or 13-cRA. A third group was intravenously injected with OLF644 $^{-/-}$ macrophages followed by 13-cRA gavage. B) FACS analysis to determine the immune infiltrate: $CD206^{Bright}MHCII^{-}$ and $CD206^{-}MHCII^{+}$ are gated on $F4/80^{+}CD11b^{+}$ cells; $CD39^{+}$ cells are gated on $CD8^{+}$ T cells and $FoxP3^{+}CD25^{+}$ T reg cells are gated on $CD4^{+}$ T lymphocytes. C). We injected macrophages from C57BL/6-CD45.2 mice, in which GFP is ubiquitinally expressed, in C57BL/6-CD45.1 recipient mice. The FACS plot show the overlapping of GFP and CD45.2 signal (conversely, CD45.1 do not overlap with GFP). D-G) Mice were injected IV with LGP-Macs or OLF644 $^{-/-}$ Macs pre-treated for 4 hours with DMSO or 13-cRA. The bar graphs show the FACS analysis to determine the immune infiltrate: D) Cells are gated on $GFP^{+}F4/80^{+}CD11b^{+}$ cells E) Cells are gated on $GFP^{-}F4/80^{+}CD11b^{+}$ cells F) Cells are gated on $CD45.2^{+}F4/80^{+}CD11b^{+}$ cells G) Cells are gated on $CD45.2^{-}F4/80^{+}CD11b^{+}$ cells. Statistical analyses were performed using two-tailed unpaired Student's t test. Values are presented as mean \pm SEM (*, $P < 0.05$; **, $P < 0.01$; ***, $P < 0.001$).

Figure 2 of the rebuttal: A) Representative FACS plot showing CD206 expression and its isotype control (IgG2a, κ) on LGP (CTRL) and Stat6 KO (g1 and g2) macrophages. Macrophages were either untreated or stimulated with conditioned media from Pten^{-/-}; Trp53^{-/-} cells. B) Bar graph showing the quantification of the MFI of the isotype (anti IgG2a, κ), CD206 at basal level (untreated macrophages), CD206 after exposure to conditioned media and the fold change of treated versus untreated macrophages. The fold change was calculated as follow: $nMFI_{cm}/nMFI_{untr}$ where nMFI is the MFI of CD206-MFI of the isotype. C) Representative plot showing the screening sorting strategy. We sorted four populations: MHCII⁺, CD206^{Low}, CD206^{Mid} and CD206^{Bright}. D) Volcano plot showing Olfr78 related to the differentially enriched sgRNA guides from CD206⁻ MHCII⁺ vs CD206^{bright} MHCII⁻ cells. E-F) Immunofluorescence showing CD31 (red) and Collagen I (white) on tumor sections of mice injected with LGP, VMN2R29^{-/-} and OLFR644^{-/-} macrophages. Results are expressed as mean of fluorescence per field. Statistical analyses were performed using two-tailed unpaired Student's t test. Values are presented as mean \pm SEM (*, P<0.05; **, P<0.01; ***, P<0.001).

Revision Nature Immunology NI-A37606

Title: A genome-wide CRISPR screening reveals chemosensor receptors as lipid-detecting regulators of macrophage functions in cancer

To the Editor of Nature Immunology and the referees

We appreciate that both Referee #1 and #2 recognized the improvements of the manuscript and are satisfied with the experiments performed and that Referee #3 agreed with our screening strategy. I'm including below a brief (250 words max) summary of the paper as requested:

“The authors show that olfactory receptors in tumor-associated macrophages sense lipids in the tumor microenvironment, promoting cancer growth. Targeting chemosensors reprograms macrophages and enhances anti-tumor immunity across cancers”

Here my account Twitter for the diffusion of the manuscript, as requested: @DDiMitre_Lab

Also, please find below our point-by-point response to the referees.

Referee #1

The authors significantly improved the manuscript following the first round of revisions particularly strengthening their claims with the generation of additional data sets using (sc)RNAseq and spatial transcriptomic approaches. The authors strengthened the rationale of their study by expanding on the characterization of macrophage heterogeneity in mouse models of prostate cancer. Additionally, this allowed the authors to reinforce the relevance of the described TAM phenotypes in vitro that were used as a readout for the genetic screening. In the revised version, the in vivo validation of targets from the genetic screen is particularly improved, confirming the significance of the findings in this manuscript and their potential therapeutic impact. However, the main limitation of the manuscript remains the design of the genetic screen that lacks guidance by a clear hypothesis, resulting in a rather broad and untargeted, lacking controlled approach. This may have complicated the choice of targets, potentially undermining the significance of the results and making it difficult to justify the choice of employing this approach. This is also demonstrated by the lack of specificity of the selected target genes. An example that highlights this shortcoming is the identification of relevant olfactory receptors in human prostate cancer where the authors show that OR51E2 is highly upregulated with OLF78 being the corresponding mouse orthologue. However, OLF78 was not identified or chosen as one of the positive regulators in the genetic screen. This also resulted in the human cellular experiments being conducted based on OR51E2 deletion which, technically, was not a chosen target from the screen. Therefore, it is unclear how the choice of validated targets was guided, and it raises the questions regarding whether more relevant targets could also be identified using a more simplified approach.

Nevertheless, the authors have significantly enriched the revised manuscript with valuable data and insights into macrophage functionality in prostate cancer. Additionally, the authors demonstrate strong validation in mouse models and human cellular systems, which is a clear strength of the manuscript. Yet, the technical execution and design of the CRISPR screen as well as the choice of targets could have been refined and revisited, as suggested in the first round of reviews, ultimately leaving some flaws in this study unaddressed.

Response:

We appreciate that Reviewer #1 acknowledged our efforts in the revision process and recognized the significant improvements in validating both target genes and mechanisms in mouse models and human settings.

However, we respectfully disagree with the reviewer's assessment of our screening approach. The reviewer states that "the main limitation of the manuscript remains the design of the genetic screen that lacks guidance by a clear hypothesis." However, our approach was guided by a well-defined rationale in the field of macrophage re-education. Macrophages in the tumor microenvironment exhibit plasticity, acquiring diverse functional states in response to environmental factors. These states can be summarized into two extremes: CD206^{bright}MHCII⁻ pro-tumoral macrophages and CD206^{neg}MHCII⁺ anti-tumoral macrophages. Certainly these represent extreme phenotypes, as it is widely accepted in the literature that macrophage states exist along a continuum. A major goal in macrophage-targeted cancer therapy is the identification of molecular targets capable of shifting macrophage polarization from such pro-tumoral to the anti-tumoral state. In our screening approach, we compared these two functional states to identify genes whose modulation could induce this switch. As indicated in our rebuttal, we then also included all possible comparisons in our analysis for completeness:

- 1- CD206^{bright}MHCII⁻ vs CD206^{neg}MHCII⁻;
- 2- CD206^{bright}MHCII⁻ vs CD206^{mid}MHCII⁻;
- 3- CD206^{bright}MHCII⁻ vs CD206^{neg}MHCII⁺;
- 4- CD206^{mid}MHCII⁻ vs CD206^{neg}MHCII⁻;
- 5- CD206^{mid}MHCII⁻ vs CD206^{neg}MHCII⁺;
- 6- CD206^{neg}MHCII⁻ vs CD206^{neg}MHCII⁺.

All data are available in the first rebuttal.

I believe that our strategy followed a strong rationale, generally accepted in our field of research.

Regarding the reviewer's suggestion of a targeted screening strategy, we believe that hypothesis-free, genome-wide genetic screenings play a crucial role in uncovering previously unknown molecular mechanisms (PMID: 37214176). Such approaches are essential for identifying novel genes, pathways, and regulatory networks that might otherwise remain undiscovered, and this actually represents the beauty of this approach. Our decision to employ a genome-wide CRISPR screen was driven by the need to explore macrophage functional regulators in an unbiased manner, without restricting the search to preselected candidates. In fact, one of the main challenges with macrophage-based therapies today is that, as monotherapies, none have shown strong clinical efficacy in trials. This highlights the urgent need for additional strategies. While we agree that targeted screens have clear advantages in terms of precision, they inherently limit discoveries to known pathways, potentially overlooking unexpected but biologically relevant mechanisms. In our study, a targeted screen would have probably not led to the discovery of chemosensory and sensing receptors as key regulators of macrophage function in cancer, which is the central finding of our paper.

The reviewer's concern regarding the absence of OLF78 in our CRISPR ranking is understandable and was also a point of consideration for us. However, it is well recognized that genome-wide screens can yield false negatives (and false positives) due to statistical limitations (PMID: 36727483) as in any similar technology, and validation is a must do to compensate this issue. This is the reason why we extensively validated the targets discovered by the screen by means of *in vitro* and *ex vivo* approaches that were appreciated by the referees. I would like to highlight that we validated *in vitro* the top ranked gene, Stat6, and 5 additional chemosensors among the positive and negative regulators. I believe that this validation proves the key role of the chemosensing in macrophage biology.

On this line, the fact that OLF78 was not highly ranked does not negate its biological relevance. Indeed, our independent knockout experiments confirmed its pro-tumoral role in macrophages. This suggests that its low statistical ranking in the screen may have been due to a reduced effect size rather than a true lack of function. Additionally, for human validation, we prioritized OR51E2, a chemosensory receptor highly expressed in the

context of prostate cancer. This choice was made to enhance the translational value of our findings. Notably, there are no direct human orthologues for our main targets, Olf644 and Vmn2r29, further supporting the need to identify functionally relevant human counterparts. Finally, through the revision process, we further strengthened the evidence that multiple olfactory receptors converge on shared downstream pathways in macrophages. Therefore, this might explain why different chemosensory receptors, including OR51E2 in humans and our selected mouse olfactory receptors, exhibit similar functional roles in macrophages.

In conclusion, while we understand the reviewer's point of view, we strongly stand by our rationale and screening approach. We believe that our extensive validation experiments in mouse and human demonstrate the robustness of our strategy and confirm the relevance of the selected target.

Reviewer #2

The authors sufficiently addressed my concerns

We thank the referee for her/his comments which have significantly strengthened the manuscript.

Reviewer #3

Following review of the paper and both the reviewer comments and the authors rebuttal I am in favor of accepting the manuscript. The basis of the paper is taking an unbiased approach to identify new factors associated with tumor associated macrophage phenotypes and therefore their screen as presented makes sense. While it is unfortunate that OLFR78 did not emerge as a top hit they did go on to show that individual removal of this did impact the phenotype and therefore there are a number of reasons it might not have emerged in the screen (for example the sgRNAs did not target it efficiently. Their subsequent screen hits such as chemosensory receptors OR51E2 are followed up and validated which argues for the power of the screening approach to identify novel regulators.

Responses to reviewer 3 comments are satisfactory.

In Figure 7 they use siRNA in primary cells. But in their legend they use the term "partial genetic deletion". This is not accurate and should be described as RNA knockdown.

One small additional point is in Figure 3 F they show a venn diagram. It would be preferable to have this be correctly sized to reflect the numbers. This would be more accurate and in keeping with the highly technical figures.

We thank the referee for his/her comments. We now modified the legend in Figure 7 in accordance with the request. Also, VENN diagram has been sized to reflect the numbers.